# A novel criteria on exponentially passive analysis for Takagi-Sugeno fuzzy of neutral dynamic system with various time-varying delays

Janejira Tranthi, Thongchai Botmart 🔟*

Department of Mathematics, Khon Kaen University, Khon Kaen, Thailand

* thongbo@kku.ac.th

## Abstract

This paper is the first studying on designing exponentially passive analysis for T-S fuzzy of dynamic systems with various time-varying delays such as neutral, discrete, and distributed time-varying delays. Constructing the new Lyapunov-Krasovskii function and the Newton-Leibniz theory, the zero equations, and the matrix inequality techniques, the multiple delay-dependent criteria, with assuring exponentially passive on the discussed T-S fuzzy system, are defined in respect of linear matrix inequalities (LMIs) that can be checked easily using the LMI toolbox of MATLAB. Those approaches give less conservative, exponentially passive criteria for special cases of general stability of a neutral differential system. Furthermore, the results of this study are delay-dependent, which depend on the lower and upper bound with the time-varying delay. Lastly, some numerical examples illustrate the performance of our criteria based on the results obtained and summarize some of the previous achievements.

## 1 Introduction

The research of Takagi and Sugeno created the Takagi-Sugeno (T-S) fuzzy system [1], which explained the time-delays frequently occurring in many dynamic systems, practically (e.g., biological systems neural, networks, metallurgical processes, and chemical processes). The researchers stated to handling with the synthesis and analysis problems of nonlinear systems can be proven by the fuzzy-logic theory. Especially, the T-S fuzzy model uses a set of IF-THEN rules built on linguistic variables and values by quantifying the semantics of linguistic values using a member function. In consequence, the analysis and class synthesis of non-linear systems, and many nonlinear analytical problems with traditional linear system theories were studied based on this fuzzy model of T-S. For instance, Zhang et al. [2] presented guarantee cost network control method of the T-S fuzzy systems with delay on the neural networks. Li et al. [3] were investigated the stability of an unstable randomized neural network for

**Data Availability Statement:** All relevant data are within the paper.

**Funding:** This research has received funding support from the NSRF via the Program

Management Unit for Human Resources &
Institutional Development, Research and
Innovation [grant number B05F640088].

**Competing interests:** The authors have declared
that no competing interests exist.

mixed-delayed neutral types. Moreover, Li et al. [4] demonstrated a stabilization and exponential stability analysis issue of T-S fuzzy systems under periodic sampling as well. Xu at al. [5] presents stability of uncertain systems, which the stability of the discrete singular fuzzy system at discrete time.

Time delays is of significance both in theory and application due to its detrimental effects on stability and performance of systems and its wide existence in practical dynamic processes. The cases of delays can be usually considered as time delay, multiple delays, interval delay, input and state delays and so on. All of them were discussed around two basic group, i.e. delay-independent and delay-dependent. The delay-dependent stability criterion are investigated with the extent upper bound of delay. Hence, the criterion of delay-dependent stabilization are proposed to guarantee that the delay system is stable for any value of time delay less than the provided upper bound. In different circumstances, the delay-independent stability criteria are proposed without consideration of the extent for time delay. In ordinary, the delay-dependent conditions are preferable than the delay-independent conditions while the effect of time delay is not acute. According to Zhu and Yang [6] illustrated Jensen's inequality approach in synthesis the stability for continuous time systems with time-varying delay. The study all of delay, which defined by Lien [7], guarantee cost control for uncertain neutral system through the LMI system. Likewise, Chen et al. [8] applied guarantee cost control of T-S fuzzy system with input delays and state. The research of Lien et al. [9] supported the stability criterion of interval time-varying delay systems during the uncertain T-S fuzzy systems. According to Jiang and Han [10] researched the delay-dependent criterion of uncertain system with time-varying delays. However, the above mentioned, there is still room for further improvement: the fuzzy T-S method with delay-dependent based on latency to the possible extent of the thresholds for exponential stability and passivity performance.

In addition, passivity theory is another proficient tool for analyzing system stability. The passive theory is the main pointed to the system can keep the system's internal stability which is the passive properties. So that, the problem of inactivity is therefore an important part of recent research. Then, the passive control uses the product of output and input as the power rating, which captures the attenuation properties of the system under the bounded input. In particular, passivity theory is more general than stability theory because it can be illustrated Lyapunov function under the theory of stabilization. This theory is used for issue of engineering i.e. electric circuits and heat energy systems. Nowadays many researchers have studied passive theory and passive control problems extensively, for instance, Zhang et al. [11]who studied the passive controller design issue with both state and input delays for a class of continuous-time T-S fuzzy systems. Another researcher such as Wu et al. [12] identified the problem of passive control for fuzzy network systems, considering the random uncertainty variable sampling interval and the delay caused by the fixed network. Similarly, Song and He [13] who researched the robust passive control is offered for a limited time for nonlinear systems with time-delay. The studied of Yu et al. [14] focused both passive analysis and passive control for erratic intermittent switching delay systems through a simple switching signal design. Likewise, Yotha et al. Improved delay-dependent approach to passivity analysis for uncertain neural networks with discrete interval and distributed time-varying delays [15]. So, it is challenging to solve exponentially passive for T-S fuzzy of neutral dynamic system with various time-varying delays.

Despite, in a specific physical system, mathematical models are described by functional differential equations of the neutral type. The neutral type of functional differential equation depends on the lag of the state and state derivatives. Approximately, neutral-type phenomena often appear in automatic control studies, chemical reactor, distributed network, the dynamic

process such as steam pipes and water pipes. Also, population ecology, heat exchange, microwave oscillator, turbojet engine system, lossless transmission line, vibrating mass attached to an elastic band, etc. Likewise, the research of Zhou et al. [16] examined the problem of adaptive synchronization for neutral type random neural networks with Markovian switching parameters. Chen et al. [17] supported that stability of global exponential in mean squares and exponential stability are almost certain for randomly delayed neural networks, and in term of neutral differential system with stochastic effects stated by Arthi et al. [18]. Moreover, Zhu et al. [19] investigated the synthesis of stability neutral system with distributed and discrete time delays. According to [20–23] have illustrated the stability criteria for the neutral neural network with Marcovian jump parameters and mixed time delays. Therefore, passive analyzes for neutral neural networks have been discussed in the last few years. For instance, the studied of Balasubramaniam [24] demonstrated inertia analysis for neutral neural networks with Marcovian jump parameters and time leakage delay term. According to Samidurai [25] analyzed of passivity with mixed and leakage delays for neutral-type neural networks. Unfortunately, the exponential analysis of stability and the passivity performance of a neutral differential system with a time delay is of little concern, in nowadays.

According to the above discussion, this research the exponentially passive analysis was considered for the class of uncertainty neutral fuzzy differential systems generated by the Lyapunov-Krasovkii functional (LKF) method. Also, the systems created by stability theory and integral inequality techniques. The all of delays consist of discrete, neutral, and distributed delays that vary with time. In addition, this research offers a new approach to the resulting manipulation of exponential and inertial steady-states and more efficient compared to existing methods. The following topics to promote a clear understanding and objectives of this study are given

- This study is the first ever exponentially passive analysis for Takagi-Sugeno fuzzy system of a dynamic system (1) consisting of time-varying, discrete, neutral, and distributed delays.

- Especially, if $C_i + \Delta C_i(t) = 0$, $D_i + \Delta D_i(t) = 0$ and $E_i + \Delta E_i(t) = 0$, The system (1) becomes the T-S fuzzy of differential system presented by Fang Liu, et al. [26], Li et al. [27], Lien et al. [7, 28] and Pin-Lin Liu [29].

- Over and above, if $D_i + \Delta D_i(t) = 0$ and $E_i + \Delta E_i(t) = 0$, Also, the system (1) becomes the T-S fuzzy of neutral differential system presented by Ding et al. [30]. Then, the system (1) is more advanced differential replica than the former times.

- This study attained exponential stability of the T-S fuzzy system, where the upper boundary of delay was more effective than other studies. This present in the Examples 3, and Examples 5 with uncertain conditions.

- Some methods and a new LKF have been presented to achieve the exponentially passive benchmarks of T-S fuzzy for uncertain dynamic systems with range discrete, neutral range and distributed time-delay.

- Lastly, for the first time, an improved Wirtinger inequality, a new triple integral inequality, and zero equation together with convex combination approach are used in this work; as a result, we obtain more general results and maximum allowable delay bounds greater than in previous literature [7, 26–30].

**Remark 1** *This study constructs the suitable Lyapunov-Krasovskii functional, which consists of single, double, triple, and quadruple integral terms containing information about the lower and upper bounds of the delays $\sigma_2$, $\tau_2$ and a state $x(t)$. Furthermore, the LKF contains new triple*

*integral terms as follows*:

$$\sigma_2^2 \int_{-\sigma_2}^0 \int_\theta^0 \int_{t+\beta}^t e^{2\alpha(s-t)} \dot{x}^T(s) W_1 \dot{x}(s) ds d\beta d\theta,$$

$$\tau_2^2 \int_{-\tau_2}^0 \int_\theta^0 \int_{t+\beta}^t e^{2\alpha(s-t)} \dot{x}^T(s) W_2 \dot{x}(s) ds d\beta d\theta$$

*and new quadruple integral terms*

$$\sigma_2^3 \int_{-\sigma_2}^0 \int_v^0 \int_\theta^0 \int_{t+\beta}^t e^{2\alpha(s-t)} \dot{x}^T(s) U_1 \dot{x}(s) ds d\beta d\theta dv,$$

$$\tau_2^3 \int_{-\tau_2}^0 \int_v^0 \int_\theta^0 \int_{t+\beta}^t e^{2\alpha(s-t)} \dot{x}^T(s) U_2 \dot{x}(s) ds d\beta d\theta dv$$

*that do not appear in* [7, 26–30]. *These improvement techniques enhance to get better results.*

Henceforward, this study is divided into 5 Section: Section 2, the generalization for neutral differential of fuzzy replica is defined, and definitions and lemmas. Section 3, the exponentially passive criteria for the generalized fuzzy of dynamical system and will present a special case of the generalized fuzzy of neutral differential system. Section 4 will illustration the numerical examples to indicate the exponentially passive for the common fuzzy of dynamical systems. This includes the special case of the general phase value system for the neutral dynamic system. Lastly, Section 5.

## 2 Problem statement and preliminaries

Consider Takagi-Sugeno fuzzy of the neutral dynamic system with time-varying delays of the ensuing form:

Rule $i$: if $\kappa_1(t)$ imply $\mu_{i1}$ and ... and $\kappa_P(t)$ imply $\mu_{ip}$ hence

$$\begin{cases} \dot{x}(t) & = (A_i + \Delta A_i(t))x(t) + (B_i + \Delta B_i(t))x(t - \sigma(t)) \\ & \quad + (C_i + \Delta C_i(t))\dot{x}(t - \tau(t)) + (D_i + \Delta D_i(t)) \int_{t-h(t)}^t x(s)ds \\ & \quad + (E_i + \Delta E_i(t))u(t) \\ z(t) & = \tilde{A}_i x(t + \tilde{B}_i x(t - \sigma(t))) + \tilde{E}_i u(t) \\ x(t) & = \varphi(t), t \in [-n, 0], n = \max\{\tau_2, \sigma_2, h_2\}, \end{cases}$$

where $\mu_{ij}, i = 1, 2, \ldots, r, j = 1, 2, \ldots, p$ implies the set for fuzzy, $x(t) \in \mathbb{R}^n$ implies the state vector, $u(t) \in \mathbb{R}$ stands for the external inputs, $z(t) \in \mathbb{R}$ is the output of the system, $A_i, B_i, C_i, D_i, E_i, \tilde{A}_i, \tilde{B}_i, \tilde{E}_i$ implies constant matrices, constant $r$ implies the amount of IF-Then rule, $\kappa_1(t), \kappa(t), \ldots, \kappa_P(t)$ implies premise variables. $\tau(t), \sigma(t)$ and $h(t)$ implies neutral discrete and distributed interval time-varying delays, successively, agreeable

$$0 \le \tau_1 \le \tau(t) \le \tau_2, \quad \dot{\tau}(t) < \tau_d,$$
$$0 \le \sigma_1 \le \sigma(t) \le \sigma_2, \quad \dot{\sigma}(t) < \sigma_d,$$
$$0 \le h_1 \le h(t) \le h_2, \quad \dot{h}(t) < h_d.$$

Furthermore, $\Delta A_i(t), \Delta B_i(t), \Delta C_i(t), \Delta D_i(t)$ and $\Delta E_i(t)$ implies the terms of uncertain on system

and specify

$$[\Delta A_i(t) \quad \Delta B_i(t) \quad \Delta C_i(t) \quad \Delta D_i(t) \quad \Delta E_i(t)]$$
$$= F_i G(t) [H_{1i} \quad H_{2i} \quad H_{3i} H_{4i} \quad H_{5i}],$$

where $F$, $H_{1i}$, $H_{2i}$, $H_{3i}$, $H_{4i}$ and $H_{5i}$ are known constant matrices and $G(t)$ is a real-unknown matrix function, agreeable,

$$G^T(t)G(t) \le I, \quad \forall t,$$

when $I$ is a suitable dimension identity matrices. By fuzzy blending, the entire fuzzy replica is compiled as following:

$$
\begin{cases}
\dot{x}(t) & = \dfrac{1}{\sum\limits_{i=1}^{r} w_i(\theta(t))} \sum\limits_{i=1}^{r} w_i(\theta(t))[(A_i + \Delta A_i(t))x(t) \\
& \quad + (B_i + \Delta B_i(t))x(t - \sigma(t)) + (C_i + \Delta C_i(t))\dot{x}(t - \tau(t)) \\
& \quad + (D_i + \Delta D_i(t)) \int_{t-h(t)}^{t} x(s)ds + (E_i + \Delta E_i(t))u(t)] \\
& = \sum\limits_{i=1}^{r} \rho_i(\theta(t))[(A_i + \Delta A_i(t))x(t) + (B_i + \Delta B_i(t))x(t - \sigma(t)) \\
& \quad + (C_i + \Delta C_i(t))\dot{x}(t - \tau(t)) + (D_i + \Delta D_i(t)) \int_{t-h(t)}^{t} x(s)ds \\
& \quad + (E_i + \Delta E_i(t))u(t)] \\
& = Ax(t) + Bx(t - \sigma(t)) + C(\dot{x} - \tau(t)) + D\int_{t-h(t)}^{t} x(s)ds + Eu(t) \\
z(t) & = \dfrac{1}{\sum\limits_{i=1}^{r} w_i(\theta(t))} \sum\limits_{i=1}^{r} w_i(\theta(t))[\tilde{A}_i x(t) + \tilde{B}_i x(t - \sigma(t)) + \tilde{E}_i u(t)] \\
& = \sum\limits_{i=1}^{r} \rho_i(\theta(t))[\tilde{A}_i x(t) + \tilde{B}_i x(t - \sigma(t)) + \tilde{E}_i u(t)] \\
& = \tilde{A}x(t) + \tilde{B}x(t - \sigma(t)) + \tilde{E}u(t) \\
x(t) & = \varphi(t), t \in [-n, 0], n = \max\{\tau_2, \sigma_2, h_2\},
\end{cases}
\tag{1}
$$

where $\theta = [\theta_1, \theta_2, \ldots, \theta_p]$, $w_i : \mathbb{R}^p \to [0, 1]$, $i = 1, \ldots, r$ implies membership function for

system which agreeable the rule $i$, $\rho_i(\theta(t)) = w_i(\theta(t))/\sum_{i=1}^{r} w_i(\theta(t))$ and

$$
\begin{aligned}
A &= \sum_{i=1}^{r}\rho_i(\theta(t))(A_i + \Delta A_i(t)), \quad B = \sum_{i=1}^{r}\rho_i(\theta(t))(B_i + \Delta B_i(t)), \\
C &= \sum_{i=1}^{r}\rho_i(\theta(t))C_i + \Delta C_i(t)), \quad D = \sum_{i=1}^{r}\rho_i(\theta(t))(D_i + \Delta D_i(t)), \\
E &= \sum_{i=1}^{r}\rho_i(\theta(t))(E_i + \Delta E_i(t)), \quad \tilde{A} = \sum_{i=1}^{r}\rho_i(\theta(t))\tilde{A}_i, \\
\tilde{B} &= \sum_{i=1}^{r}\rho_i(\theta(t))\tilde{B}_i, \quad \tilde{E} = \sum_{i=1}^{r}\rho_i(\theta(t))\tilde{E}_i
\end{aligned}
$$

It is observed as to the fuzzy weighting function $\rho_i(\theta(t))$ agreeable

$$
\rho_i(\theta(t)) \geq 0, \sum_{i=1}^{r}\rho_i(\theta(t)) = 1.
$$

**Remark 2** *In the uncertain fuzzy differential system, the interval time delay ($\sigma_1 \leq \sigma(t) \leq \sigma_2$) is considered to be longer than the constant time delay ($\sigma(t) = \sigma_2$) and bounded time-varying delay ($0 \leq \sigma(t) \leq \sigma_2$). Then the system* (1) *is more general.*

**Definition 1** [31] *The system* (1) *is exponentially passive from input $u(t)$ to out put $z(t)$, if there is a Lyapunov function $V(t)$ and positive real number $k$ satisfy:*

$$
\dot{V}(t) + kV(t) \leq 2z(t)u(t), \quad t \geq 0,
$$

*for all $u(t)$, all initial condition $X(t_0)$.*

**Lemma 1** [32] *Let any $A \in \mathbb{R}^{n \times n}$ is positive definite constant matrices, $0 \leq g_1 \leq g(t) \leq g_2$, vector function $\omega : [-g_2, 0] \to \mathbb{R}^n$ hence the integration connected are defined, so*

$$
-[g_2 - g_1]\int_{-g_2}^{-g_1} y^T(s)Ay(s)ds
$$
$$
\leq -\int_{-g(t)}^{-g_1} y^T(s)ds A \int_{-g(t)}^{-g_1} y(s)ds - \int_{-g_2}^{-g(t)} y^T(s)ds A \int_{-g_2}^{-g(t)} y(s)ds.
$$

**Lemma 2** [33, 34] *Let $R \in \mathbb{R}^{n \times n}$ is positive definite matrix, for any continuously differentiable function $z : [\alpha_1, \alpha_2] \to \mathbb{R}^n$, the following inequality holds:*

$$
\int_{\alpha_1}^{\alpha_2} \dot{z}^T(t)R\dot{z}(t)ds \geq \frac{1}{\alpha_2 - \alpha_1}X_1^T RX_1 + \frac{3}{\alpha_2 - \alpha_1}X_2^T RX_2 + \frac{5}{\alpha_2 - \alpha_1}X_3^T RX_3
$$
$$
+ \frac{7}{\alpha_2 - \alpha_1}X_4^T RX_4
$$

*where*

$$X_1 = z(\alpha_2) - z(\alpha_1), \qquad X_2 = z(\alpha_2) + z(\alpha_1) - \frac{2}{\alpha_2 - \alpha_1} \int_{\alpha_1}^{\alpha_2} z(s) ds,$$

$$X_3 = z(\alpha_2) - z(\alpha_1) + \frac{6}{\alpha_2 - \alpha_1} \int_{\alpha_1}^{\alpha_2} z(s) ds - \frac{12}{(\alpha_2 - \alpha_1)^2} \int_{\alpha_1}^{\alpha_2} \int_s^{\alpha_2} z(u) du ds,$$

$$X_4 = z(\alpha_2) + z(\alpha_1) - \frac{12}{\alpha_2 - \alpha_1} \int_{\alpha_1}^{\alpha_2} z(s) ds + \frac{60}{(\alpha_2 - \alpha_1)^2} \int_{\alpha_1}^{\alpha_2} \int_s^{\alpha_2} z(u) du ds$$

$$- \frac{120}{(\alpha_2 - \alpha_1)^3} \int_{\alpha_1}^{\alpha_2} \int_s^{\alpha_2} \int_u^{\alpha_2} z(v) dv du ds.$$

**Lemma 3** [35] *Let* $R \in \mathbb{R}^{n \times n}$ *is positive definite matrix, for any continuously differentiable function* $z : [\alpha_1, \alpha_2] \to \mathbb{R}^n$, *the following inequality holds*:

$$\int_{\alpha_1}^{\alpha_2} \int_s^{\alpha_2} \dot{z}^T(u) R \dot{z} du ds \geq 2X_1^T R X_1 + 4X_2^T R X_2 + 6X_3^T R X_3,$$

*where*

$$X_1 = z(\alpha_2) - \frac{1}{\alpha_2 - \alpha_1} \int_{\alpha_1}^{\alpha_2} z(s) ds,$$

$$X_2 = z(\alpha_2) + \frac{2}{\alpha_2 - \alpha_1} \int_{\alpha_1}^{\alpha_2} z(s) ds - \frac{6}{(\alpha_2 - \alpha_1)^2} \int_{\alpha_1}^{\alpha_2} \int_s^{\alpha_2} z(u) du ds,$$

$$X_3 = z(\alpha_2) - \frac{3}{\alpha_2 - \alpha_1} \int_{\alpha_1}^{\alpha_2} z(s) ds + \frac{24}{(\alpha_2 - \alpha_1)^2} \int_{\alpha_1}^{\alpha_2} \int_s^{\alpha_2} z(u) du ds,$$

$$- \frac{60}{(\alpha_2 - \alpha_1)^3} \int_{\alpha_1}^{\alpha_2} \int_s^{\alpha_2} \int_u^{\alpha_2} z(v) dv du ds.$$

**Lemma 4** [36] *Let* $R \in \mathbb{R}^{n \times n}$ *is positive definite matrix, for any continuously differentiable function* $z : [\alpha_1, \alpha_2] \to \mathbb{R}^n$, *the following inequality holds*:

$$\int_{\alpha_1}^{\alpha_2} \int_s^{\alpha_2} \int_u^{\alpha_2} \dot{z}^T(v) R \dot{z}(v) dv du ds \geq \frac{6}{(\alpha_2 - \alpha_1)^3} X_1^T R X_1 + \frac{10}{(\alpha_2 - \alpha_1)^3} X_2^T R X_2$$

*where*

$$X_1 = \frac{(\alpha_2 - \alpha_1)^2}{2} z(\alpha_2) - \int_{\alpha_1}^{\alpha_2} \int_s^{\alpha_2} z(u) du ds,$$

$$X_2 = \frac{(\alpha_2 - \alpha_1)^2}{6} z(\alpha_2) - \int_{\alpha_1}^{\alpha_2} \int_s^{\alpha_2} z(u) du ds + \frac{4}{\alpha_2 - \alpha 1} \int_{\alpha_1}^{\alpha_2} \int_s^{\alpha_2} \int_u^{\alpha_2} z(v) dv du ds.$$

**Lemma 5** [37] *Give* $L = L^T$, $J$, $S$ *and* $Q(t)$ *agreeable* $Q^T(t)Q(t) \leq I$ *are matrices that suitable dimensions, hence the inequality as ensuing*:

$$L + JX(t)S + S^T Q^T(t) J^T < 0$$

*is real, if it's tantamount the following inequality holds for any* $\varepsilon > 0$,

$$L + \varepsilon^{-1} JJ^T + \varepsilon S^T S < 0.$$

**Lemma 6** [38] *(Jensen's Inequality) Let A is positive definite matrix, $g \in \mathbb{R}^+$ and $\dot{\omega}(t)$ :*
*$[-g, 0] \to \mathbb{R}^n$ is vector function hence the inequality as ensuing:*

$$-g \int_{-g}^{0} \dot{\omega}^T(s+t)A\dot{\omega}(s+t)ds + \left( \int_{-g}^{0} \dot{\omega}(s+t)ds \right)^T A \left( \int_{-g}^{0} \dot{\omega}(s+t)ds \right) \leq 0.$$

**Lemma 7** [39] *(Schur complement) For constant matrices $M_1$, $M_2$ and $M_3$ with suitable*
*dimensions, when $M_1 = M_1^T$ and $M_2 = M_2^T$, hence*

$$M_1 + M_3^T M_2^{-1} M_3 < 0$$

*if and only if*

$$\begin{bmatrix} M_1 & M_3^T \\ * & -M_2 \end{bmatrix} \quad or \quad \begin{bmatrix} -M_2 & M_3 \\ * & M_1 \end{bmatrix}.$$

## 3 Main results

**Theorem 1** *For given constants $\sigma_1$, $\sigma_2$, $\tau_1$, $\tau_2$, $h_1$, $h_2 \geq 0$ system (1) with certain terms is exponentially passive. If there are real positive definite matrices $L_1$, $Q_1$, $Q_2$, $Q_3$, $R_1$, $R_2$, $R_3$, $R_4$, $Z_1$, $Z_2$, $Z_3$, $W_1$, $W_2$, $W_3$, $W_4$, $U_1$, $U_2$, $U_3$, $U_4$, $S_1$, $S_2$, $S_3$, $S_4$ and a positive $\lambda$ agreeable the ensuing LMI holds for $k = 1, 2, \ldots, m$:*

$$\psi_{1k} < 0, \tag{2}$$

*where*

$$\psi_{1k} = \Sigma,$$

$$\Sigma = \Xi_{1k} + \Xi_2 + \Xi_3 + \Xi_4 + \Xi_5 + \Xi_6 + \Xi_7 + \Xi_8,$$

$$\begin{aligned} \Xi_{1k} =& [2e_1 L_1 A_k e_1 + 2e_1 L_1 B_k e_3 + 2e_1 L_1 C_k e_4 + 2e_1 L_1 D_k e_5 + e_1 L_1 E_k e_{30} + 2e_1 L_2 e_2 \\ & - 2e_1 L_2 A_k e_1 - 2e_1 L_2 B_k e_3 - 2e_1 L_2 C_k e_4 - 2e_1 L_2 D_k e_5 - 2e_1 L_2 E_k e_{30} \\ & - 2e_2 L_3 e_2 + 2e_2 L_3 A_k e_1 + 2e_2 L_3 B_k e_3 + e_2 C_k L_3 e_4 + 2e_2 D_k L_3 e_5 + 2e_2 L_3 E_k e_{30} \\ & + 2e_3 L_4 e_2 - 2e_3 L_4 A_k e_1 - 2e_3 L_4 B_k e_3 - e_3 C_k L_4 e_4 - 2e_3 D_k L_4 e_5 - 2e_3 L_4 E_k e_{30} \\ & + 2e_4 L_5 e_2 - 2e_4 L_5 A_k e_1 - 2e_4 L_5 B_k e_3 - 2e_4 L_5 C_k e_4 - 2e_4 L_5 D_k e_5 \\ & - 2e_4 L_5 E_k e_{30} + 2e_5 L_6 e_2 - 2e_5 L_5 A_k e_1 - 2e_5 L_6 B_k e_3 - 2e_5 L_6 C_k e_4 - 2e_5 L_6 D_k e_5 \\ & - 2e_5 L_6 E_k e_{30} + 2e_{30} L_7 e_2 - 2e_{30} L_7 A_k e_1 - 2e_{30} L_7 B_k e_3 - 2e_{30} L_7 C_k e_4 \\ & - 2e_{30} L_7 D_k e_5 - 2e_{30} L_7 E_k e_{24}], \end{aligned}$$

$$\Xi_2 = [e_1 Q_1 e_1 - e^{-2\alpha\tau_1} e_6 Q_1 e_6 + e_1 Q_2 e_1 - e^{-2\alpha\sigma_1} e_7 Q_2 e_7 + e_1 Q_3 e_1 - e^{-2\alpha h_1} e_8 Q_3 e_8$$

$$+ e_1 R_1 e_1 - e^{-2\alpha\tau_2} e_9 R_1 e_9 + \tau_d e^{-2\alpha\tau_1} e_9 R_1 e_9 + e_1 R_2 e_1 - e^{-2\alpha\sigma_2} e_3 R_2 e_3$$

$$+ \sigma_d e^{-2\alpha\sigma_1} e_3 R_2 e_3 + e_1 R_3 e_1 - e^{-2\alpha h_2} e_{10} R_3 e_{10} + h_d e^{-2\alpha h_1} e_{10} R_3 e_{10} + e_2 R_4 e_2$$

$$- e^{-2\alpha\tau_2} e_4 R_4 e_4 + \tau_d e^{-2\alpha\tau_1} e_4 R_4 e_4],$$

$$\Xi_3 \quad = \sigma_2^2 e_2 S_1 e_2 - e^{-2\alpha\sigma_2} [e_1 - e_{11}]^T S_1 [e_1 - e_{11}]$$

$$- 3 e^{-2\alpha\sigma_2} \left[ e_1 + e_{11} - \frac{2}{\sigma_2} e_{12} \right]^T S_1 \left[ e_1 + e_{11} - \frac{2}{\sigma_2} e_{12} \right]$$

$$- 5 e^{-2\alpha\sigma_2} \left[ e_1 - e_{11} + \frac{6}{\sigma_2} e_{12} - \frac{12}{\sigma_2^2} e_{13} \right]^T S_1 \left[ e_1 - e_{11} + \frac{6}{\sigma_2} e_{12} - \frac{12}{\sigma_2^2} e_{13} \right] - 7 e^{-2\sigma_2} \times$$

$$\left[ e_1 + e_{11} - \frac{12}{\sigma_2} e_{12} + \frac{60}{\sigma_2^2} e_{13} - \frac{120}{\sigma_2^3} e_{14} \right]^T S_1 \left[ e_1 + e_{11} - \frac{12}{\sigma_2} e_{12} + \frac{60}{\sigma_2^2} e_{13} - \frac{120}{\sigma_2^3} e_{14} \right]$$

$$+ \sigma_1^2 e_2 S_2 e_2 - e^{-2\alpha\sigma_1} [e_1 - e_7]^T S_2 [e_1 - e_7]$$

$$- 3 e^{-2\alpha\sigma_1} \left[ e_1 + e_7 - \frac{2}{\sigma_1} e_{15} \right]^T S_2 \left[ e_1 + e_7 - \frac{2}{\sigma_1} e_{15} \right]$$

$$- 5 e^{-2\alpha\sigma_1} \left[ e_1 - e_7 + \frac{6}{\sigma_1} e_{15} - \frac{12}{\sigma_1^2} e_{16} \right]^T S_2 \left[ e_1 - e_7 + \frac{6}{\sigma_1} e_{15} - \frac{12}{\sigma_1^2} e_{16} \right] - 7 e^{-2\sigma_1} \times$$

$$\left[ e_1 + e_7 - \frac{12}{\sigma_1} e_{15} + \frac{60}{\sigma_1^2} e_{16} - \frac{120}{\sigma_1^3} e_{17} \right]^T S_2 \left[ e_1 + e_7 - \frac{12}{\sigma_1} e_{15} + \frac{60}{\sigma_1^2} e_{16} - \frac{120}{\sigma_1^3} e_{17} \right]$$

$$+ \tau_2^2 e_2 S_3 e_2 - e^{-2\alpha\tau_2} [e_1 - e_{18}]^T S_3 [e_1 - e_{18}]$$

$$- 3 e^{-2\alpha\tau_2} \left[ e_1 + e_{18} - \frac{2}{\tau_2} e_{19} \right]^T S_3 \left[ e_1 + e_{18} - \frac{2}{\tau_2} e_{19} \right]$$

$$- 5 e^{-2\alpha\tau_2} \left[ e_1 - e_{18} + \frac{6}{\tau_2} e_{19} - \frac{12}{\tau_2^2} e_{20} \right]^T S_3 \left[ e_1 - e_{18} + \frac{6}{\tau_2} e_{19} - \frac{12}{\tau_2^2} e_{20} \right] - 7 e^{-2\tau_2} \times$$

$$\left[ e_1 + e_{18} - \frac{12}{\tau_2} e_{19} + \frac{60}{\tau_2^2} e_{20} - \frac{120}{\tau_2^3} e_{21} \right]^T S_3 \left[ e_1 + e_{18} - \frac{12}{\tau_2} e_{19} + \frac{60}{\tau_2^2} e_{20} - \frac{120}{\tau_2^3} e_{21} \right]$$

$$+ \tau_1^2 e_2 S_4 e_2 - e^{-2\alpha\tau_1} [e_1 - e_6]^T S_4 [e_1 - e_6]$$

$$- 3 e^{-2\alpha\tau_1} \left[ e_1 + e_6 - \frac{2}{\tau_1} e_{22} \right]^T S_4 \left[ e_1 + e_6 - \frac{2}{\tau_1} e_{22} \right]$$

$$- 5e^{-2\alpha\tau_1} \left[ e_1 - e_6 + \frac{6}{\tau_1} e_{22} - \frac{12}{\tau_1^2} e_{23} \right]^T S_4 \left[ e_1 - e_6 + \frac{6}{\tau_1} e_{22} - \frac{12}{\tau_1^2} e_{23} \right] - 7e^{-2\tau_1} \times$$

$$\left[ e_1 + e_6 - \frac{12}{\tau_1} e_{22} + \frac{60}{\tau_1^2} e_{23} - \frac{120}{\tau_1^3} e_{24} \right]^T S_4 \left[ e_1 + e_6 - \frac{12}{\tau_1} e_{22} + \frac{60}{\tau_1^2} e_{23} - \frac{120}{\tau_1^3} e_{24} \right],$$

$$\Xi_4 = [(\tau_2 - \tau_1)^2 e_2 Z_1 e_2 - e^{-2\alpha\tau_2} e_{25} Z_1 e_{25} - e^{-2\alpha\tau_2} e_{26} Z_1 e_{26} + (\sigma_2 - \sigma_1)^2 e_2 Z_2 e_2$$

$$- e^{-2\alpha\sigma_2} e_{27} Z_2 e_{27} - e^{-2\alpha\tau_2} e_{28} Z_2 e_{28} + h_2^2 e_2 Z_3 e_2 - e^{2\alpha h_2} e_{29} Z_3 e_{29}],$$

$$\Xi_5 = \sigma_2^2 e_2 W_1 e_2 - 2e^{-2\alpha\sigma_2} \left[ e_1 - \frac{1}{\sigma_2} e_{12} \right] W_1 \left[ e_1 - \frac{1}{\sigma_2} e_{12} \right]$$

$$- 4e^{-2\alpha\sigma_2} \left[ e_1 + \frac{2}{\sigma_2} e_{12} - \frac{6}{\sigma_2^2} e_{13} \right] W_1 \left[ e_1 + \frac{2}{\sigma_2} e_{12} - \frac{6}{\sigma_2^2} e_{13} \right]$$

$$- 6e^{-2\alpha\sigma_2} \left[ e_1 - \frac{3}{\sigma_2} e_{12} + \frac{24}{\sigma_2^2} e_{13} - \frac{60}{\sigma_2^3} e_{14} \right] W_1 \left[ e_1 - \frac{3}{\sigma_2} e_{12} + \frac{24}{\sigma_2^2} e_{13} - \frac{60}{\sigma_2^3} e_{14} \right]$$

$$+ \sigma_1^2 e_2 W_2 e_2 - 2e^{-2\alpha\sigma_1} \left[ e_1 - \frac{1}{\sigma_1} e_{15} \right] W_2 \left[ e_1 - \frac{1}{\sigma_1} e_{15} \right]$$

$$- 4e^{-2\alpha\sigma_1} \left[ e_1 + \frac{2}{\sigma_1} e_{15} - \frac{6}{\sigma_1^2} e_{16} \right] W_2 \left[ e_1 + \frac{2}{\sigma_1} e_{15} - \frac{6}{\sigma_1^2} e_{16} \right]$$

$$- 6e^{-2\alpha\sigma_1} \left[ e_1 - \frac{3}{\sigma_1} e_{15} + \frac{24}{\sigma_1^2} e_{16} - \frac{60}{\sigma_1^3} e_{17} \right] W_2 \left[ e_1 - \frac{3}{\sigma_1} e_{15} + \frac{24}{\sigma_1^2} e_{16} - \frac{60}{\sigma_1^3} e_{17} \right]$$

$$+ \tau_2^2 e_2 W_3 e_2 - 2e^{-2\alpha\tau_2} \left[ e_1 - \frac{1}{\tau_2} e_{19} \right] W_3 \left[ e_1 - \frac{1}{\tau_2} e_{19} \right]$$

$$- 4e^{-2\alpha\tau_2} \left[ e_1 + \frac{2}{\tau_2} e_{19} - \frac{6}{\tau_2^2} e_{20} \right] W_3 \left[ e_1 + \frac{2}{\tau_2} e_{19} - \frac{6}{\tau_2^2} e_{20} \right]$$

$$- 6e^{-2\alpha\tau_2} \left[ e_1 - \frac{3}{\tau_2} e_{19} + \frac{24}{\tau_2^2} e_{20} - \frac{60}{\tau_2^3} e_{21} \right] W_3 \left[ e_1 - \frac{3}{\tau_2} e_{19} + \frac{24}{\tau_2^2} e_{20} - \frac{60}{\tau_2^3} e_{21} \right]$$

$$+ \tau_1^2 e_2 W_4 e_2 - 2e^{-2\alpha\tau_1} \left[ e_1 - \frac{1}{\tau_1} e_{22} \right] W_4 \left[ e_1 - \frac{1}{\tau_1} e_{22} \right]$$

$$- 4e^{-2\alpha\tau_1} \left[ e_1 + \frac{2}{\tau_1} e_{22} - \frac{6}{\tau_1^2} e_{23} \right] W_4 \left[ e_1 + \frac{2}{\tau_1} e_{22} - \frac{6}{\tau_1^2} e_{23} \right]$$

$$- 6e^{-2\alpha\tau_1}\left[e_1 - \frac{3}{\tau_1}e_{22} + \frac{24}{\tau_1^2}e_{23} - \frac{60}{\tau_1^3}e_{24}\right]W_4\left[e_1 - \frac{3}{\tau_1}e_{22} + \frac{24}{\tau_1^2}e_{23} - \frac{60}{\tau_1^3}e_{24}\right],$$

$$\Xi_6 = \frac{\sigma_2^3}{6}e_2 U_1 e_2 - 6e^{-2\alpha\sigma_2}\left[\frac{\sigma_2^2}{2}e_1 - e_{13}\right]U_1\left[\frac{\sigma_2^2}{2}e_1 - e_{13}\right]$$

$$- 10e^{-2\alpha\sigma_2}\left[\frac{\sigma_2^2}{2}e_1 - e_{13} + \frac{4}{\sigma_2}e_{14}\right]U_1\left[\frac{\sigma_2^2}{2}e_1 - e_{13} + \frac{4}{\sigma_2}e_{14}\right]$$

$$+ \frac{\sigma_1^3}{6}e_2 U_2 e_2 - 6e^{-2\alpha\sigma_1}\left[\frac{\sigma_1^2}{2}e_1 - e_{16}\right]U_2\left[\frac{\sigma_1^2}{2}e_1 - e_{16}\right]$$

$$- 10e^{-2\alpha\sigma_1}\left[\frac{\sigma_1^2}{2}e_1 - e_{16} + \frac{4}{\sigma_1}e_{17}\right]U_2\left[\frac{\sigma_1^2}{2}e_1 - e_{16} + \frac{4}{\sigma_1}e_{17}\right]$$

$$+ \frac{\tau_2^3}{6}e_2 U_3 e_2 - 6e^{-2\alpha\tau_2}\left[\frac{\tau_2^2}{2}e_1 - e_{20}\right]U_3\left[\frac{\tau_2^2}{2}e_1 - e_{20}\right]$$

$$- 10e^{-2\alpha\tau_2}\left[\frac{\tau_2^2}{2}e_1 - e_{20} + \frac{4}{\tau_2}e_{21}\right]U_3\left[\frac{\tau_2^2}{2}e_1 - e_{20} + \frac{4}{\tau_2}e_{21}\right]$$

$$+ \frac{\tau_1^3}{6}e_2 U_4 e_2 - 6e^{-2\alpha\tau_1}\left[\frac{\tau_1^2}{2}e_1 - e_{23}\right]U_4\left[\frac{\tau_1^2}{2}e_1 - e_{23}\right]$$

$$- 10e^{-2\alpha\tau_1}\left[\frac{\tau_1^2}{2}e_1 - e_{23} + \frac{4}{\tau_1}e_{24}\right]U_4\left[\frac{\tau_1^2}{2}e_1 - e_{23} + \frac{4}{\tau_1}e_{24}\right]$$

$$\Xi_7 = [e_6 - e_9 - e_{25}]y_1[-e_6 + e_9 + e_{25}] + [e_9 - e_{18} - e_{26}]y_2[-e_9 + e_{18} + e_{26}]$$

$$+ [e_7 - e_3 - e_{27}]y_3[-e_7 + e_3 + e_{27}] + [e_3 - e_{11} - e_{28}]y_4[-e_3 + e_{11} + e_{28}]$$

$$+ [e_1 - e_{10} - e_{29}]y_5[e_1 + e_{10} + e_{29}],$$

$$\Xi_8 = -e_1\tilde{A}_i e_{30} - e_3\tilde{B}_i e_{30} - e_{24}\tilde{E}_i e_{30}$$

$$\xi(t) = [x(t), \dot{x}(t), x(t - \sigma(t)), \dot{x}(t - \tau(t)), \int_{t-h(t)}^t x(s)ds, x(t - \tau_1), x(t - \sigma_1), x(t - h_1),$$

$$x(t - \tau(t)), x(t - h(t)), x(t - \sigma_2), \int_{t-\sigma_2}^t x(s)ds, \int_{t-\sigma_2}^t \int_s^t x(u)duds,$$

$$\int_{t-\sigma_2}^t \int_s^t \int_u^t x(v)dvduds, \int_{t-\sigma_1}^t x(s)ds, \int_{t-\sigma_1}^t \int_s^t x(u)duds,$$

$$\int_{t-\sigma_1}^t \int_s^t \int_u^t x(v)dvduds, x(t - \tau_2), \int_{t-\tau_2}^t x(s)ds, \int_{t-\tau_2}^t \int_s^t x(u)duds,$$

$$\int_{t-\tau_2}^t \int_s^t \int_u^t x(v)dvduds, \int_{t-\tau_1}^t x(s)ds, \int_{t-\tau_1}^t \int_s^t x(u)duds,$$

$$\int_{t-\tau_1}^t \int_s^t \int_u^t x(v)dvduds, \int_{t-\tau(t)}^{t-\tau_1} \dot{x}(s)ds, \int_{t-\tau_2}^{t-\tau(t)} \dot{x}(s)ds, \int_{t-\sigma(t)}^{t-\sigma_1} \dot{x}(s)ds,$$

$$\int_{t-\sigma_2}^{t-\sigma(t)} \dot{x}(s)ds, \int_{t-h(t)}^{t} \dot{x}(s)ds, u(t)]^T.$$

*proof* This study focal point in the following Lyapunov-Krasovskii function of the system (1)

$$V(x(t)) = \sum_{i=1}^{6} V_i(x(t)),$$

where

$$V_1 = x^T(t)L_1 x(t),$$

$$V_2 = \int_{t-\tau_1}^{t} e^{2\alpha(s-t)}x^T(s)Q_1 x(s)ds + \int_{t-\sigma_1}^{t} e^{2\alpha(s-t)}x^T(s)Q_2 x(s)ds$$

$$+ \int_{t-h_1}^{t} e^{2\alpha(s-t)}x^T(s)Q_3 x(s)ds + \int_{t-\tau(t)}^{t} e^{2\alpha(s-t)}x^T(s)R_1 x(s)ds$$

$$+ \int_{t-\sigma(t)}^{t} e^{2\alpha(s-t)}\dot{x}^T(s)R_2 \dot{x}(s)ds + \int_{t-h(t)}^{t} e^{2\alpha(s-t)}\dot{x}^T(s)R_3 \dot{x}(s)ds$$

$$+ \int_{t-\tau(t)}^{t} e^{2\alpha(s-t)}\dot{x}^T(s)R_4 \dot{x}(s)ds,$$

$$V_3 = \sigma_2 \int_{-\sigma_2}^{0} \int_{t+\theta}^{t} e^{2\alpha(s-t)}\dot{x}^T(s)S_1 \dot{x}(s)dsd\theta + \sigma_1 \int_{-\sigma_1}^{0} \int_{t+\theta}^{t} e^{2\alpha(s-t)}\dot{x}^T(s)S_2 \dot{x}(s)dsd\theta$$

$$+ \tau_2 \int_{-\tau_2}^{0} \int_{t+\theta}^{t} e^{2\alpha(s-t)}\dot{x}^T(s)S_3 \dot{x}(s)dsd\theta + \tau_1 \int_{-\tau_1}^{0} \int_{t+\theta}^{t} e^{2\alpha(s-t)}\dot{x}^T(s)S_4 \dot{x}(s)dsd\theta,$$

$$V_4 = (\tau_2 - \tau_1) \int_{-\tau_2}^{-\tau_1} \int_{t+\theta}^{t} e^{2\alpha(s-t)}\dot{x}^T(s)Z_1 \dot{x}(s)dsd\theta$$

$$+ (\sigma_2 - \sigma_1) \int_{-\sigma_2}^{-\sigma_1} \int_{t+\theta}^{t} e^{2\alpha(s-t)}\dot{x}^T(s)Z_2 \dot{x}(s)dsd\theta$$

$$+ h(t) \int_{-h(t)}^{0} \int_{t+\theta}^{t} e^{2\alpha(s-t)}\dot{x}^T(s)Z_3 \dot{x}(s)dsd\theta,$$

$$V_5 = \sigma_2^2 \int_{-\sigma_2}^{0} \int_{\theta}^{0} \int_{t+\beta}^{t} e^{2\alpha(s-t)}\dot{x}^T(s)W_1 \dot{x}(s)dsd\beta d\theta$$

$$+ \sigma_1^2 \int_{-\sigma_1}^{0} \int_{\theta}^{0} \int_{t+\beta}^{t} e^{2\alpha(s-t)}\dot{x}^T(s)W_2 \dot{x}(s)dsd\beta d\theta$$

$$+ \tau_2^2 \int_{-\tau_2}^{0} \int_{\theta}^{0} \int_{t+\beta}^{t} e^{2\alpha(s-t)} \dot{x}^T(s) W_3 \dot{x}(s) ds d\beta d\theta$$

$$+ \tau_1^2 \int_{-\tau_1}^{0} \int_{\theta}^{0} \int_{t+\beta}^{t} e^{2\alpha(s-t)} \dot{x}^T(s) W_4 \dot{x}(s) ds d\beta d\theta,$$

$$V_6 = \tau_2^3 \int_{-\tau_2}^{0} \int_{\nu}^{0} \int_{\theta}^{0} \int_{t+\beta}^{t} e^{2\alpha(s-t)} \dot{x}^T(s) U_1 \dot{x}(s) ds d\beta d\theta d\nu$$

$$+ \tau_1^3 \int_{-\tau_1}^{0} \int_{\nu}^{0} \int_{\theta}^{0} \int_{t+\beta}^{t} e^{2\alpha(s-t)} \dot{x}^T(s) U_2 \dot{x}(s) ds d\beta d\theta d\nu$$

$$+ \sigma_2^3 \int_{-\sigma_2}^{0} \int_{\nu}^{0} \int_{\theta}^{0} \int_{t+\beta}^{t} e^{2\alpha(s-t)} \dot{x}^T(s) U_3 \dot{x}(s) ds d\beta d\theta d\nu$$

$$+ \sigma_1^3 \int_{-\sigma_1}^{0} \int_{\nu}^{0} \int_{\theta}^{0} \int_{t+\beta}^{t} e^{2\alpha(s-t)} \dot{x}^T(s) U_4 \dot{x}(s) ds d\beta d\theta d\nu.$$

Add derivative with $V(x(t))$ in accordance direction of result for system (1) is specific as:

$$\dot{V}(x(t)) = \sum_{i=1}^{6} \dot{V}_i(x(t)),$$

where

$$
\begin{aligned}
\dot{V}_1(x(t)) &= 2x^T L_1 x(t) \\
&= 2[x^T(t) L_1 (Ax(t) + Bx(t-\sigma(t)) + C\dot{x}(x-\tau(t)) + D \int_{t-h(t)}^{t} x(s)ds \\
&\quad + Eu(t))] \\
&\quad + 2[x^T(t) L_2 (\dot{x}(t) - Ax(t) - Bx(t-\sigma(t)) - C\dot{x}(x-\tau(t)) \\
&\quad - D \int_{t-h(t)}^{t} x(s)ds - Eu(t))] \\
&\quad + 2[\dot{x}^T(t) L_3 (-\dot{x}(t) + Ax(t) + Bx(t-\sigma(t)) + C\dot{x}(x-\tau(t)) \\
&\quad + D \int_{t-h(t)}^{t} x(s)ds + Eu(t))] \\
&\quad + 2[x^T(t-\sigma(t)) L_4 (\dot{x}(t) - Ax(t) - Bx(t-\sigma(t)) - C\dot{x}(x-\tau(t)) \\
&\quad - D \int_{t-h(t)}^{t} x(s)ds - Eu(t))] \\
&\quad + 2[\int_{t-h(t)}^{t} x(s)ds L_5 (\dot{x}(t) - Ax(t) - Bx(t-\sigma(t)) - C\dot{x}(x-\tau(t)) \\
&\quad - D \int_{t-h(t)}^{t} x(s)ds - Eu(t))] \\
&\quad + 2[\dot{x}^T(t-\tau(t)) L_6 (\dot{x}(t) - Ax(t) - Bx(t-\sigma(t)) - C\dot{x}(x-\tau(t)) \\
&\quad - D \int_{t-h(t)}^{t} x(s)ds - Eu(t))] + 2\alpha x^T(t) L_1 x(t) - 2\alpha V_1(x(t)) \\
&= \xi^T(t) \Xi_1 \xi(t) - 2\alpha V_1(x(t)).
\end{aligned}
$$

(3)

$$
\begin{aligned}
\dot{V}_2(x(t)) \;=\; & x^T(t)Q_1 x(t) - e^{-2\alpha\tau_1} x^T(t-\tau_1)Q_1 x(t-\tau_1) + x^T(t)Q_2 x(t) \\
& - e^{-2\alpha\sigma_1} x^T(t-\sigma_1)Q_2 x(t-\sigma_1) \\
& + x^T(t)Q_3 x(t) - e^{-2\alpha h_1} x^T(t-h_1)Q_3 x(t-h_1) \\
& + x^T(t)R_1 x(t) - (1-\dot{\tau}(t))e^{2\alpha(t-\tau(t))} x^T(t-\tau(t))R_1 x(t-\tau(t)) \\
& + \dot{x}^T(t)R_2 \dot{x}(t) - (1-\dot{\sigma}(t))e^{2\alpha(t-\sigma(t))} \dot{x}^T(t-\sigma(t))R_2 \dot{x}(t-\sigma(t)) \\
& + \dot{x}^T(t)R_3 \dot{x}(t) - (1-\dot{h}(t))e^{2\alpha(t-h(t))} \dot{x}^T(t-h(t))R_3 \dot{x}(t-h(t)) \\
& + \dot{x}^T(t)R_4 \dot{x}(t) - (1-\dot{\tau}(t))e^{2\alpha(t-\tau(t))} \dot{x}^T(t-\tau(t))R_4 \dot{x}(t-\tau(t)) \\
& - 2\alpha V_2(x(t)) \\
\leq\; & \xi^T(t)\Xi_2\xi(t) - 2\alpha V_2(x(t)).
\end{aligned}
\tag{4}
$$

$$
\begin{aligned}
\dot{V}_3(x(t)) \;=\; & \sigma_2 e^{-2\alpha t}\Big[\int_{-\sigma_2}^{0} e^{2\alpha t}\dot{x}^T(t)S_1\dot{x}(t)d\theta - \int_{-\sigma_2}^{0} e^{2\alpha(t+\theta)}\dot{x}^T(t+\theta)S_1\dot{x}(t+\theta)d\theta\Big] \\
& + \sigma_1 e^{-2\alpha t}\Big[\int_{-\sigma_1}^{0} e^{2\alpha t}\dot{x}^T(t)S_2\dot{x}(t)d\theta - \int_{-\sigma_1}^{0} e^{2\alpha(t+\theta)}\dot{x}^T(t+\theta)S_2\dot{x}(t+\theta)d\theta\Big] \\
& + \tau_2 e^{-2\alpha t}\Big[\int_{-\tau_2}^{0} e^{2\alpha t}\dot{x}^T(t)S_3\dot{x}(t)d\theta - \int_{-\tau_2}^{0} e^{2\alpha(t+\theta)}\dot{x}^T(t+\theta)S_3\dot{x}(t+\theta)d\theta\Big] \\
& + \tau_1 e^{-2\alpha t}\Big[\int_{-\tau_1}^{0} e^{2\alpha t}\dot{x}^T(t)S_4\dot{x}(t)d\theta - \int_{-\tau_1}^{0} e^{2\alpha(t+\theta)}\dot{x}^T(t+\theta)S_4\dot{x}(t+\theta)d\theta\Big] \\
& - 2\alpha V_3(x(t)) \\
\leq\; & \sigma_2^2 \dot{x}^T S_1\dot{x}(t) - \sigma_2 \int_{t-\sigma_2}^{t} \dot{x}^T(s)S_1\dot{x}(s)ds + \sigma_1^2 \dot{x}^T S_2\dot{x}(t) - \sigma_1 \int_{t-\sigma_1}^{t} \dot{x}^T(s)S_2\dot{x}(s)ds \\
& + \tau_2^2 \dot{x}^T S_3\dot{x}(t) - \tau_2 \int_{t-\tau_2}^{t} \dot{x}^T(s)S_3\dot{x}(s)ds + \tau_1^2 \dot{x}^T S_4\dot{x}(t) - \tau_1 \int_{t-\tau_1}^{t} \dot{x}^T(s)S_4\dot{x}(s)ds \\
& - 2\alpha V_3(x(t)).
\end{aligned}
$$

Lemma 2 is used to obtain

$$
\begin{aligned}
\dot{V}_3(x(t)) \;\leq\; & \sigma_2^2 \dot{x}^T S_1\dot{x}(t) - e^{-2\alpha\sigma_2}\big[x(t) - x(t-\sigma_2)\big]^T S_1\big[x(t) - x(t-\sigma_2)\big] \\
& - 3e^{-2\alpha\sigma_2}\Big[x(t) + x(t-\sigma_2) - \frac{2}{\sigma_2}\int_{t-\sigma_2}^{t} x(s)ds\Big]^T \\
& \times S_1\Big[x(t) + x(t-\sigma_2) - \frac{2}{\sigma_2}\int_{t-\sigma_2}^{t} x(s)ds\Big] \\
& - 5e^{-2\alpha\sigma_2}\Big[x(t) - x(t-\sigma_2) + \frac{6}{\sigma_2}\int_{t-\sigma_2}^{t} x(s)ds - \frac{12}{\sigma_2^2}\int_{t-\sigma_2}^{t}\int_{s}^{t} x(u)du\,ds\Big]^T \\
& \times S_1\Big[x(t) - x(t-\sigma_2) + \frac{6}{\sigma_2}\int_{t-\sigma_2}^{t} x(s)ds - \frac{12}{(\sigma_2)^2}\int_{t-\sigma_2}^{t}\int_{s}^{t} x(u)du\,ds\Big]
\end{aligned}
$$

$$-7e^{-2\alpha\sigma_2}\left[x(t)+x(t-\sigma_2)-\frac{12}{\sigma_2}\int_{t-\sigma_2}^t x(s)ds+\frac{60}{\sigma_2^2}\int_{t-\sigma_2}^t\int_s^t x(u)duds\right.$$

$$\left.-\frac{120}{\sigma_2^3}\int_{t-\sigma_2}^t\int_s^t\int_u^t x(v)dvduds\right]^T S_1\left[x(t)+x(t-\sigma_2)-\frac{12}{\sigma_2}\int_{t-\sigma_2}^t x(s)ds\right.$$

$$\left.+\frac{60}{\sigma_2^2}\int_{t-\sigma_2}^t\int_s^t x(u)duds-\frac{120}{\sigma_2^3}\int_{t-\sigma_2}^t\int_s^t\int_u^t x(v)dvduds\right]$$

$$+\sigma_1^2\dot{x}^T S_2\dot{x}(t)-e^{-2\alpha\sigma_1}[x(t)-x(t-\sigma_1)]^T S_2[x(t)-x(t-\sigma_1)]$$

$$-3e^{-2\alpha\sigma_1}\left[x(t)+x(t-\sigma_1)-\frac{2}{\sigma_1}\int_{t-\sigma_1}^t x(s)ds\right]^T S_2$$

$$\times\left[x(t)+x(t-\sigma_1)-\frac{2}{\sigma_1}\int_{t-\sigma_1}^t x(s)ds\right]$$

$$-5e^{-2\alpha\sigma_1}\left[x(t)-x(t-\sigma_1)+\frac{6}{\sigma_1}\int_{t-\sigma_1}^t x(s)ds-\frac{12}{\sigma_1^2}\int_{t-\sigma_1}^t\int_s^t x(u)duds\right]^T$$

$$\times S_2\left[x(t)-x(t-\sigma_1)+\frac{6}{\sigma_1}\int_{t-\sigma_1}^t x(s)ds-\frac{12}{(\sigma_1)^2}\int_{t-\sigma_1}^t\int_s^t x(u)duds\right]$$

$$-7e^{-2\alpha\sigma_1}\left[x(t)+x(t-\sigma_1)-\frac{12}{\sigma_1}\int_{t-\sigma_1}^t x(s)ds+\frac{60}{\sigma_1^2}\int_{t-\sigma_1}^t\int_s^t x(u)duds\right.$$

$$\left.-\frac{120}{\sigma_1^3}\int_{t-\sigma_1}^t\int_s^t\int_u^t x(v)dvduds\right]^T S_2\left[x(t)+x(t-\sigma_1)-\frac{12}{\sigma_1}\int_{t-\sigma_1}^t x(s)ds\right.$$

$$\left.+\frac{60}{\sigma_1^2}\int_{t-\sigma_1}^t\int_s^t x(u)duds-\frac{120}{\sigma_1^3}\int_{t-\sigma_1}^t\int_s^t\int_u^t x(v)dvduds\right]$$

$$-5e^{-2\alpha\tau_2}\left[x(t)-x(t-\tau_2)+\frac{6}{\tau_2}\int_{t-\tau_2}^t x(s)ds-\frac{12}{\tau_2^2}\int_{t-\tau_2}^t\int_s^t x(u)duds\right]^T$$

$$\times S_3\left[x(t)-x(t-\tau_2)+\frac{6}{\tau_2}\int_{t-\tau_2}^t x(s)ds-\frac{12}{(\tau_2)^2}\int_{t-\tau_2}^t\int_s^t x(u)duds\right]$$

$$-7e^{-2\alpha\tau_2}\left[x(t)+x(t-\tau_2)-\frac{12}{\tau_2}\int_{t-\tau_2}^t x(s)ds+\frac{60}{\tau_2^2}\int_{t-\tau_2}^t\int_s^t x(u)duds\right.$$

$$-\frac{120}{\tau_2^3}\int_{t-\tau_2}^{t}\int_{s}^{t}\int_{u}^{t}x(v)dvduds]^{T}S_3\left[x(t)+x(t-\tau_2)-\frac{12}{\tau_2}\int_{t-\tau_2}^{t}x(s)ds\right.$$

$$+\frac{60}{\tau_2^2}\int_{t-\tau_2}^{t}\int_{s}^{t}x(u)duds-\frac{120}{\tau_2^3}\int_{t-\tau_2}^{t}\int_{s}^{t}\int_{u}^{t}x(v)dvduds]$$

$$+\tau_1^2\dot{x}^{T}S_4\dot{x}(t)-e^{-2\alpha\tau_1}[x(t)-x(t-\tau_1)]^{T}S_4[x(t)-x(t-\tau_1)]$$

$$-3e^{-2\alpha\tau_1}\left[x(t)+x(t-\tau_1)-\frac{2}{\tau_1}\int_{t-\tau_1}^{t}x(s)ds\right]^{T}S_4$$

$$\times\left[x(t)+x(t-\tau_1)-\frac{2}{\tau_1}\int_{t-\tau_1}^{t}x(s)ds\right]$$

$$-5e^{-2\alpha\tau_1}\left[x(t)-x(t-\tau_1)+\frac{6}{\tau_1}\int_{t-\tau_1}^{t}x(s)ds-\frac{12}{\tau_1^2}\int_{t-\tau_1}^{t}\int_{s}^{t}x(u)duds\right]^{T}$$

$$\times S_4\left[x(t)-x(t-\tau_1)+\frac{6}{\tau_1}\int_{t-\tau_1}^{t}x(s)ds-\frac{12}{(\tau_1)^2}\int_{t-\tau_1}^{t}\int_{s}^{t}x(u)duds\right]$$

$$-7e^{-2\alpha\tau_1}\left[x(t)+x(t-\tau_1)-\frac{12}{\tau_1}\int_{t-\tau_1}^{t}x(s)ds+\frac{60}{\tau_1^2}\int_{t-\tau_1}^{t}\int_{s}^{t}x(u)duds\right.$$

$$-\frac{120}{\tau_1^3}\int_{t-\tau_1}^{t}\int_{s}^{t}\int_{u}^{t}x(v)dvduds]^{T}S_4\left[x(t)+x(t-\tau_1)-\frac{12}{\tau_1}\int_{t-\tau_1}^{t}x(s)ds\right.$$

$$+\frac{60}{\tau_1^2}\int_{t-\tau_1}^{t}\int_{s}^{t}x(u)duds-\frac{120}{\tau_1^3}\int_{t-\tau_1}^{t}\int_{s}^{t}\int_{u}^{t}x(v)dvduds]-2\alpha V_3(x(t))$$

$$=\xi^{T}(t)\Xi_3\xi(t)-2\alpha V_3(x(t)). \tag{5}$$

$$\begin{aligned}\dot{V}_4(x(t))\quad&=(\tau_2-\tau_1)e^{-2\alpha t}[\int_{-\tau_2}^{-\tau_1}e^{2\alpha t}\dot{x}^{T}(t)Z_1\dot{x}(t)d\theta\\&-\int_{-\tau_2}^{-\tau_1}e^{2\alpha(t+\theta)}\dot{x}^{T}(t+\theta)Z_1\dot{x}(t+\theta)d\theta]\\&+(\sigma_2-\sigma_1)e^{-2\alpha T}[\int_{-\tau_2}^{-\tau_1}e^{2\alpha t}\dot{x}^{T}(t)Z_2\dot{x}(t)d\theta\\&-\int_{-\sigma_2}^{-\sigma_1}e^{2\alpha(t+\theta)}\dot{x}^{T}(t+\theta)Z_2\dot{x}(t+\theta)d\theta]\\&+h(t)e^{-2\alpha t}[\int_{-h(t)}^{0}e^{2\alpha t}x^{T}(t)Z_3x(t)d\theta\\&-\int_{-h(t)}^{0}e^{2\alpha(t+\theta)}x^{T}(t+\theta)Z_1x(t+\theta)d\theta]\\&-2\alpha V_4(x(t)).\end{aligned}$$

Lemma 1 and Lemma 6, are used to obtain

$$
\begin{aligned}
\dot{V}_4(x(t)) \quad &\le (\tau_2 - \tau_1)^2 \dot{x}^T(t) Z_1 \dot{x}(t) - e^{-2\alpha\tau_2} \Big( \int_{t-\tau(t)}^{t-\tau_1} \dot{x}^T(s) ds Z_1 \int_{t-\tau(t)}^{t-\tau_1} \dot{x}^T(s) ds \\
&\quad - \int_{t-\tau_2}^{t-\tau(t)} \dot{x}^T(s) ds Z_1 \int_{t-\tau_2}^{t-\tau(t)} \dot{x}^T(s) ds \Big) \\
&\quad + (\sigma_2 - \sigma_1)^2 \dot{x}^T(t) Z_2 \dot{x}(t) - e^{-2\alpha\sigma_2} \Big( \int_{t-\sigma(t)}^{t-\sigma_1} \dot{x}^T(s) ds Z_2 \int_{t-\sigma(t)}^{t-\sigma_1} \dot{x}^T(s) ds \\
&\quad + \int_{t-\sigma_2}^{t-\sigma(t)} \dot{x}^T(s) ds Z_2 \int_{t-\sigma_2}^{t-\sigma(t)} \dot{x}^T(s) ds \Big) \\
&\quad + h_2^2 \dot{x}^T(t) Z_3 \dot{x}(t) - e^{-2\alpha h_2} \int_{t-h(t)}^{t} \dot{x}(s) ds Z_3 \int_{t-h(t)}^{t} \dot{x}(s) ds - 2\alpha V_4(x(t)) \\
&= \xi^T(t) \Xi_4 \xi(t) - 2\alpha V_4(x(t)).
\end{aligned}
\tag{6}
$$

$$
\begin{aligned}
\dot{V}_5(x(t)) \quad &= \sigma_2^2 e^{-2\alpha t} \Big[ \int_{-\sigma_2}^{0} \int_{\theta}^{0} e^{2\alpha t} \dot{x}^T(t) W_1 \dot{x}(t) d\beta d\theta \\
&\quad - \int_{-\sigma_2}^{0} \int_{\theta}^{0} e^{2\alpha(t+\beta)} \dot{x}^T(t+\beta) W_1 \dot{x}(t+\beta) d\beta d\theta \Big] \\
&\quad + \sigma_1^2 e^{-2\alpha t} \Big[ \int_{-\sigma_1}^{0} \int_{\theta}^{0} e^{2\alpha t} \dot{x}^T(t) W_2 \dot{x}(t) d\beta d\theta \\
&\quad - \int_{-\sigma_1}^{0} \int_{\theta}^{0} e^{2\alpha(t+\beta)} \dot{x}^T(t+\beta) W_2 \dot{x}(t+\beta) d\beta d\theta \Big] \\
&\quad + \tau_2^2 e^{-2\alpha t} \Big[ \int_{-\tau_2}^{0} \int_{\theta}^{0} e^{2\alpha t} \dot{x}^T(t) W_2 \dot{x}(t) d\beta d\theta \\
&\quad - \int_{-\tau_2}^{0} \int_{\theta}^{0} e^{2\alpha(t+\beta)} \dot{x}^T(t+\beta) W_2 \dot{x}(t+\beta) d\beta d\theta \Big] \\
&\quad + \tau_1^2 e^{-2\alpha t} \Big[ \int_{-\tau_1}^{0} \int_{\theta}^{0} e^{2\alpha t} \dot{x}^T(t) W_4 \dot{x}(t) d\beta d\theta \\
&\quad - \int_{-\tau_1}^{0} \int_{\theta}^{0} e^{2\alpha(t+\beta)} \dot{x}^T(t+\beta) W_4 \dot{x}(t+\beta) d\beta d\theta \Big] - 2\alpha V_5(x(t)).
\end{aligned}
$$

Lemma 3 is used to obtain

$$
\dot{V}_5(x(t)) \quad \le \frac{\sigma_2^2}{2} \dot{x}^T(t) W_1 \dot{x}(t)
$$

$$
- 2e^{-2\alpha\sigma_2} \left[ x(t) - \frac{1}{\sigma_2} \int_{t-\sigma_2}^{t} x(s) ds \right]^T W_1 \left[ x(t) - \frac{1}{\sigma_2} \int_{t-\sigma_2}^{t} x(s) ds \right]
$$

$$
- 4e^{-2\alpha\sigma_2} \left[ x(t) + \frac{2}{\sigma_2} \int_{t-\sigma_2}^{t} x(s) ds - \frac{6}{\sigma_2^2} \int_{t-\sigma_2}^{t} \int_{s}^{t} x(u) du ds \right]^T
$$

$$\times W_1 \left[ x(t) + \frac{2}{\sigma_2} \int_{t-\sigma_2}^t x(s)ds - \frac{6}{\sigma_2^2} \int_{t-\sigma_2}^t \int_s^t x(u)duds \right]$$

$$- 6e^{-2\alpha\sigma_2} \left[ x(t) - \frac{3}{\sigma_2} \int_{t-\sigma_2}^t x(s)ds + \frac{24}{\sigma_2^2} \int_{t-\sigma_2}^t \int_s^t x(u)duds \right.$$

$$- \frac{60}{\sigma_2^3} \int_{t-\sigma_2}^t \int_s^t \int_u^t x(v)dvduds \right]^T W_1 \left[ x(t) - \frac{3}{\sigma_2} \int_{t-\sigma_2}^t x(s)ds \right.$$

$$+ \frac{24}{\sigma_2^2} \int_{t-\sigma_2}^t \int_s^t x(u)duds - \frac{60}{\sigma_2^3} \int_{t-\sigma_2}^t \int_s^t \int_u^t x(v)dvduds \right]$$

$$+ \frac{\sigma_1^2}{2} \dot{x}^T(t) W_2 \dot{x}(t)$$

$$- 2e^{-2\alpha\sigma_1} \left[ x(t) - \frac{1}{\sigma_1} \int_{t-\sigma_1}^t x(s)ds \right]^T W_2 \left[ x(t) - \frac{1}{\sigma_1} \int_{t-\sigma_1}^t x(s)ds \right]$$

$$- 4e^{-2\alpha\sigma_1} \left[ x(t) + \frac{2}{\sigma_1} \int_{t-\sigma_1}^t x(s)ds - \frac{6}{\sigma_1^2} \int_{t-\sigma_1}^t \int_s^t x(u)duds \right]^T$$

$$\times W_2 \left[ x(t) + \frac{2}{\sigma_1} \int_{t-\sigma_1}^t x(s)ds - \frac{6}{\sigma_1^2} \int_{t-\sigma_1}^t \int_s^t x(u)duds \right]$$

$$- 6e^{-2\alpha\sigma_1} \left[ x(t) - \frac{3}{\sigma_1} \int_{t-\sigma_1}^t x(s)ds + \frac{24}{\sigma_1^2} \int_{t-\sigma_1}^t \int_s^t x(u)duds \right.$$

$$- \frac{60}{\sigma_1^3} \int_{t-\sigma_1}^t \int_s^t \int_u^t x(v)dvduds \right]^T W_2 \left[ x(t) - \frac{3}{\sigma_1} \int_{t-\sigma_1}^t x(s)ds \right.$$

$$+ \frac{24}{\sigma_1^2} \int_{t-\sigma_1}^t \int_s^t x(u)duds - \frac{60}{\sigma_1^3} \int_{t-\sigma_1}^t \int_s^t \int_u^t x(v)dvduds \right]$$

$$+ \frac{\tau_2^2}{2} \dot{x}^T(t) W_3 \dot{x}(t)$$

$$- 2e^{-2\alpha\tau_2} \left[ x(t) - \frac{1}{\tau_2} \int_{t-\tau_2}^t x(s)ds \right]^T W_3 \left[ x(t) - \frac{1}{\tau_2} \int_{t-\tau_2}^t x(s)ds \right]$$

$$- 4e^{-2\alpha\tau_2} \left[ x(t) + \frac{2}{\tau_2} \int_{t-\tau_2}^t x(s)ds - \frac{6}{\tau_2^2} \int_{t-\tau_2}^t \int_s^t x(u)duds \right]^T$$

$$\times W_3\left[x(t)+\frac{2}{\tau_2}\int_{t-\tau_2}^t x(s)ds-\frac{6}{\tau_2^2}\int_{t-\tau_2}^t\int_s^t x(u)duds\right]$$

$$-6e^{-2\alpha\tau_2}\left[x(t)-\frac{3}{\tau_2}\int_{t-\tau_2}^t x(s)ds+\frac{24}{\tau_2^2}\int_{t-\tau_2}^t\int_s^t x(u)duds\right.$$

$$\left.-\frac{60}{\tau_2^3}\int_{t-\tau_2}^t\int_s^t\int_u^t x(v)dvduds\right]^T W_3\left[x(t)-\frac{3}{\tau_2}\int_{t-\tau_2}^t x(s)ds\right.$$

$$+\frac{24}{\tau_2^2}\int_{t-\tau_2}^t\int_s^t x(u)duds-\frac{60}{\tau_2^3}\int_{t-\tau_2}^t\int_s^t\int_u^t x(v)dvduds\right]$$

$$+\frac{\tau_1^2}{2}\dot{x}^T(t)W_4\dot{x}(t)$$

$$-2e^{-2\alpha\tau_1}\left[x(t)-\frac{1}{\tau_1}\int_{t-\tau_1}^t x(s)ds\right]^T W_4\left[x(t)-\frac{1}{\tau_1}\int_{t-\tau_1}^t x(s)ds\right]$$

$$-4e^{-2\alpha\tau_1}\left[x(t)+\frac{2}{\tau_1}\int_{t-\tau_1}^t x(s)ds-\frac{6}{\tau_1^2}\int_{t-\tau_1}^t\int_s^t x(u)duds\right]^T$$

$$\times W_4\left[x(t)+\frac{2}{\tau_1}\int_{t-\tau_1}^t x(s)ds-\frac{6}{\tau_1^2}\int_{t-\tau_1}^t\int_s^t x(u)duds\right]$$

$$-6e^{-2\alpha\tau_1}\left[x(t)-\frac{3}{\tau_1}\int_{t-\tau_1}^t x(s)ds+\frac{24}{\tau_1^2}\int_{t-\tau_1}^t\int_s^t x(u)duds\right.$$

$$\left.-\frac{60}{\tau_1^3}\int_{t-\tau_1}^t\int_s^t\int_u^t x(v)dvduds\right]^T W_4\left[x(t)-\frac{3}{\tau_1}\int_{t-\tau_1}^t x(s)ds\right.$$

$$+\frac{24}{\tau_1^2}\int_{t-\tau_1}^t\int_s^t x(u)duds-\frac{60}{\tau_1^3}\int_{t-\tau_1}^t\int_s^t\int_u^t x(v)dvduds\right]-2\alpha V_5(x(t))$$

$$=\xi^T(t)\Xi_5\xi(t)-2\alpha V_5(x(t)). \tag{7}$$

$$\dot{V}_6(x(t))\quad =\sigma_2^3 e^{-2\alpha t}\left[\int_{-\sigma_2}^0\int_v^0\int_\theta^0 e^{2\alpha t}\dot{x}^T(t)U_1\dot{x}(t)d\beta d\theta dv\right.$$

$$\left.-\int_{-\sigma_2}^0\int_v^0\int_\theta^0 e^{2\alpha(t+\beta)}\dot{x}^T(t+\beta)U_1\dot{x}(t+\beta)d\beta d\theta dv\right]$$

$$+\sigma_1^3 e^{-2\alpha t}\left[\int_{-\sigma_1}^0\int_v^0\int_\theta^0 e^{2\alpha t}\dot{x}^T(t)U_2\dot{x}(t)d\beta d\theta dv\right.$$

$$\left.-\int_{-\sigma_1}^0\int_v^0\int_\theta^0 e^{2\alpha(t+\beta)}\dot{x}^T(t+\beta)U_2\dot{x}(t+\beta)d\beta d\theta dv\right]$$

$$+\tau_2^3 e^{-2\alpha t}\left[\int_{-\tau_2}^0\int_v^0\int_\theta^0 e^{2\alpha t}\dot{x}^T(t)U_3\dot{x}(t)d\beta d\theta dv\right.$$

$$\left.-\int_{-\tau_2}^0\int_v^0\int_\theta^0 e^{2\alpha(t+\beta)}\dot{x}^T(t+\beta)U_3\dot{x}(t+\beta)d\beta d\theta dv\right]$$

$$+\tau_1^3 e^{-2\alpha t}\left[\int_{-\tau_1}^0\int_v^0\int_\theta^0 e^{2\alpha t}\dot{x}^T(t)U_4\dot{x}(t)d\beta d\theta dv\right.$$

$$\left.-\int_{-\tau_1}^0\int_v^0\int_\theta^0 e^{2\alpha(t+\beta)}\dot{x}^T(t+\beta)U_4\dot{x}(t+\beta)d\beta d\theta dv\right]-2\alpha V_6(x(t)).$$

Lemma 4 is used to obtain

$$
\dot{V}_6(x(t)) \leq \frac{\sigma_2^6}{6}\dot{x}^T U_1 \dot{x}(t) - 6e^{-2\alpha\sigma_2}\left[\frac{\sigma_2^2}{2}x(t) - \int_{t-\sigma_2}^{t}\int_{s}^{t}x(u)duds\right]^T
$$

$$
\times U_1\left[\frac{\sigma_2^2}{2}x(t) - \int_{t-\sigma_2}^{t}\int_{s}^{t}x(u)duds\right]
$$

$$
- 10e^{-2\alpha\sigma_2}\left[\frac{\sigma_2^2}{2}x(t) - \int_{t-\sigma_2}^{t}\int_{s}^{t}x(u)duds + \frac{4}{\sigma_2}\int_{t-\sigma_2}^{t}\int_{s}^{t}\int_{u}^{t}x(v)dvduds\right]^T
$$

$$
\times U_1\left[\frac{\sigma_2^2}{2}x(t) - \int_{t-\sigma_2}^{t}\int_{s}^{t}x(u)duds + \frac{4}{\sigma_2}\int_{t-\sigma_2}^{t}\int_{s}^{t}\int_{u}^{t}x(v)dvduds\right]
$$

$$
+ \frac{\sigma_1^6}{6}\dot{x}^T U_2 \dot{x}(t) - 6e^{-2\alpha\sigma_1}\left[\frac{\sigma_1^2}{2}x(t) - \int_{t-\sigma_1}^{t}\int_{s}^{t}x(u)duds\right]^T
$$

$$
\times U_2\left[\frac{\sigma_1^2}{2}x(t) - \int_{t-\sigma_1}^{t}\int_{s}^{t}x(u)duds\right]
$$

$$
- 10e^{-2\alpha\sigma_1}\left[\frac{\sigma_1^2}{2}x(t) - \int_{t-\sigma_1}^{t}\int_{s}^{t}x(u)duds + \frac{4}{\sigma_1}\int_{t-\sigma_1}^{t}\int_{s}^{t}\int_{u}^{t}x(v)dvduds\right]^T
$$

$$
\times U_2\left[\frac{\sigma_1^2}{2}x(t) - \int_{t-\sigma_1}^{t}\int_{s}^{t}x(u)duds + \frac{4}{\sigma_1}\int_{t-\sigma_1}^{t}\int_{s}^{t}\int_{u}^{t}x(v)dvduds\right]
$$

$$
+ \frac{\tau_2^6}{6}\dot{x}^T U_3 \dot{x}(t) - 6e^{-2\alpha\tau_2}\left[\frac{\tau_2^2}{2}x(t) - \int_{t-\tau_2}^{t}\int_{s}^{t}x(u)duds\right]^T
$$

$$
\times U_3\left[\frac{\tau_2^2}{2}x(t) - \int_{t-\tau_2}^{t}\int_{s}^{t}x(u)duds\right]
$$

$$
- 10e^{-2\alpha\tau_2}\left[\frac{\tau_2^2}{2}x(t) - \int_{t-\tau_2}^{t}\int_{s}^{t}x(u)duds + \frac{4}{\tau_2}\int_{t-\tau_2}^{t}\int_{s}^{t}\int_{u}^{t}x(v)dvduds\right]^T
$$

$$
\times U_3\left[\frac{\tau_2^2}{2}x(t) - \int_{t-\tau_2}^{t}\int_{s}^{t}x(u)duds + \frac{4}{\tau_2}\int_{t-\tau_2}^{t}\int_{s}^{t}\int_{u}^{t}x(v)dvduds\right]
$$

$$
+ \frac{\tau_1^6}{6}\dot{x}^T U_4 \dot{x}(t) - 6e^{-2\alpha\tau_1}\left[\frac{\tau_1^2}{2}x(t) - \int_{t-\tau_1}^{t}\int_{s}^{t}x(u)duds\right]^T
$$

$$
\times U_4\left[\frac{\tau_1^2}{2}x(t) - \int_{t-\tau_1}^{t}\int_{s}^{t}x(u)duds\right]
$$

$$
- 10e^{-2\alpha\tau_1}\left[\frac{\tau_1^2}{2}x(t) - \int_{t-\tau_1}^{t}\int_{s}^{t}x(u)duds + \frac{4}{\tau_1}\int_{t-\tau_1}^{t}\int_{s}^{t}\int_{u}^{t}x(v)dvduds\right]^T
$$

$$
\times U_4\left[\frac{\tau_1^2}{2}x(t) - \int_{t-\tau_1}^{t}\int_{s}^{t}x(u)duds + \frac{4}{\tau_1}\int_{t-\tau_1}^{t}\int_{s}^{t}\int_{u}^{t}x(v)dvduds\right] - 2\alpha V_6(x(t))
$$

$$
= \xi^T(t)\Xi_6\xi(t) - 2\alpha V_6(x(t)). \tag{8}
$$

From the Newton-Leibniz formula, it can be expressed as

$$
\begin{aligned}
[x^T(t-\tau_1)- \quad & x^T(t-\tau(t)) - \int_{t-\tau(t)}^{t-\tau_1} \dot{x}^T(s)ds] \times \\
& y_1[-x(t-\tau_1) + x(t-\tau(t)) + \int_{t-\tau(t)}^{t-\tau_1} \dot{x}(s)ds] = 0, \\
[x(t-\tau(t))- \quad & x(t-\tau_2) - \int_{t-\tau_2}^{t-\tau(t)}] \times \\
& y_2[-x(t-\tau(t)) + x(t-\tau_2) + \int_{t-\tau_2}^{t-\tau(t)}] = 0, \\
[x^T(t-\sigma_1)- \quad & x^T(t-\sigma(t)) - \int_{t-\sigma(t)}^{t-\sigma_1} \dot{x}^T(s)ds] \times \\
& y_3[-x(t-\sigma_1) + x(t-\sigma(t)) + \int_{t-\sigma(t)}^{t-\sigma_1} \dot{x}(s)ds] = 0, \\
[x^T(t-\sigma(t))- \quad & x^T(t-\sigma_2) - \int_{t-\sigma_2}^{t-\sigma(t)} \dot{x}^T(s)ds] \times \\
& y_4[-x(t-\sigma(t)) + x(t-\sigma_2) + \int_{t-\sigma_2}^{t-\sigma(t)} \dot{x}(s)ds] = 0, \\
[x(t) \quad & -x(t-h(t)) - \int_{t-h(t)}^{t} \dot{x}(s)ds] \times \\
& y_5[-x(t) + x(t-h(t)) + \int_{t-h(t)}^{t} \dot{x}(s)ds] = 0.
\end{aligned}
\tag{9}
$$

Combining Eqs (3)–(9), it can be expressed as

$$
\dot{V}(x(t)) \quad \leq \quad \xi^T(t)\Sigma\xi(t) < 0.
$$

In addition, we possess $0 \leq \sum\limits_{i=1}^{r} \rho_i(\theta(t))$ hence

$$
\dot{V}(x(t)) \quad \leq \quad \sum_{i=1}^{r}\rho_i(\theta(t))\xi^T(t)\Sigma\xi(t) < 0.
$$

It is can be concluded the following inequality by (3)–(9) and $z(t)$

$$
\dot{V}(x(t)) + 2\alpha V(x(t)) - 2z(t)u(t) \leq \xi^T(t)\Sigma\xi(t).
$$

Therefore, the system (1) is guaranteed to be exponentially passive from Definition 1. The proof is completed.

Based on Theorem 1, we can perform the robust stability analysis for system (1) with uncertainty.

**Theorem 2** *For scalars $\sigma_1$, $\sigma_2$, $\tau_1$, $\tau_2$, $h_1$, $h_2 \geq 0$ system (1) with uncertain terms is exponentially passive. If there are matrices $L_1, Q_1, Q_2, Q_3, R_1, R_2, R_3, R_4, Z_1, Z_2, Z_3, W_1, W_2, U_1, U_2 > 0$ and a positive $\lambda$ satisfying the ensuing LMI holds*:

$$
\Omega_1 = \begin{bmatrix} \psi_{1k} & \Phi_1 \\ * & -\lambda_1 I \end{bmatrix} < 0,
$$

*where*

$$
\begin{aligned}
\psi_{1k} \quad &= \psi_{1k} + \Xi_9, \\
\Xi_9 \quad &= [\lambda e_1 H_{1i}^T H_{1i} e_1 + \lambda e_1 H_{1i}^T H_{2i} e_3 + \lambda e_1 H_{1i}^T H_{3i} e_4 + \lambda e_1 H_{1i}^T H_{4i} e_5 + \lambda e_1 H_{1i}^T H_{5i} e_{24} \\
&\quad + \lambda e_3 H_{2i}^T H_{2i} e_3 + \lambda e_3 H_{2i}^T H_{3i} e_4 + \lambda e_3 H_{2i}^T H_{4i} e_5 + \lambda e_3 H_{2i}^T H_{5i} e_{24} + \lambda e_4 H_{3i}^T H_{3i} e_4 \\
&\quad + \lambda e_4 H_{3i}^T H_{4i} e_5 + \lambda e_4 H_{3i}^T H_{5i} e_{24} + \lambda e_5 H_{4i}^T H_{4i} e_5 + \lambda e_5 H_{4i}^T H_{5i} e_{24} \\
&\quad + \lambda e_{25} H_{5i}^T H_{5i} e_{24}], \\
\Phi_1 \quad &= [L_1 F + L_2 F, L_3 F, L_4 F, L_5 F, L_6 F, \overbrace{0, \cdots, 0}^{23 \; times}, L_7 F]^T
\end{aligned}
$$

*proof* Replacing $A_i$, $B_i$, $C_i$, $D_i$ and $E_i$ with $A_i + FG(t)H_{1i}$, $B_i + FG(t)H_{2i}$, $C_i + FG(t)H_{3i}$, $D_i + FG(t)H_{4i}$ and $E_i + FG(t)H_{5i}$ in (2), respectively,

$$
\Omega_1 + \begin{bmatrix} L_1 F + L_2 F \\ L_3 F \\ L_4 F \\ L_5 F \\ L_6 F \\ 0 \\ \vdots \\ 0 \\ L_7 F \end{bmatrix} G(t) \begin{bmatrix} H_{1i} & 0 & H_{2i} & h_{3i} & H_{4i} & 0 & \cdots & 0 H_{5i} \end{bmatrix}
$$

$$
+ \lambda \begin{bmatrix} H_{1i}^T \\ 0 \\ H_{2i}^T \\ H_{3i}^T \\ H_{4i}^T \\ 0 \\ \vdots \\ 0 \\ H_{5i} \end{bmatrix} G^T(t) \times \begin{bmatrix} L_1 F^T + L_2 F^T & L_3 F^T & L_4 F^T & L_5 F^T & L_6 F^T & 0 & \cdots & 0 & L_7 F^T \end{bmatrix} < 0. \quad (10)
$$

Since the lemma 5, there are some real numbers $\lambda > 0$ to result in system (10) true that lead to following inequality:

$$
\Omega_1 + \lambda^{-1}
\begin{bmatrix}
L_1F + L_2F \\
L_3F \\
L_4F \\
L_5F \\
L_6F \\
0 \\
\vdots \\
0 \\
L_7F
\end{bmatrix}
\times
\begin{bmatrix} L_1F^T + L_2F^T & L_3F^T & L_4F^T & L_5F^T & L_6F^T & 0 & \cdots & 0 & L_7F^T \end{bmatrix}
$$

$$
+ \lambda
\begin{bmatrix}
H_{1i}^T \\
0 \\
H_{2i}^T \\
H_{3i}^T \\
H_{4i}^T \\
0 \\
\vdots \\
0 \\
H_{5i}^T
\end{bmatrix}
\begin{bmatrix} H_{1i} & 0 & H_{2i} & H_{3i} & H_{4i} & 0 & \cdots & 0 & H_{5i}^T \end{bmatrix} < 0. \tag{11}
$$

From Lemma 7, Eq (11) is equivalent to Eq (2). The proof is completed. Now the system (1) when $E_i + \Delta E_i(t) = 0$ is demonstrated.

**Corollary 1** *For given constants $\sigma_1, \sigma_2, \tau_1, \tau_2, h_1, h_2 \geq 0$ system (1) with uncertain terms is exponential stable. If there are real positive definite matrices $L_1, Q_1, Q_2, Q_3, R_1, R_2, R_3, R_4, Z_1, Z_2, Z_3, W_1, W_2, W_3, W_4, U_1, U_2, U_3, U_4, S_1, S_2, S_3, S_4$ and a positive $\lambda$ agreeable the ensuing LMI holds for $k = 1, 2, \ldots, m$:*

$$
\Omega_{2k} =
\begin{bmatrix}
\psi_{3k} & \Phi_2 \\
* & -\lambda_2 I
\end{bmatrix} < 0, \tag{12}
$$

*where*

$$\psi_{3k} = \Sigma,$$

$$\Sigma = \Xi_{1k} + \Xi_2 + \Xi_3 + \Xi_4 + \Xi_5 + \Xi_6 + \Xi_7 + \Xi_8,$$

$$\Xi_{1k} = [2e_1 L_1 A_k e_1 + 2e_1 L_1 B_k e_3 + 2e_1 L_1 C_k e_4 + 2e_1 L_1 D_k e_5 + 2e_1 L_2 e_2 - 2e_1 L_2 A_k e_1$$

$$- 2e_1 L_2 B_k e_3 - 2e_1 L_2 C_k e_4 - 2e_1 L_2 D_k e_5 - 2e_2 L_3 e_2 + 2e_2 L_3 A_k e_1 + 2e_2 L_3 B_k e_3$$

$$+ e_2 C_k L_3 e_4 + 2e_2 D_k L_3 e_5 + 2e_3 L_4 e_2 - 2e_3 L_4 A_k e_1 - 2e_3 L_4 B_k e_3 - e_3 C_k L_4 e_4$$

$$- 2e_3 D_k L_4 e_5 + 2e_4 L_5 e_2 - 2e_4 L_5 A_k e_1 - 2e_4 L_5 B_k e_3 - 2e_4 L_5 C_k e_4 - 2e_4 L_5 D_k e_5$$

$$+ 2e_5 L_6 e_2 - 2e_5 L_5 A_k e_1 - 2e_5 L_6 B_k e_3 - 2e_5 L_6 C_k e_4 - 2e_5 L_6 D_k e_5],$$

$$\Xi_2 = [e_1 Q_1 e_1 - e^{-2\alpha\tau_1} e_6 Q_1 e_6 + e_1 Q_2 e_1 - e^{-2\alpha\sigma_1} e_7 Q_2 e_7 + e_1 Q_3 e_1 - e^{-2\alpha h_1} e_8 Q_3 e_8$$

$$+ e_1 R_1 e_1 - e^{-2\alpha\tau_2} e_9 R_1 e_9 + \tau_d e^{-2\alpha\tau_1} e_9 R_1 e_9 + e_1 R_2 e_1 - e^{-2\alpha\sigma_2} e_3 R_2 e_3$$

$$+ \sigma_d e^{-2\alpha\sigma_1} e_3 R_2 e_3 + e_1 R_3 e_1 - e^{-2\alpha h_2} e_{10} R_3 e_{10} + h_d e^{-2\alpha h_1} e_{10} R_3 e_{10} + e_2 R_4 e_2$$

$$- e^{-2\alpha\tau_2} e_4 R_4 e_4 + \tau_d e^{-2\alpha\tau_1} e_4 R_4 e_4],$$

$$\Xi_3 = \sigma_2^2 e_2 S_1 e_2 - e^{-2\alpha\sigma_2}[e_1 - e_{11}]^T S_1 [e_1 - e_{11}]$$

$$- 3e^{-2\alpha\sigma_2}\left[e_1 + e_{11} - \frac{2}{\sigma_2}e_{12}\right]^T S_1 \left[e_1 + e_{11} - \frac{2}{\sigma_2}e_{12}\right]$$

$$- 5e^{-2\alpha\sigma_2}\left[e_1 - e_{11} + \frac{6}{\sigma_2}e_{12} - \frac{12}{\sigma_2^2}e_{13}\right]^T S_1 \left[e_1 - e_{11} + \frac{6}{\sigma_2}e_{12} - \frac{12}{\sigma_2^2}e_{13}\right]$$

$$- 7e^{-2\sigma_2}\left[e_1 + e_{11} - \frac{12}{\sigma_2}e_{12} + \frac{60}{\sigma_2^2}e_{13} - \frac{120}{\sigma_2^3}e_{14}\right]^T$$

$$\times S_1 \left[e_1 + e_{11} - \frac{12}{\sigma_2}e_{12} + \frac{60}{\sigma_2^2}e_{13} - \frac{120}{\sigma_2^3}e_{14}\right]$$

$$+ \sigma_1^2 e_2 S_2 e_2 - e^{-2\alpha\sigma_1}[e_1 - e_7]^T S_2 [e_1 - e_7]$$

$$- 3e^{-2\alpha\sigma_1}\left[e_1 + e_7 - \frac{2}{\sigma_1}e_{15}\right]^T S_2 \left[e_1 + e_7 - \frac{2}{\sigma_1}e_{15}\right]$$

$$- 5e^{-2\alpha\sigma_1}\left[e_1 - e_7 + \frac{6}{\sigma_1}e_{15} - \frac{12}{\sigma_1^2}e_{16}\right]^T S_2 \left[e_1 - e_7 + \frac{6}{\sigma_1}e_{15} - \frac{12}{\sigma_1^2}e_{16}\right]$$

$$- 7e^{-2\sigma_1}\left[e_1 + e_7 - \frac{12}{\sigma_1}e_{15} + \frac{60}{\sigma_1^2}e_{16} - \frac{120}{\sigma_1^3}e_{17}\right]^T$$

$$\times S_2\left[e_1 + e_7 - \frac{12}{\sigma_1}e_{15} + \frac{60}{\sigma_1^2}e_{16} - \frac{120}{\sigma_1^3}e_{17}\right]$$

$$+ \tau_2^2 e_2 S_3 e_2 - e^{-2\alpha\tau_2}[e_1 - e_{18}]^T S_3 [e_1 - e_{18}]$$

$$- 3e^{-2\alpha\tau_2}\left[e_1 + e_{18} - \frac{2}{\tau_2}e_{19}\right]^T S_3\left[e_1 + e_{18} - \frac{2}{\tau_2}e_{19}\right]$$

$$- 5e^{-2\alpha\tau_2}\left[e_1 - e_{18} + \frac{6}{\tau_2}e_{19} - \frac{12}{\tau_2^2}e_{20}\right]^T S_3\left[e_1 - e_{18} + \frac{6}{\tau_2}e_{19} - \frac{12}{\tau_2^2}e_{20}\right]$$

$$- 7e^{-2\tau_2}\left[e_1 + e_{18} - \frac{12}{\tau_2}e_{19} + \frac{60}{\tau_2^2}e_{20} - \frac{120}{\tau_2^3}e_{21}\right]^T$$

$$\times S_3\left[e_1 + e_{18} - \frac{12}{\tau_2}e_{19} + \frac{60}{\tau_2^2}e_{20} - \frac{120}{\tau_2^3}e_{21}\right]$$

$$+ \tau_1^2 e_2 S_4 e_2 - e^{-2\alpha\tau_1}[e_1 - e_6]^T S_4 [e_1 - e_6]$$

$$- 3e^{-2\alpha\tau_1}\left[e_1 + e_6 - \frac{2}{\tau_1}e_{22}\right]^T S_4\left[e_1 + e_6 - \frac{2}{\tau_1}e_{22}\right]$$

$$- 5e^{-2\alpha\tau_1}\left[e_1 - e_6 + \frac{6}{\tau_1}e_{22} - \frac{12}{\tau_1^2}e_{23}\right]^T S_4\left[e_1 - e_6 + \frac{6}{\tau_1}e_{22} - \frac{12}{\tau_1^2}e_{23}\right]$$

$$- 7e^{-2\tau_1}\left[e_1 + e_6 - \frac{12}{\tau_1}e_{22} + \frac{60}{\tau_1^2}e_{23} - \frac{120}{\tau_1^3}e_{24}\right]^T$$

$$\times S_4\left[e_1 + e_6 - \frac{12}{\tau_1}e_{22} + \frac{60}{\tau_1^2}e_{23} - \frac{120}{\tau_1^3}e_{24}\right],$$

$$\Xi_4 = [(\tau_2 - \tau_1)^2 e_2 Z_1 e_2 - e^{-2\alpha\tau_2}e_{25}Z_1 e_{25} - e^{-2\alpha\tau_2}e_{26}Z_1 e_{26} + (\sigma_2 - \sigma_1)^2 e_2 Z_2 e_2$$

$$- e^{-2\alpha\sigma_2}e_{27}Z_2 e_{27} - e^{-2\alpha\tau_2}e_{28}Z_2 e_{28} + h_2^2 e_2 Z_3 e_2 - e^{2\alpha h_2}e_{29}Z_3 e_{29}],$$

$$\Xi_5 = \sigma_2^2 e_2 W_1 e_2 - 2e^{-2\alpha\sigma_2}\left[e_1 - \frac{1}{\sigma_2}e_{12}\right]W_1\left[e_1 - \frac{1}{\sigma_2}e_{12}\right]$$

$$- 4e^{-2\alpha\sigma_2}\left[e_1 + \frac{2}{\sigma_2}e_{12} - \frac{6}{\sigma_2^2}e_{13}\right]W_1\left[e_1 + \frac{2}{\sigma_2}e_{12} - \frac{6}{\sigma_2^2}e_{13}\right]$$

$$- 6e^{-2\alpha\sigma_2} \left[ e_1 - \frac{3}{\sigma_2}e_{12} + \frac{24}{\sigma_2^2}e_{13} - \frac{60}{\sigma_2^3}e_{14} \right] W_1 \left[ e_1 - \frac{3}{\sigma_2}e_{12} + \frac{24}{\sigma_2^2}e_{13} - \frac{60}{\sigma_2^3}e_{14} \right]$$

$$+ \sigma_1^2 e_2 W_2 e_2 - 2e^{-2\alpha\sigma_1} \left[ e_1 - \frac{1}{\sigma_1}e_{15} \right] W_2 \left[ e_1 - \frac{1}{\sigma_1}e_{15} \right]$$

$$- 4e^{-2\alpha\sigma_1} \left[ e_1 + \frac{2}{\sigma_1}e_{15} - \frac{6}{\sigma_1^2}e_{16} \right] W_2 \left[ e_1 + \frac{2}{\sigma_1}e_{15} - \frac{6}{\sigma_1^2}e_{16} \right]$$

$$- 6e^{-2\alpha\sigma_1} \left[ e_1 - \frac{3}{\sigma_1}e_{15} + \frac{24}{\sigma_1^2}e_{16} - \frac{60}{\sigma_1^3}e_{17} \right] W_2 \left[ e_1 - \frac{3}{\sigma_1}e_{15} + \frac{24}{\sigma_1^2}e_{16} - \frac{60}{\sigma_1^3}e_{17} \right]$$

$$+ \tau_2^2 e_2 W_3 e_2 - 2e^{-2\alpha\tau_2} \left[ e_1 - \frac{1}{\tau_2}e_{19} \right] W_3 \left[ e_1 - \frac{1}{\tau_2}e_{19} \right]$$

$$- 4e^{-2\alpha\tau_2} \left[ e_1 + \frac{2}{\tau_2}e_{19} - \frac{6}{\tau_2^2}e_{20} \right] W_3 \left[ e_1 + \frac{2}{\tau_2}e_{19} - \frac{6}{\tau_2^2}e_{20} \right]$$

$$- 6e^{-2\alpha\tau_2} \left[ e_1 - \frac{3}{\tau_2}e_{19} + \frac{24}{\tau_2^2}e_{20} - \frac{60}{\tau_2^3}e_{21} \right] W_3 \left[ e_1 - \frac{3}{\tau_2}e_{19} + \frac{24}{\tau_2^2}e_{20} - \frac{60}{\tau_2^3}e_{21} \right]$$

$$+ \tau_1^2 e_2 W_4 e_2 - 2e^{-2\alpha\tau_1} \left[ e_1 - \frac{1}{\tau_1}e_{22} \right] W_4 \left[ e_1 - \frac{1}{\tau_1}e_{22} \right]$$

$$- 4e^{-2\alpha\tau_1} \left[ e_1 + \frac{2}{\tau_1}e_{22} - \frac{6}{\tau_1^2}e_{23} \right] W_4 \left[ e_1 + \frac{2}{\tau_1}e_{22} - \frac{6}{\tau_1^2}e_{23} \right]$$

$$- 6e^{-2\alpha\tau_1} \left[ e_1 - \frac{3}{\tau_1}e_{22} + \frac{24}{\tau_1^2}e_{23} - \frac{60}{\tau_1^3}e_{24} \right] W_4 \left[ e_1 - \frac{3}{\tau_1}e_{22} + \frac{24}{\tau_1^2}e_{23} - \frac{60}{\tau_1^3}e_{24} \right],$$

$$\Xi_6 = \frac{\sigma_2^3}{6} e_2 U_1 e_2 - 6e^{-2\alpha\sigma_2} \left[ \frac{\sigma_2^2}{2}e_1 - e_{13} \right] U_1 \left[ \frac{\sigma_2^2}{2}e_1 - e_{13} \right]$$

$$- 10e^{-2\alpha\sigma_2} \left[ \frac{\sigma_2^2}{2}e_1 - e_{13} + \frac{4}{\sigma_2}e_{14} \right] U_1 \left[ \frac{\sigma_2^2}{2}e_1 - e_{13} + \frac{4}{\sigma_2}e_{14} \right]$$

$$+ \frac{\sigma_1^3}{6} e_2 U_2 e_2 - 6e^{-2\alpha\sigma_1} \left[ \frac{\sigma_1^2}{2}e_1 - e_{16} \right] U_2 \left[ \frac{\sigma_1^2}{2}e_1 - e_{16} \right]$$

$$- 10e^{-2\alpha\sigma_1} \left[ \frac{\sigma_1^2}{2}e_1 - e_{16} + \frac{4}{\sigma_1}e_{17} \right] U_2 \left[ \frac{\sigma_1^2}{2}e_1 - e_{16} + \frac{4}{\sigma_1}e_{17} \right]$$

$$+ \frac{\tau_2^3}{6} e_2 U_3 e_2 - 6e^{-2\alpha\tau_2} \left[ \frac{\tau_2^2}{2}e_1 - e_{20} \right] U_3 \left[ \frac{\tau_2^2}{2}e_1 - e_{20} \right]$$

$$- 10e^{-2\alpha\tau_2} \left[ \frac{\tau_2^2}{2} e_1 - e_{20} + \frac{4}{\tau_2} e_{21} \right] U_3 \left[ \frac{\tau_2^2}{2} e_1 - e_{20} + \frac{4}{\tau_2} e_{21} \right]$$

$$+ \frac{\tau_1^3}{6} e_2 U_4 e_2 - 6e^{-2\alpha\tau_1} \left[ \frac{\tau_1^2}{2} e_1 - e_{23} \right] U_4 \left[ \frac{\tau_1^2}{2} e_1 - e_{23} \right]$$

$$- 10e^{-2\alpha\tau_1} \left[ \frac{\tau_1^2}{2} e_1 - e_{23} + \frac{4}{\tau_1} e_{24} \right] U_4 \left[ \frac{\tau_1^2}{2} e_1 - e_{23} + \frac{4}{\tau_1} e_{24} \right]$$

$$\Xi_7 = [e_6 - e_9 - e_{25}]y_1[-e_6 + e_9 + e_{25}] + [e_9 - e_{18} - e_{26}]y_2[-e_9 + e_{18} + e_{26}]$$

$$+ [e_7 - e_3 - e_{27}]y_3[-e_7 + e_3 + e_{27}] + [e_3 - e_{11} - e_{28}]y_4[-e_3 + e_{11} + e_{28}]$$

$$+ [e_1 - e_{10} - e_{29}]y_5[-e_1 + e_{10} + e_{29}],$$

$$\Xi_8 = [\lambda e_1 H_{1i}^T H_{1i} e_1 + \lambda e_1 H_{1i}^T H_{2i} e_3 + \lambda e_1 H_{1i}^T H_{3i} e_4 + \lambda e_1 H_{1i}^T H_{4i} e_5 + \lambda e_3 H_{2i}^T H_{2i} e_3$$

$$+ \lambda e_3 H_{2i}^T H_{3i} e_4 + \lambda e_4 H_{3i}^T H_{4i} e_5 + \lambda e_5 H_{4i}^T H_{4i} e_5],$$

$$\Phi_2 = [L_1 F + L_2 F, L_3 F, L_4 F, L_5 F, \overbrace{0, \cdots, 0}^{23 \; times}, L_6 F]^T,$$

$$\xi(t) = [x(t), \dot{x}(t), x(t - \sigma(t)), \dot{x}(t - \tau(t)), \int_{t-h(t)}^{t} x(s)ds, x(t - \tau_1), x(t - \sigma_1), x(t - h_1),$$

$$x(t - \tau(t)), x(t - h(t)), x(t - \sigma_2), \int_{t-\sigma_2}^{t} x(s)ds, \int_{t-\sigma_2}^{t} \int_{s}^{t} x(u)duds,$$

$$\int_{t-\sigma_2}^{t} \int_{s}^{t} \int_{u}^{t} x(v)dvduds, \int_{t-\sigma_1}^{t} x(s)ds, \int_{t-\sigma_1}^{t} \int_{s}^{t} x(u)duds,$$

$$\int_{t-\sigma_1}^{t} \int_{s}^{t} \int_{u}^{t} x(v)dvduds, x(t - \tau_2), \int_{t-\tau_2}^{t} x(s)ds, \int_{t-\tau_2}^{t} \int_{s}^{t} x(u)duds,$$

$$\int_{t-\tau_2}^{t} \int_{s}^{t} \int_{u}^{t} x(v)dvduds, \int_{t-\tau_1}^{t} x(s)ds, \int_{t-\tau_1}^{t} \int_{s}^{t} x(u)duds, \int_{t-\tau_1}^{t} \int_{s}^{t} \int_{u}^{t} x(v)dvduds,$$

$$\int_{t-\tau(t)}^{t-\tau_1} \dot{x}(s)ds, \int_{t-\tau_2}^{t-\tau(t)} \dot{x}(s)ds, \int_{t-\sigma(t)}^{t-\sigma_1} \dot{x}(s)ds, \int_{t-\sigma_2}^{t-\sigma(t)} \dot{x}(s)ds, \int_{t-h(t)}^{t} \dot{x}(s)ds]^T.$$

*Then the system* (1) *when $E_i + \Delta E_i(t) = 0$ is exponential stability.*

After that, this study shall present the delay-dependent condition of the passivity and exponential stability for system (1) when $C_i + \Delta C_i(t) = D_i + \Delta D_i(t) = 0$.

**Theorem 3** *For given a constant $\sigma_2 \geq 0$, system* (1) *where $C_i + \Delta C_i(t) = D_i + \Delta D_i(t) = 0$ with uncertain terms is exponentially passive. If there are real positive definite matrices $L_1$, $R_1$, $Z_1$, $W_1$,*

$U_1$, $S_1$, $S_2$ and a positive $\lambda$ agreeable the following LMI holds for $k = 1, 2, \ldots, m$:

$$\Omega_{3k} = \begin{bmatrix} \psi_{4k} & \Phi_3 \\ * & -\lambda_3 I \end{bmatrix} < 0, \tag{13}$$

*where*

$$\psi_{4k} = \Sigma^*,$$

$$\Sigma^* = \Xi_{1k}^* + \Xi_2^* + \Xi_3^* + \Xi_4^* + \Xi_5^* + \Xi_6^* + \Xi_7^* + \Xi_8^* + \Xi_9^*,$$

$$\Xi_{1k}^* = [2e_1 L_1 A_k e_1 + 2e_1 L_1 B_k e_3 + 2e_1 L_1 E_k e_{11} + 2e_1 L_2 e_2 - 2e_1 L_2 A_k e_1 - 2e_1 L_2 B_k e_3$$

$$- 2e_1 L_2 E_k e_{11} - 2e_2 L_3 e_2 + 2e_2 L_2 A_k e_1 + 2e_2 L_3 B_k e_3 + 2e_2 L_3 E_k e_{11} + 2e_3 L_4 e_2$$

$$- 2e_3 L_4 A_k e_1 - 2e_3 L_4 B_k e_3 - 2e_3 L_4 E_k e_{11} + 2e_{11} L_5 e_2 - 2e_{11} L_5 A_k e_1 - 2e_{11} L_5 B_k e_3$$

$$- 2e_{11} L_5 E_k e_{11} + 2\alpha e_1 L_1 e_1],$$

$$\Xi_2^* = [e_1 Q_1 e_1 - \sigma_2 e^{-2\alpha\sigma_2} e_4 Q_1 e_4 + e_1 R_1 e_1 - (1 - \dot{\sigma}) e^{-2\alpha\sigma_2} e_8 R_1 e_8],$$

$$\Xi_3^* = \sigma_2^2 e_2 S_1 e_2 - e^{-2\alpha\sigma_2} [e_1 - e_4]^T S_1 [e_1 - e_4]$$

$$- 3e^{-2\alpha\sigma_2} \left[ e_1 + e_4 - \frac{2}{\sigma_2} e_5 \right]^T S_1 \left[ e_1 + e_4 - \frac{2}{\sigma_2} e_5 \right]$$

$$- 5e^{-2\alpha\sigma_2} \left[ e_1 - e_4 + \frac{6}{\sigma_2} e_5 - \frac{12}{(\sigma_2)^2} e_6 \right]^T S_1 \left[ e_1 - e_4 + \frac{6}{\sigma_2} e_5 - \frac{12}{(\sigma_2)^2} e_6 \right]$$

$$- 7e^{-2\sigma_2} \left[ e_1 + e_4 - \frac{12}{\sigma_2} e_5 + \frac{60}{(\sigma_2)^2} e_6 - \frac{120}{(\sigma_2)^3} e_7 \right]^T$$

$$\times S_1 \left[ e_1 + e_4 - \frac{12}{\sigma_2} e_5 + \frac{60}{(\sigma_2)^2} e_6 - \frac{120}{(\sigma_2)^3} e_7 \right],$$

$$\Xi_4^* = [\sigma_2^2 e_2 Z_1 e_2 - e^{-2\alpha\sigma_2} e_9 Z_1 e_9 - e^{-2\alpha\sigma_2} e_{10} Z_1 e_{10}],$$

$$\Xi_5^* = \frac{\sigma_2^4}{2} e^{-2\alpha\sigma_2} e_2 W_1 e_2 - 2e^{-2\alpha\sigma_2} \sigma_2^2 \left[ e_1 - \frac{1}{\sigma_2} e_5 \right] W_1 \left[ e_1 - \frac{1}{\sigma_2} e_5 \right]$$

$$- 4\sigma_2^2 e^{-2\alpha\sigma_2} \left[ e_1 + \frac{2}{(\sigma_2} e_5 - \frac{6}{\sigma_2^2} e_6 \right] W_1 \left[ e_1 + \frac{2}{\sigma_2} e_5 - \frac{6}{\sigma_2^2} e_6 \right]$$

$$- 6\sigma_2^2 e^{-2\alpha\sigma_2} \left[ e_1 - \frac{3}{\sigma_2} e_5 + \frac{24}{\sigma_2^2} e_6 - \frac{60}{\sigma_2^3} e_7 \right] W_1 \left[ e_1 - \frac{3}{\sigma_2} e_5 + \frac{24}{\sigma_2^2} e_6 - \frac{60}{\sigma_2^3} e_7 \right]$$

$$\Xi_6^* = \frac{\sigma_2^6}{6} e_2 U_1 e_2 - 6e^{-2\alpha\sigma_2} \left[ \frac{\sigma_2^2}{2} e_1 - e_6 \right] U_1 \left[ \frac{\sigma_2^2}{2} e_1 - e_6 \right]$$

$$- 10e^{-2\alpha\sigma_2} \left[ \frac{\sigma_2^2}{6} e_1 - e_6 + \frac{4}{\sigma_2} e_7 \right] U_1 \left[ \frac{\sigma_2^2}{6} e_1 - e_6 + \frac{4}{\sigma_2} e_7 \right],$$

$$\Xi_7^* = [e_1 y_1 e_1 - e_3 y_1 e_3 - e_9 y_1 e_9 + e_3 y_2 e_3 - e_4 y_2 e_4 - e_{10} y_2 e_{10}],$$

$$\Xi_8^* = [\lambda e_1 H_{1i} H_{1i} e_1 + \lambda e_1 H_{1i} H_{2i} e_3 + \lambda e_1 H_{1i} H_{3i} e_{11} + \lambda e_3 H_{2i} H_{2i} e_3 + \lambda e_3 H_{2i} H_{3i} e_{11}$$

$$+ \lambda e_{11} H_{3i} H_{3i} e_{11}]$$

$$\Xi_9^* = [-2e_1 \tilde{A} e_{11} - 2e_3 \tilde{B} e_{11} - 2e_{11} \tilde{E} e_{11}],$$

$$\Phi_3 = [L_1 F + L_2 F, L_3 F, L_4 F, \overbrace{0, \cdots, 0}^{7 \ times}, L_5 F]^T,$$

$$\xi(t) = \left[ x(t), \dot{x}(t), x(t - \sigma(t)), x(t - \sigma_2), \int_{t-\sigma_2}^t x(s)ds, \int_{t-\sigma_2}^t \int_s^t x(u)duds, \right.$$

$$\left. \int_{t-\sigma_2}^t \int_s^t \int_u^t x(v)dvduds, \dot{x}(t - \sigma(t)), \int_{t-\sigma(t)}^t \dot{x}(s)ds, \int_{t-\sigma_2}^{t-\sigma(t)} \dot{x}(s)ds, u(t) \right]^T.$$

*proof* This study focal point in the following Lyapunov-Krasovskii function of the system (1) where $C_i + \Delta C_i(t) = D_i + \Delta D_i(t) = 0$

$$V(x(t)) = \sum_{i=1}^6 V_i(x(t)),$$

*where*

$$V_1 = x^T(t) L_1 x(t),$$

$$V_2 = \int_{t-\sigma_2}^t e^{2\alpha(s-t)} x^T(s) Q_1 x(s)ds + \int_{t-\sigma(t)}^t e^{2\alpha(s-t)} \dot{x}^T(s) R_1 \dot{x}(s)ds,$$

$$V_3 = \sigma_2 \int_{-\sigma_2}^0 \int_{t+\theta}^t e^{2\alpha(s-t)} \dot{x}^T(s) S_1 \dot{x}(s)dsd\theta,$$

$$V_4 = \sigma_2 \int_{-\sigma_2}^0 \int_{t+\theta}^t e^{2\alpha(s-t)} \dot{x}^T(s) Z_1 \dot{x}(s)dsd\theta,$$

$$V_5 = \sigma_2^2 \int_{-\sigma_2}^0 \int_\theta^0 \int_{t+\beta}^t e^{2\alpha(s-t)} \dot{x}^T(s) W_1 \dot{x}(s)dsd\beta d\theta,$$

$$V_6 = \sigma_2^3 \int_{-\sigma_2}^0 \int_v^0 \int_\theta^0 \int_{t+\beta}^t e^{2\alpha(s-t)} \dot{x}^T(s) U_1 \dot{x}(s)dsd\beta d\theta dv.$$

Abovementioned by Theorem 1 and Theorem 2, this study obtain the exponentially passive for delay-dependent criteria of systems (1) when $C_i + \Delta C_i(t) = D_i + \Delta D_i(t) = 0$.

Now the system (1) when $C_i + \Delta C_i(t) = D_i + \Delta D_i(t) = 0$ and $E_i + \Delta E_i(t) = 0$ is demonstrated.

**Remark 3** *If $C_i + \Delta C_i(t) = D_i + \Delta D_i(t) = 0$ and $E_i + \Delta E_i(t) = 0$ the fuzzy replica (1) become the T-S fuzzy of neutral differential system presented by [7, 26–29].*

**Corollary 2** *For given a constant $\sigma_2 \geq 0$, system (1) when $C_i + \Delta C_i(t) = D_i + \Delta D_i(t) = 0$ and $E_i + \Delta E_i(t) = 0$ with uncertain terms is exponentially. If there are matrices $L_1, R_1, Z_1, W_1, U_1, S_1, S_2 > 0$ and a positive $\lambda$ agreeable the LMI for $k = 1, 2, \ldots, m$:*

$$\Omega_{4k} = \begin{bmatrix} \psi_{5k} & \Phi_4 \\ * & -\lambda_4 I \end{bmatrix} < 0,$$

*where*

$$\psi_{5k} = \Sigma^*,$$

$$\Sigma^* = \Xi_{1k}^* + \Xi_2^* + \Xi_3^* + \Xi_4^* + \Xi_5^* + \Xi_6^* + \Xi_7^* + \Xi_8^*,$$

$$\Xi_{1k}^* = [2e_1 L_1 A_k e_1 + 2e_1 L_1 B_k e_3 + 2e_1 L_1 E_k e_{11} + 2e_1 L_2 e_2 - 2e_1 L_2 A_k e_1 - 2e_1 L_2 B_k e_3$$

$$- 2e_2 L_3 e_2 + 2e_2 L_2 A_k e_1 + 2e_2 L_3 B_k e_3 + 2e_3 L_4 e_2 - 2e_3 L_4 A_k e_1 - 2e_3 L_4 B_k e_3$$

$$+ 2\alpha e_1 L_1 e_1],$$

$$\Xi_2^* = [e_1 Q_1 e_1 - \sigma_2 e^{-2\alpha\sigma_2} e_4 Q_1 e_4 + e_1 R_1 e_1 - (1 - \dot{\sigma}) e^{-2\alpha\sigma_2} e_8 R_1 e_8],$$

$$\Xi_3^* = \sigma_2^2 e_2 S_1 e_2 - e^{-2\alpha\sigma_2} [e_1 - e_4]^T S_1 [e_1 - e_4]$$

$$- 3e^{-2\alpha\sigma_2} \left[ e_1 + e_4 - \frac{2}{\sigma_2} e_5 \right]^T S_1 \left[ e_1 + e_4 - \frac{2}{\sigma_2} e_5 \right]$$

$$- 5e^{-2\alpha\sigma_2} \left[ e_1 - e_4 + \frac{6}{\sigma_2} e_5 - \frac{12}{(\sigma_2)^2} e_6 \right]^T S_1 \left[ e_1 - e_4 + \frac{6}{\sigma_2} e_5 - \frac{12}{(\sigma_2)^2} e_6 \right]$$

$$- 7e^{-2\sigma_2} \left[ e_1 + e_4 - \frac{12}{\sigma_2} e_5 + \frac{60}{(\sigma_2)^2} e_6 - \frac{120}{(\sigma_2)^3} e_7 \right]^T$$

$$\times S_1 \left[ e_1 + e_4 - \frac{12}{\sigma_2} e_5 + \frac{60}{(\sigma_2)^2} e_6 - \frac{120}{(\sigma_2)^3} e_7 \right],$$

$$\Xi_4^* = [\sigma_2^2 e_2 Z_1 e_2 - e^{-2\alpha\sigma_2} e_9 Z_1 e_9 - e^{-2\alpha\sigma_2} e_{10} Z_1 e_{10}],$$

$$\Xi_5^* = \frac{\sigma_2^4}{2} e^{-2\alpha\sigma_2} e_2 W_1 e_2 - 2e^{-2\alpha\sigma_2} \sigma_2^2 \left[ e_1 - \frac{1}{\sigma_2} e_5 \right] W_1 \left[ e_1 - \frac{1}{\sigma_2} e_5 \right]$$

$$- 4\sigma_2^2 e^{-2\alpha\sigma_2} \left[ e_1 + \frac{2}{\sigma_2} e_5 - \frac{6}{\sigma_2^2} e_6 \right] W_1 \left[ e_1 + \frac{2}{\sigma_2} e_5 - \frac{6}{\sigma_2^2} e_6 \right]$$

$$-6\sigma_2^2 e^{-2\alpha\sigma_2}\left[e_1 - \frac{3}{\sigma_2}e_5 + \frac{24}{\sigma_2^2}e_6 - \frac{60}{\sigma_2^3}e_7\right]W_1\left[e_1 - \frac{3}{\sigma_2}e_5 + \frac{24}{\sigma_2^2}e_6 - \frac{60}{\sigma_2^3}e_7\right]$$

$$\Xi_6^* = \frac{\sigma_2^6}{6}e_2 U_1 e_2 - 6e^{-2\alpha\sigma_2}\left[\frac{\sigma_2^2}{2}e_1 - e_6\right]U_1\left[\frac{\sigma_2^2}{2}e_1 - e_6\right]$$

$$-10e^{-2\alpha\sigma_2}\left[\frac{\sigma_2^2}{6}e_1 - e_6 + \frac{4}{\sigma_2}e_7\right]U_1\left[\frac{\sigma_2^2}{6}e_1 - e_6 + \frac{4}{\sigma_2}e_7\right],$$

$$\Xi_7^* = [e_1 y_1 e_1 - e_3 y_1 e_3 - e_9 y_1 e_9 + e_3 y_2 e_3 - e_4 y_2 e_4 - e_{10} y_2 e_{10}],$$

$$\Xi_8^* = [\lambda e_1 H_{1i} H_{1i} e_1 + \lambda e_1 H_{1i} H_{2i} e_3 + \lambda e_3 H_{2i} H_{2i} e_3],$$

$$\Phi_4 = [L_1 F + L_2 F, L_3 F, \overbrace{0, \cdots, 0}^{7 \ times}, L_4 F]^T,$$

$$\xi(t) = [x(t), \dot{x}(t), x(t - \sigma(t)), x(t - \sigma_2), \int_{t-\sigma_2}^t x(s)ds, \int_{t-\sigma_2}^t \int_s^t x(u)duds,$$

$$\int_{t-\sigma_2}^t \int_s^t \int_u^t x(v)dvduds, \dot{x}(t - \sigma(t)), \int_{t-\sigma(t)}^t \dot{x}(s)ds, \int_{t-\sigma_2}^{t-\sigma(t)} \dot{x}(s)ds]^T.$$

Then the system (1) when $C_i + \Delta C_i(t) = D_i + \Delta D_i(t) = 0$ and $E_i + \Delta E_i(t) = 0$ is exponentially stability.

**Remark 4** *According to Corollary 2 that using Lemmas 2, 3 and Lemma 4 yielded fewer conservative outcomes than other results, [7, 26–29] which illustrate in Table 3. Even, these lemmas contain a large number of free weighting matrices, that could bring about their more calculation intricately.*

After that, this study shall present the delay-dependent condition of the passivity and exponential stability for system (1) when $D_i + \Delta D_i(t) = 0$.

**Theorem 4** *For given constants $\sigma_2, \tau_2 \geq 0$ systems (1) where $D_i + \Delta D_i(t) = 0$ with uncertain is exponentially passive. If there are positive real symmetric matrices $L_1, R_1, R_2, Q_1, Q_2, Z_1, Z_2, W_1, W_2, U_1, U_2$ and a positive $\lambda$ agreeable the LMI for k = 1, 2, . . ., m:*

$$\Omega_5 = \begin{bmatrix} \psi_6 & \Phi_5 \\ * & -\lambda_5 I \end{bmatrix} < 0, \tag{14}$$

*where*

$$\psi_{6k} \quad = \Sigma^{**},$$

$$\Sigma^{**} \quad = \Xi_{1k}^{**} + \Xi_2^{**} + \Xi_3^{**} + \Xi_4^{**} + \Xi_5^{**} + \Xi_6^{**} + \Xi_7^{**}, + \Xi_8^{**} + \Xi_9^{**},$$

$$\Xi_{1k}^{**} \quad = [2e_1 L_1 e_1 + 2e_1 L_1 A_k e_3 + 2e_1 L_1 C_k e_4 + 2e_1 L_1 E_k e_{18} + 2e_1 L_2 e_2 - 2e_1 L_2 A_k e_1$$

$$- 2e_1 L_2 B_k e_3 - 2e_1 L_2 C_k e_4 - 2e_2 L_2 E_k e_{18} - 2e_2 L_3 e_2 + 2e_2 L_3 A_k e1 + 2e_2 B_k L_3 e_3$$

$$+ 2e_2 L_3 C_k e_4 + 2e_2 L_3 E_k e_{18} + 2e_3 L_4 A_k e_2 - 2e_3 A_k L_4 e_1 - 2e_3 B_k L_4 e_3$$

$$- 2e_3 C_k L_4 e_4 - 2e_3 E_k L_4 e_{18} + 2e_4 L_5 e_2 - 2e_4 A_k L_5 e_1 - 2e_4 B_k L_5 e_3 - 2e_4 C_k L_5 e_4$$

$$- 2e_4 E_k L_5 e_{18} + 2e_{18} L_6 e_2 - 2e_{18} A_k L_6 e_1 - 2e_{18} B_k L_6 e_3 - 2e_{18} C_k L_6 e_{18}$$

$$- 2e_{18} E_k L_6 e_{18} + 2e_1 L_1 e_1],$$

$$\Xi_2^{**} \quad = [e_1 Q_1 e_1 - e^{-2\alpha\sigma_1} e_7 Q_1 e_7 + e_1 Q_2 e_1 - e^{-2\alpha\tau_1} e_5 Q_2 e_5 + e_1 R_1 e_1 - e^{-2\alpha\sigma_2} e_6 R_1 e_6$$

$$+ \sigma_d e^{-2\alpha\sigma_1} e_6 R_1 e_6 + e_2 R_2 e_2 - e^{-2\alpha\tau_2} e_4 R_2 e_4 + \tau_d e^{-2\alpha\tau_1} e_4 R_2 e_4,$$

$$\Xi_3^{**} \quad = \sigma_2^2 e_2 S_1 e_2 - e^{-2\alpha\sigma_2}[e_1 - e_5]^T S_1 [e_1 - e_5]$$

$$- 3e^{-2\alpha\sigma_2}\left[e_1 + e_5 - \frac{2}{\sigma_2}e_8\right]^T S_1\left[e_1 + e_5 - \frac{2}{\sigma_2}e_8\right]$$

$$- 5e^{-2\alpha\sigma_2}\left[e_1 - e_5 + \frac{6}{\sigma_2}e_8 - \frac{12}{\sigma_2^2}e_9\right]^T S_1\left[e_1 - e_5 + \frac{6}{\sigma_2}e_8 - \frac{12}{\sigma_2^2}e_9\right]$$

$$- 7e^{-2\sigma_2}\left[e_1 + e_5 - \frac{12}{\sigma_2}e_8 + \frac{60}{\sigma_2^2}e_9 - \frac{120}{\sigma_2^3}e_{10}\right]^T$$

$$\times S_1\left[e_1 + e_5 - \frac{12}{\sigma_2}e_8 + \frac{60}{\sigma_2^2}e_9 - \frac{120}{\sigma_2^3}e_{10}\right]$$

$$+ \tau_2^2 e_2 S_2 e_2 - e^{-2\alpha\tau_2}[e_1 - e_7]^T S_2 [e_1 - e_7]$$

$$- 3e^{-2\alpha\tau_2}\left[e_1 + e_7 - \frac{2}{\tau_2}e_{11}\right]^T S_2\left[e_1 + e_7 - \frac{2}{\tau_2}e_{11}\right]$$

$$- 5e^{-2\alpha\tau_2}\left[e_1 - e_7 + \frac{6}{\tau_2}e_{11} - \frac{12}{\tau_2^2}e_{12}\right]^T S_2\left[e_1 - e_7 + \frac{6}{\tau_2}e_{11} - \frac{12}{\tau_2^2}e_{12}\right]$$

$$- 7e^{-2\alpha\tau_2}\left[e_1 + e_7 - \frac{12}{\tau_2}e_{11} + \frac{60}{\tau_2^2}e_{12} - \frac{120}{\tau_2^3}e_{13}\right]^T$$

$$\times S_2\left[e_1 + e_7 - \frac{12}{\tau_2}e_{11} + \frac{60}{\tau_2^2}e_{12} - \frac{120}{\tau_2^3}e_{13}\right],$$

$$\Xi_4^{**} = [\tau_2^2 e_2 Z_1 e_2 - e^{-2\alpha\tau_2}e_{16}Z_1 e_{16} - e^{-2\alpha\tau_2}e_{17}Z_1 e_{17} + \sigma_2^2 e_2 Z_2 e_2 - e^{-2\alpha\sigma_2}e_{14}Z_2 e_{14}$$

$$- e^{-2\alpha\tau_2}e_{15}Z_2 e_{15}],$$

$$\Xi_5^{**} = \sigma_2^4 e_2 W_1 e_2 - 2e^{-2\alpha\sigma_2}\sigma_2^2\left[e_1 - \frac{1}{\sigma_2}e_8\right]W_1\left[e_1 - \frac{1}{\sigma_2}e_8\right]$$

$$- 4\sigma_2^2 e^{-2\alpha\sigma_2}\left[e_1 + \frac{2}{\sigma_2}e_8 - \frac{6}{\sigma_2^2}e_9\right]W_1\left[e_1 + \frac{2}{\sigma_2}e_8 - \frac{6}{\sigma_2^2}e_9\right]$$

$$- 6\sigma_2^2 e^{-2\alpha\sigma_2}\left[e_1 - \frac{3}{\sigma_2}e_8 + \frac{24}{\sigma_2^2}e_9 - \frac{60}{\sigma_2^3}e_{10}\right]W_1\left[e_1 - \frac{3}{\sigma_2}e_8 + \frac{24}{\sigma_2^2}e_9 - \frac{60}{\sigma_2^3}e_{10}\right]$$

$$+ \tau_2^4 e_2 W_2 e_2 - 2e^{-2\alpha\tau_2}\tau_2^2\left[e_1 - \frac{1}{\tau_2}e_{11}\right]W_2\left[e_1 - \frac{1}{\tau_2}e_{11}\right]$$

$$- 4\tau_2^2 e^{-2\alpha\tau_2}\left[e_1 + \frac{2}{\tau_2}e_{11} - \frac{6}{\tau_2^2}e_{12}\right]W_2\left[e_1 + \frac{2}{\tau_2}e_{11} - \frac{6}{\tau_2^2}e_{12}\right]$$

$$- 6\tau_2^2 e^{-2\alpha\tau_2}\left[e_1 - \frac{3}{\tau_2}e_{11} + \frac{24}{\tau_2^2}e_{12} - \frac{60}{\tau_2^3}e_{13}\right]W_2\left[e_1 - \frac{3}{\tau_2}e_{11} + \frac{24}{\tau_2^2}e_{12} - \frac{60}{\tau_2^3}e_{13}\right],$$

$$\Xi_6^{**} = \frac{\sigma_2^6}{6}e_2 U_1 e_2 - 6e^{-2\alpha\sigma_2}\left[\frac{\sigma_2^2}{2}e_1 - e_9\right]U_1\left[\frac{\sigma_2^2}{2}e_1 - e_9\right]$$

$$- 10e^{-2\alpha\sigma_2}\left[\frac{\sigma_2^2}{2}e_1 - e_9 + \frac{4}{\sigma_2}e_{10}\right]U_1\left[\frac{\sigma_2^2}{2}e_1 - e_9 + \frac{4}{\sigma_2}e_{10}\right]$$

$$+ \frac{\tau_2^6}{6}e_2 U_2 e_2 - 6e^{-2\alpha\tau_2}\left[\frac{\tau_2^2}{2}e_1 - e_{12}\right]U_2\left[\frac{\tau_2^2}{2}e_1 - e_{12}\right]$$

$$- 10e^{-2\alpha\tau_2}\left[\frac{\tau_2^2}{2}e_1 - e_{12} + \frac{4}{\tau_2}e_{13}\right]U_2\left[\frac{\tau_2^2}{2}e_1 - e_{12} + \frac{4}{\tau_2}e_{13}\right],$$

$$\Xi_7^{**} = [e_1 y_1 e_1 - e_3 y_1 e_3 - e_{14} y_1 e_{14} + e_3 y_2 e_3 - e_6 y_2 e_6 - e_{15} y_2 e_{15}],$$

$$\Xi_8^{**} = [\lambda e_1 H_{1i}^T H_{1i} e_1 + \lambda e_1 H_{1i}^T H_{2i} e_3 + \lambda e_1 H_{1i}^T H_{3i} e_4 + \lambda e_1 H_{1i}^T H_{4i} e_{18} + \lambda e_3 H_{2i}^T H_{2i} e_3$$

$$+ \lambda e_3 H_{2i}^T H_{3i} e_4 + \lambda e_3 H_{2i}^T H_{4i} e_{18} + \lambda e_4 H_{3i}^T H_{3i} e_4 + \lambda e_4 H_{3i}^T H_{4i} e_{18} + \lambda e_{18} H_{4i}^T H_{4i} e_{18}],$$

$$\Xi_9^{**} = -2e_1\tilde{A}e_{18} - 2e_3\tilde{B}e_{18} - 2e_{18}\tilde{E}e_{18},$$

$$\Phi_5 \quad = [L_1F + L_2F, L_3F, L_4F, L_5F, \overbrace{0, \cdots, 0}^{13 \quad times}, L_6F]^T,$$

$$\xi(t) \quad = [x(t), \dot{x}(t), x(t - \sigma(t)), \dot{x}(t - \tau(t)), x(t - \sigma_2), \dot{x}(t - \sigma(t)), x(t - \tau_2), \int_{t-\sigma_2}^{t} x(s)ds,$$

$$\int_{t-\sigma_2}^{t} \int_{s}^{t} x(u)duds, \int_{t-\sigma_2}^{t} \int_{s}^{t} \int_{u}^{t} x(v)dvduds, \int_{t-\tau_2}^{t} x(s)ds, \int_{t-\tau_2}^{t} \int_{s}^{t} x(u)duds,$$

$$\int_{t-\tau_2}^{t} \int_{s}^{t} \int_{u}^{t} x(v)dvduds, \int_{t-\sigma(t)}^{t-\sigma_1} \dot{x}(s)ds, \int_{t-\sigma_2}^{t-\sigma(t)} \dot{x}(s)ds, \int_{t-\tau(t)}^{t-\tau_1} \dot{x}(s)ds,$$

$$\int_{t-\tau_2}^{t-\tau(t)} \dot{x}(s)ds, u(t)]^T.$$

*proof* This study focal point in the following Lyapunov-Krasovskii function of the system (1) when $D_i + \Delta D_i(t) = 0$.

$$V(x(t)) = \sum_{i=1}^{6} V_i(x(t)),$$

*where*

$$V_1 \quad = x^T(t)L_1x(t),$$

$$V_2 \quad = \int_{t-\sigma_2}^{t} e^{2\alpha(s-t)}x^T(s)Q_1x(s)ds + \int_{t-\tau_2}^{t} e^{2\alpha(s-t)}x^T(s)Q_2x(s)ds$$

$$\qquad + \int_{t-\sigma_2}^{t} e^{2\alpha(s-t)}x^T(s)R_1x(s)ds + \int_{t-\tau_2}^{t} e^{2\alpha(s-t)}\dot{x}^T(s)R_2\dot{x}(s)ds$$

$$V_3 \quad = \sigma_2 \int_{-\sigma_2}^{0} \int_{t+\theta}^{t} e^{2\alpha(s-t)}\dot{x}^T(s)S_1\dot{x}(s)dsd\theta + \tau_2 \int_{-\tau_2}^{0} \int_{t+\theta}^{t} e^{2\alpha(s-t)}\dot{x}^T(s)S_2\dot{x}(s)dsd\theta,$$

$$V_4 \quad = \sigma_2 \int_{-\sigma_2}^{0} \int_{t+\theta}^{t} e^{2\alpha(s-t)}\dot{x}^T(s)Z_1\dot{x}(s)dsd\theta + \tau_2 \int_{-\tau_2}^{0} \int_{t+\theta}^{t} e^{2\alpha(s-t)}\dot{x}^T(s)Z_2\dot{x}(s)dsd\theta+,$$

$$V_5 \quad = \sigma_2^2 \int_{-\sigma_2}^{0} \int_{\theta}^{0} \int_{t+\beta}^{t} e^{2\alpha(s-t)}\dot{x}^T(s)W_1\dot{x}(s)dsd\beta d\theta$$

$$\qquad + \tau_2^2 \int_{-\tau_2}^{0} \int_{\theta}^{0} \int_{t+\beta}^{t} e^{2\alpha(s-t)}\dot{x}^T(s)W_2\dot{x}(s)dsd\beta d\theta,$$

$$V_6 \quad = \sigma_2^3 \int_{-\sigma_2}^{0} \int_{v}^{0} \int_{\theta}^{0} \int_{t+\beta}^{t} e^{2\alpha(s-t)}\dot{x}^T(s)U_1\dot{x}(s)dsd\beta d\theta dv$$

$$\qquad + \tau_2^3 \int_{-\tau_2}^{0} \int_{v}^{0} \int_{\theta}^{0} \int_{t+\beta}^{t} e^{2\alpha(s-t)}\dot{x}^T(s)U_2\dot{x}(s)dsd\beta d\theta dv.$$

Abovementioned by Theorem 1 and Theorem 2, this study attain the exponentially passive synthesis of delay-dependent condition for systems (1) when $D_i + \Delta D_i(t) = 0$.

Acquired from Corollary 3, the purpose of this study is for the consequences of uncertainty for T-S fuzzy system (1) when $D_i + \Delta D_i(t) = 0$ and $E_i + \Delta E_i(t) = 0$.

**Remark 5** *If $D_i + \Delta D_i(t) = 0$ and $E_i + \Delta E_i(t) = 0$, the uncertainty fuzzy replica (1) become the T-S fuzzy of neutral differential system presented by* [30].

**Corollary 3** *For given constants $\sigma_2$, $\tau_2 \geq 0$ systems (1) where $D_i + \Delta D_i(t) = 0$ and $E_i + \Delta E_i(t) = 0$ with uncertain is exponentially stability. If there are symmetric matrices $L_1$, $R_1$, $R_2$, $Q_1$, $Q_2$, $Z_1$, $Z_2$, $W_1$, $W_2$, $U_1$, $U_2 > 0$ and a positive $\lambda$ agreeable the LMI for $k = 1, 2, \ldots, m$ as ensuing:*

$$\Omega_6 = \begin{bmatrix} \psi_7 & \Phi_6 \\ * & -\lambda_6 I \end{bmatrix} < 0,$$

*where*

$$\psi_{7k} = \Sigma^{**},$$

$$\Sigma^{**} = \Xi_{1k}^{**} + \Xi_2^{**} + \Xi_3^{**} + \Xi_4^{**} + \Xi_5^{**} + \Xi_6^{**} + \Xi_7^{**}, + \Xi_8^{**},$$

$$\Xi_{1k}^{**} = [2e_1 L_1 e_1 + 2e_1 L_1 A_k e_3 + 2e_1 L_1 C_k e_4 + 2e_1 L_1 E_k e_{18} + 2e_1 L_2 e_2 - 2e_1 L_2 A_k e_1$$

$$- 2e_1 L_2 B_k e_3 - 2e_1 L_2 C_k e_4 - 2e_2 L_2 E_k e_{18} - 2e_2 L_3 e_2 + 2e_2 L_3 A_k e1 + 2e_2 B_k L_3 e_3$$

$$+ 2e_2 L_3 C_k e_4 + 2e_2 L_3 E_k e_{18} + 2e_3 L_4 A_k e_2 - 2e_3 A_k L_4 e_1 - 2e_3 B_k L_4 e_3$$

$$- 2e_3 C_k L_4 e_4 - 2e_3 E_k L_4 e_{18} + 2e_4 L_5 e_2 - 2e_4 A_k L_5 e_1 - 2e_4 B_k L_5 e_3 - 2e_4 C_k L_5 e_4$$

$$- 2e_4 E_k L_5 e_{18} + 2e_{18} L_6 e_2 - 2e_{18} A_k L_6 e_1 - 2e_{18} B_k L_6 e_3 - 2e_{18} C_k L_6 e_{18}$$

$$- 2e_{18} E_k L_6 e_{18} + 2e_1 L_1 e_1],$$

$$\Xi_2^{**} = [e_1 Q_1 e_1 - e^{-2\alpha\sigma_1} e_7 Q_1 e_7 + e_1 Q_2 e_1 - e^{-2\alpha\tau_1} e_5 Q_2 e_5 + e_1 R_1 e_1 - e^{-2\alpha\sigma_2} e_6 R_1 e_6$$

$$+ \sigma_d e^{-2\alpha\sigma_1} e_6 R_1 e_6 + e_2 R_2 e_2 - e^{-2\alpha\tau_2} e_4 R_2 e_4 + \tau_d e^{-2\alpha\tau_1} e_4 R_2 e_4,$$

$$\Xi_3^{**} = \sigma_2^2 e_2 S_1 e_2 - e^{-2\alpha\sigma_2} [e_1 - e_5]^T S_1 [e_1 - e_5]$$

$$- 3e^{-2\alpha\sigma_2} \left[e_1 + e_5 - \frac{2}{\sigma_2} e_8\right]^T S_1 \left[e_1 + e_5 - \frac{2}{\sigma_2} e_8\right]$$

$$- 5e^{-2\alpha\sigma_2} \left[e_1 - e_5 + \frac{6}{\sigma_2} e_8 - \frac{12}{\sigma_2^2} e_9\right]^T S_1 \left[e_1 - e_5 + \frac{6}{\sigma_2} e_8 - \frac{12}{\sigma_2^2} e_9\right]$$

$$- 7e^{-2\sigma_2} \left[ e_1 + e_5 - \frac{12}{\sigma_2} e_8 + \frac{60}{\sigma_2^2} e_9 - \frac{120}{\sigma_2^3} e_{10} \right]^T$$

$$\times S_1 \left[ e_1 + e_5 - \frac{12}{\sigma_2} e_8 + \frac{60}{\sigma_2^2} e_9 - \frac{120}{\sigma_2^3} e_{10} \right]$$

$$+ \tau_2^2 e_2 S_2 e_2 - e^{-2\alpha\tau_2} [e_1 - e_7]^T S_2 [e_1 - e_7]$$

$$- 3e^{-2\alpha\tau_2} \left[ e_1 + e_7 - \frac{2}{\tau_2} e_{11} \right]^T S_2 \left[ e_1 + e_7 - \frac{2}{\tau_2} e_{11} \right]$$

$$- 5e^{-2\alpha\tau_2} \left[ e_1 - e_7 + \frac{6}{\tau_2} e_{11} - \frac{12}{\tau_2^2} e_{12} \right]^T S_2 \left[ e_1 - e_7 + \frac{6}{\tau_2} e_{11} - \frac{12}{\tau_2^2} e_{12} \right]$$

$$- 7e^{-2\alpha\tau_2} \left[ e_1 + e_7 - \frac{12}{\tau_2} e_{11} + \frac{60}{\tau_2^2} e_{12} - \frac{120}{\tau_2^3} e_{13} \right]^T$$

$$\times S_2 \left[ e_1 + e_7 - \frac{12}{\tau_2} e_{11} + \frac{60}{\tau_2^2} e_{12} - \frac{120}{\tau_2^3} e_{13} \right],$$

$$\Xi_4^{**} = [\tau_2^2 e_2 Z_1 e_2 - e^{-2\alpha\tau_2} e_{16} Z_1 e_{16} - e^{-2\alpha\tau_2} e_{17} Z_1 e_{17} + \sigma_2^2 e_2 Z_2 e_2 - e^{-2\alpha\sigma_2} e_{14} Z_2 e_{14}$$

$$- e^{-2\alpha\tau_2} e_{15} Z_2 e_{15}],$$

$$\Xi_5^{**} = \sigma_2^4 e_2 W_1 e_2 - 2e^{-2\alpha\sigma_2} \sigma_2^2 \left[ e_1 - \frac{1}{\sigma_2} e_8 \right] W_1 \left[ e_1 - \frac{1}{\sigma_2} e_8 \right]$$

$$- 4\sigma_2^2 e^{-2\alpha\sigma_2} \left[ e_1 + \frac{2}{\sigma_2} e_8 - \frac{6}{\sigma_2^2} e_9 \right] W_1 \left[ e_1 + \frac{2}{\sigma_2} e_8 - \frac{6}{\sigma_2^2} e_9 \right]$$

$$- 6\sigma_2^2 e^{-2\alpha\sigma_2} \left[ e_1 - \frac{3}{\sigma_2} e_8 + \frac{24}{\sigma_2^2} e_9 - \frac{60}{\sigma_2^3} e_{10} \right] W_1 \left[ e_1 - \frac{3}{\sigma_2} e_8 + \frac{24}{\sigma_2^2} e_9 - \frac{60}{\sigma_2^3} e_{10} \right]$$

$$+ \tau_2^4 e_2 W_2 e_2 - 2e^{-2\alpha\tau_2} \tau_2^2 \left[ e_1 - \frac{1}{\tau_2} e_{11} \right] W_2 \left[ e_1 - \frac{1}{\tau_2} e_{11} \right]$$

$$- 4\tau_2^2 e^{-2\alpha\tau_2} \left[ e_1 + \frac{2}{\tau_2} e_{11} - \frac{6}{\tau_2^2} e_{12} \right] W_2 \left[ e_1 + \frac{2}{\tau_2} e_{11} - \frac{6}{\tau_2^2} e_{12} \right]$$

$$- 6\tau_2^2 e^{-2\alpha\tau_2} \left[ e_1 - \frac{3}{\tau_2} e_{11} + \frac{24}{\tau_2^2} e_{12} - \frac{60}{\tau_2^3} e_{13} \right] W_2 \left[ e_1 - \frac{3}{\tau_2} e_{11} + \frac{24}{\tau_2^2} e_{12} - \frac{60}{\tau_2^3} e_{13} \right],$$

$$\Xi_6^{**} = \frac{\sigma_2^6}{6} e_2 U_1 e_2 - 6e^{-2\alpha\sigma_2} \left[ \frac{\sigma_2^2}{2} e_1 - e_9 \right] U_1 \left[ \frac{\sigma_2^2}{2} e_1 - e_9 \right]$$

$$- 10e^{-2\alpha\sigma_2}\left[\frac{\sigma_2^2}{2}e_1 - e_9 + \frac{4}{\sigma_2}e_{10}\right]U_1\left[\frac{\sigma_2^2}{2}e_1 - e_9 + \frac{4}{\sigma_2}e_{10}\right]$$

$$+ \frac{\tau_2^6}{6}e_2 U_2 e_2 - 6e^{-2\alpha\tau_2}\left[\frac{\tau_2^2}{2}e_1 - e_{12}\right]U_2\left[\frac{\tau_2^2}{2}e_1 - e_{12}\right]$$

$$- 10e^{-2\alpha\tau_2}\left[\frac{\tau_2^2}{2}e_1 - e_{12} + \frac{4}{\tau_2}e_{13}\right]U_2\left[\frac{\tau_2^2}{2}e_1 - e_{12} + \frac{4}{\tau_2}e_{13}\right],$$

$$\Xi_7^{**} = [e_1 y_1 e_1 - e_3 y_1 e_3 - e_{14} y_1 e_{14} + e_3 y_2 e_3 - e_6 y_2 e_6 - e_{15} y_2 e_{15}],$$

$$\Xi_8^{**} = [\lambda e_1 H_{1i}^T H_{1i} e_1 + \lambda e_1 H_{1i}^T H_{2i} e_3 + \lambda e_1 H_{1i}^T H_{3i} e_4 + \lambda e_1 H_{1i}^T H_{4i} e_{18} + \lambda e_3 H_{2i}^T H_{2i} e_3$$

$$+ \lambda e_3 H_{2i}^T H_{3i} e_4 + \lambda e_3 H_{2i}^T H_{4i} e_{18} + \lambda e_4 H_{3i}^T H_{3i} e_4 + \lambda e_4 H_{3i}^T H_{4i} e_{18}$$

$$+ \lambda e_{18} H_{4i}^T H_{4i} e_{18}],$$

$$\Phi_6 = [L_1 F + L_2 F, L_3 F, L_4 F, \overbrace{0, \cdots, 0}^{13 \quad times}, L_5 F]^T,$$

$$\xi(t) = [x(t), \dot{x}(t), x(t - \sigma(t)), \dot{x}(t - \tau(t)), x(t - \sigma_2), \dot{x}(t - \sigma(t)), x(t - \tau_2), \int_{t-\sigma_2}^{t} x(s)ds,$$

$$\int_{t-\sigma_2}^{t}\int_{s}^{t} x(u)duds, \int_{t-\sigma_2}^{t}\int_{s}^{t}\int_{u}^{t} x(v)dvduds, \int_{t-\tau_2}^{t} x(s)ds, \int_{t-\tau_2}^{t}\int_{s}^{t} x(u)duds,$$

$$\int_{t-\tau_2}^{t}\int_{s}^{t}\int_{u}^{t} x(v)dvduds, \int_{t-\sigma(t)}^{t-\sigma_1} \dot{x}(s)ds, \int_{t-\sigma_2}^{t-\sigma(t)} \dot{x}(s)ds, \int_{t-\tau(t)}^{t-\tau_1} \dot{x}(s)ds,$$

$$\int_{t-\tau_2}^{t-\tau(t)} \dot{x}(s)ds]^T.$$

Then the system (1) when $D_i + \Delta D_i(t) = 0$ and $E_i + \Delta E_i(t) = 0$ is exponentially stability.

**Remark 6** *According to Corollary 3 that using Lemmas 2, 3 and Lemma 4 yielded fewer conservative outcomes than other results, [30] which illustrate in* Table 6. *Even, these lemmas contain a large number of free weighting matrices, that could bring about their more calculation intricately.*

## 4 Numerical simulation

In this part, the number of sample figures illustrate the performance of our key solution, by comparison of the largest allowable bound $\sigma$ and the convergent rate $\alpha$. The LMI control toolbox in MATLAB is used to find all the threshold possibilities.

**Example 1** *Analyze the uncertainty neutral of T-S fuzzy dynamic system by the parameters as following:*

$$
\begin{cases}
\dot{x}(t) &= Ax(t) + Bx(t - \sigma(t)) + C(\dot{x} - \tau(t)) + D \int_{t-h(t)}^{t} x(s)ds + Eu(t) \\
z(t) &= \tilde{A}x(t) + \tilde{B}x(t - \sigma(t)) + \tilde{E}u(t) \\
x(t) &= \phi(t), t \in [-n, 0], n = \max\{\tau_2, \sigma_2, h_2\},
\end{cases}
$$

*where*

$$
A_1 = \begin{bmatrix} -5 & -0.2 \\ -0.1 & -0.4 \end{bmatrix}, \quad A_2 = \begin{bmatrix} -3 & -0.1 \\ -0.1 & -0.5 \end{bmatrix}, \quad B_1 = \begin{bmatrix} 0.5 & 0.7 \\ 0.7 & 0.4 \end{bmatrix}, \quad B_2 = \begin{bmatrix} 0.5 & 0.2 \\ 0.1 & 0.2 \end{bmatrix}
$$

$$
C_1 = \begin{bmatrix} 1 & -0.4 \\ -0.3 & -0.1 \end{bmatrix}, \quad C_2 = \begin{bmatrix} 0.07 & 0.4 \\ 0.1 & 0.1 \end{bmatrix}, \quad D_1 = \begin{bmatrix} 0.5 & 0.2 \\ 0.3 & 0.4 \end{bmatrix}, \quad D_2 = \begin{bmatrix} -0.5 & -0.2 \\ 0.8 & 0.2 \end{bmatrix},
$$

$$
E_1 = \begin{bmatrix} -0.9 & 0.2 \\ 0.9 & -0.9 \end{bmatrix}, \quad E_2 = \begin{bmatrix} 0.1 & -0.2 \\ 0.1 & 1.1 \end{bmatrix}, \quad \tilde{A}_1 = \begin{bmatrix} -2 & 0.1 \\ 0.2 & 0.9 \end{bmatrix}, \quad \tilde{A}_2 = \begin{bmatrix} -2 & 0.1 \\ 0.3 & 0.5 \end{bmatrix},
$$

$$
\tilde{B}_1 = \begin{bmatrix} 1 & 0.2 \\ 0.1 & 0.5 \end{bmatrix} \quad \tilde{B}_2 = \begin{bmatrix} 0.5 & 0.2 \\ 0.1 & 0.3 \end{bmatrix} \quad \tilde{E}_1 = \begin{bmatrix} 2 & 0.3 \\ 0.1 & 0.8 \end{bmatrix}, \quad \tilde{E}_2 = \begin{bmatrix} 1 & 0.3 \\ 0.2 & 0.8 \end{bmatrix},
$$

$$
H_{11} = \begin{bmatrix} 1.6 & 0 \\ 0 & 0.05 \end{bmatrix}, \quad H_{12} = \begin{bmatrix} 1.6 & 0 \\ 0 & -0.05 \end{bmatrix}, \quad H_{21} = \begin{bmatrix} 0.1 & 0 \\ 0 & 0.3 \end{bmatrix}, \quad H_{22} = \begin{bmatrix} 0.1 & 0 \\ 0 & 0.3 \end{bmatrix},
$$

$$
H_{31} = \begin{bmatrix} 0.2 & -0.1 \\ 0.1 & 0.2 \end{bmatrix}, \quad H_{32} = \begin{bmatrix} 0.1 & 0.2 \\ 0.2 & 0.1 \end{bmatrix}, \quad H_{41} = \begin{bmatrix} 0.1 & 0.1 \\ -0.2 & -0.1 \end{bmatrix}, \quad H_{42} = \begin{bmatrix} -0.1 & 0.2 \\ 0.1 & 0.1 \end{bmatrix},
$$

$$
H_{51} = \begin{bmatrix} 1 & -0.6 \\ 0.5 & 0.2 \end{bmatrix}, \quad H_{52} = \begin{bmatrix} -1 & 1 \\ -0.5 & 0.4 \end{bmatrix}, \quad F = \begin{bmatrix} 0.03 & 0 \\ 0 & -0.03 \end{bmatrix} \quad I = \begin{bmatrix} 1 & 0 \\ 0 & 1 \end{bmatrix},
$$

*LMI* (2), *is solved where* $\alpha = 0.2, \sigma(t) = 0.2 + \frac{\sin(t)}{10}, \tau(t) = 0.2 + \frac{\sin(t)}{10}$ *and* $h(t) = 0.1 + \frac{\sin(t)}{10}$ *to*

*obtain set of parameters for guarantee the exponentially passive as following:*

$$L_1 = \begin{bmatrix} 0.0897 & -0.0127 \\ -0.0127 & 0.1163 \end{bmatrix} \times 10^{-7}, \qquad L_2 = \begin{bmatrix} 0.0650 & -0.0334 \\ -0.0334 & 0.1560 \end{bmatrix} \times 10^{-7},$$

$$L_3 = \begin{bmatrix} 0.1043 & -0.1481 \\ -0.1481 & 0.4701 \end{bmatrix} \times 10^{-8}, \qquad L_4 = \begin{bmatrix} 0.0822 & -0.0452 \\ -0.0452 & 0.2172 \end{bmatrix} \times 10^{-7},$$

$$L_5 = \begin{bmatrix} 0.3222 & 0.0547 \\ 0.0547 & 0.3604 \end{bmatrix} \times 10^{-9}, \qquad L_6 = \begin{bmatrix} 0.0509 & -0.0196 \\ -0.0196 & 0.1115 \end{bmatrix} \times 10^{-8},$$

$$L_7 = \begin{bmatrix} 0.1741 & -0.1352 \\ -0.1352 & 0.4321 \end{bmatrix} \times 10^{-8}, \qquad Q_1 = \begin{bmatrix} 0.2272 & 0.0125 \\ 0.0125 & 0.2129 \end{bmatrix} \times 10^{-5},$$

$$Q_2 = \begin{bmatrix} 0.2023 & -0.0119 \\ -0.0119 & 0.1880 \end{bmatrix} \times 10^{-5}, \qquad Q_3 = \begin{bmatrix} 0.1756 & 0.0083 \\ 0.0083 & 0.1653 \end{bmatrix} \times 10^{-5},$$

$$R_1 = \begin{bmatrix} 0.2383 & 0.0124 \\ 0.0124 & 0.2240 \end{bmatrix} \times 10^{-5}, \qquad R_2 = \begin{bmatrix} 0.2379 & 0.0122 \\ 0.0122 & 0.2224 \end{bmatrix} \times 10^{-5},$$

$$R_3 = \begin{bmatrix} 0.2290 & 0.114 \\ 0.0114 & 0.2149 \end{bmatrix} \times 10^{-5}, \qquad R_4 = \begin{bmatrix} 0.1939 & 0.0763 \\ 0.0763 & 0.1325 \end{bmatrix} \times 10^{-8},$$

$$S_1 = \begin{bmatrix} 0.6322 & -0.0078 \\ -0.0078 & 0.6399 \end{bmatrix} \times 10^{-9}, \qquad S_2 = \begin{bmatrix} 0.5753 & -0.0103 \\ -0.0103 & 0.5877 \end{bmatrix} \times 10^{-9},$$

$$S_3 = \begin{bmatrix} 0.3795 & -0.0028 \\ -0.0028 & 0.3825 \end{bmatrix} \times 10^{-9}, \qquad S_4 = \begin{bmatrix} 0.5099 & -0.0142 \\ -0.0142 & 0.5235 \end{bmatrix} \times 10^{-9},$$

$$Z_1 = \begin{bmatrix} 0.1157 & -0.0104 \\ -0.0104 & 0.1241 \end{bmatrix} \times 10^{-6}, \qquad Z_2 = \begin{bmatrix} 0.7200 & -0.0825 \\ -0.0825 & 0.7842 \end{bmatrix} \times 10^{-7},$$

$$Z_3 = \begin{bmatrix} 0.3602 & -0.0385 \\ -0.0385 & 0.3903 \end{bmatrix} \times 10^{-8}, \qquad W_1 = \begin{bmatrix} 0.1306 & -0.0122 \\ -0.0122 & 0.1417 \end{bmatrix} \times 10^{-8},$$

$$W_2 = \begin{bmatrix} 0.1754 & -0.0125 \\ -0.0125 & 0.1883 \end{bmatrix} \times 10^{-8}, \qquad W_3 = \begin{bmatrix} 0.1856 & 0.0335 \\ 0.0335 & 0.1628 \end{bmatrix} \times 10^{-7},$$

$$W_4 = \begin{bmatrix} 0.4204 & 0.0002 \\ 0.0002 & 0.4298 \end{bmatrix} \times 10^{-8}, \qquad U_1 = \begin{bmatrix} 0.1274 & 0.0137 \\ 0.0137 & 0.1156 \end{bmatrix} \times 10^{-5},$$

$$U_2 = \begin{bmatrix} 0.5693 & -0.0042 \\ -0.0042 & 0.5752 \end{bmatrix} \times 10^{-6}, \qquad U_3 = \begin{bmatrix} 0.2362 & 0.0313 \\ 0.0313 & 0.2143 \end{bmatrix} \times 10^{-5},$$

$$U_4 = \begin{bmatrix} 0.6208 & -0.0036 \\ -0.0036 & 0.6282 \end{bmatrix} \times 10^{-6}, \qquad \lambda = 1.7449 \times 10^{-8}.$$

*In this example, we used to discuss the exponentially passive of the T-S fuzzy for neutral differential system (1). For dissimilar values α, $\tau_d$, $\sigma_d$ in example 1 that are shown in* Table 1, *the*

**Table 1. The maximum allowable bounds of $\sigma_2$ with Example 1.**

| $\tau_d = \sigma_d$ | $\alpha = 0.5$ | $\alpha = 0.4$ | $\alpha = 0.3$ | $\alpha = 0.2$ | $\alpha = 0.1$ | $\alpha = 0$ |
|---|---|---|---|---|---|---|
| 0.5 | 18.5874 | 23.4512 | 25.8231 | 30.5935 | 48.3897 | 51.1252 |

maximum allowable bounds of $\sigma_2$ are got by solving the LMIs in Theorem 1 and Theorem 2 with the MATLAB control toolbox.

**Example 2** *Analyze the uncertainty neutral of T-S fuzzy dynamic system by the parameters as following*

$$\begin{cases} \dot{x}(t) & = Ax(t) + Bx(t - \sigma(t)) + C(\dot{x} - \tau(t)) + D \int_{t-h(t)}^{t} x(s)ds \\ x(t) & = \phi(t), t \in [-n, 0], n = \max\{\tau_2, \sigma_2, h_2\}, \end{cases}$$

*where*

$$A_1 = \begin{bmatrix} -5 & -0.2 \\ -0.1 & -0.4 \end{bmatrix}, \quad A_2 = \begin{bmatrix} -3 & -0.1 \\ -0.1 & -0.5 \end{bmatrix}, \quad B_1 = \begin{bmatrix} 0.5 & 0.7 \\ 0.7 & 0.4 \end{bmatrix}, \quad B_2 = \begin{bmatrix} 0.5 & 0.2 \\ 0.1 & 0.2 \end{bmatrix}$$

$$C_1 = \begin{bmatrix} 1 & 0.4 \\ 0.3 & 0.1 \end{bmatrix}, \quad C_2 = \begin{bmatrix} 0.07 & 0.4 \\ 0.1 & 0.1 \end{bmatrix}, \quad D_1 = \begin{bmatrix} 0.5 & 0.2 \\ 0.3 & 0.4 \end{bmatrix}, \quad D_2 = \begin{bmatrix} -0.5 & -0.2 \\ 0.8 & -0.2 \end{bmatrix},$$

$$H_{11} = \begin{bmatrix} 1.6 & 0 \\ 0 & 0.05 \end{bmatrix}, \quad H_{12} = \begin{bmatrix} 1.6 & 0 \\ 0 & -0.05 \end{bmatrix}, \quad H_{21} = \begin{bmatrix} 0.1 & 0 \\ 0 & 0.3 \end{bmatrix}, \quad H_{22} = \begin{bmatrix} 0.1 & 0 \\ 0 & 0.3 \end{bmatrix},$$

$$H_{31} = \begin{bmatrix} 0.2 & -0.1 \\ 0.1 & 0.2 \end{bmatrix}, \quad H_{32} = \begin{bmatrix} 0.1 & 0.2 \\ 0.2 & 0.1 \end{bmatrix}, \quad H_{41} = \begin{bmatrix} 0.1 & 0.1 \\ -0.2 & -0.1 \end{bmatrix},$$

$$H_{42} = \begin{bmatrix} -0.1 & 0.2 \\ 0.1 & 0.1 \end{bmatrix}, \quad F = \begin{bmatrix} 0.03 & 0 \\ 0 & -0.03 \end{bmatrix}, \quad I = \begin{bmatrix} 1 & 0 \\ 0 & 1 \end{bmatrix}.$$

*LMI* (12), *is solved where* $\alpha = 0.2, \sigma(t) = 0.2 + \frac{\sin(t)}{10}, \tau(t) = 0.2 + \frac{\sin(t)}{10}$ *and* $h(t) = 0.1 + \frac{\sin(t)}{10}$ *to obtain set of parameters for guarantee the exponentially as following*:

$$L_1 = \begin{bmatrix} 0.1918 & -0.0571 \\ -0.0571 & 0.2326 \end{bmatrix} \times 10^{-7}, \quad L_2 = \begin{bmatrix} 2.0294 & -0.5347 \\ -0.5347 & 0.9217 \end{bmatrix} \times 10^{-6},$$

$$L_3 = \begin{bmatrix} 0.0633 & -0.0919 \\ -0.091 & 0.2880 \end{bmatrix} \times 10^{-7}, \quad L_4 = \begin{bmatrix} 0.0390 & -0.0218 \\ -0.0218 & 0.1044 \end{bmatrix} \times 10^{-6},$$

$$L_5 = \begin{bmatrix} 0.1739 & 0.0388 \\ 0.0388 & 0.1895 \end{bmatrix} \times 10^{-8}, \quad L_6 = \begin{bmatrix} 0.2907 & -0.1289 \\ -0.1289 & 0.6682 \end{bmatrix} \times 10^{-8},$$

$$L_7 = \begin{bmatrix} 0.3232 & 0 \\ 0 & 0.3232 \end{bmatrix} \times 10^{-3}, \quad Q_1 = \begin{bmatrix} 0.6818 & -0.0055 \\ -0.0055 & 0.6818 \end{bmatrix} \times 10^{-5},$$

$$Q_2 = \begin{bmatrix} 0.8909 & -0.0071 \\ -0.0071 & 0.8910 \end{bmatrix} \times 10^{-5}, \quad Q_3 = \begin{bmatrix} 0.6802 & -0.0055 \\ -0.0055 & 0.6802 \end{bmatrix} \times 10^{-5},$$

$$R_1 = \begin{bmatrix} 0.6829 & -0.0056 \\ -0.0056 & 0.6829 \end{bmatrix} \times 10^{-5}, \quad R_2 = \begin{bmatrix} 0.1034 & -0.0013 \\ -0.0013 & 0.1030 \end{bmatrix} \times 10^{-4},$$

$$R_3 = \begin{bmatrix} 0.8974 & -0.0073 \\ -0.0073 & 0.8973 \end{bmatrix} \times 10^{-5}, \quad R_4 = \begin{bmatrix} 0.1151 & 0.0506 \\ 0.0506 & 0.0792 \end{bmatrix} \times 10^{-7},$$

$$S_1 = \begin{bmatrix} 0.2649 & -0.0067 \\ -0.0067 & 0.2714 \end{bmatrix} \times 10^{-7}, \quad S_2 = \begin{bmatrix} 0.1998 & -0.0040 \\ -0.0040 & 0.2024 \end{bmatrix} \times 10^{-7},$$

$$S_3 = \begin{bmatrix} 0.2518 & -0.0063 \\ -0.0063 & 0.2577 \end{bmatrix} \times 10^{-7}, \qquad S_4 = \begin{bmatrix} 0.1999 & -0.0040 \\ -0.0040 & 0.2025 \end{bmatrix} \times 10^{-7},$$

$$Z_1 = \begin{bmatrix} 0.3818 & -0.0067 \\ -0.0067 & 0.3848 \end{bmatrix} \times 10^{-6}, \qquad Z_2 = \begin{bmatrix} 0.2147 & -0.0052 \\ -0.0052 & 0.2191 \end{bmatrix} \times 10^{-6},$$

$$Z_3 = \begin{bmatrix} 0.1914 & -0.0037 \\ -0.0037 & 0.1938 \end{bmatrix} \times 10^{-7}, \qquad W_1 = \begin{bmatrix} 0.2418 & -0.0022 \\ -0.0022 & 0.2441 \end{bmatrix} \times 10^{-7},$$

$$W_2 = \begin{bmatrix} 0.4430 & 0.0003 \\ 0.0003 & 0.4457 \end{bmatrix} \times 10^{-7}, \qquad W_3 = \begin{bmatrix} 0.4010 & 0.0043 \\ 0.0043 & 0.4024 \end{bmatrix} \times 10^{-7},$$

$$W_4 = \begin{bmatrix} 0.4712 & 0.0013 \\ 0.0013 & 0.4737 \end{bmatrix} \times 10^{-7}, \qquad U_1 = \begin{bmatrix} 0.2643 & -0.0025 \\ -0.0025 & 0.2664 \end{bmatrix} \times 10^{-5},$$

$$U_2 = \begin{bmatrix} 0.5309 & 0.0002 \\ 0.0002 & 0.5326 \end{bmatrix} \times 10^{-5}, \qquad U_3 = \begin{bmatrix} 0.5016 & 0.0813 \\ 0.0813 & 0.4409 \end{bmatrix} \times 10^{-4},$$

$$U_4 = \begin{bmatrix} 0.6268 & 0.0023 \\ 0.0023 & 0.6276 \end{bmatrix} \times 10^{-5}, \qquad \lambda = 8.9650 \times 10^{-9}.$$

*In this example, we used to discuss the exponential stability criteria of the T-S fuzzy for neutral differential system* (1). *For dissimilar values $\alpha$, $\tau_d$, $\sigma_d$ in example 2 that are shown in* Table 2, *the maximum allowable bounds of $\sigma_2$ are got by solving the LMIs in Theorem 1 with the MATLAB control toolbox.*

**Example 3** *Analyze the uncertainty of T-S fuzzy dynamic system presented in* [7, 26–29] *by the parameters as following*

$$\begin{cases} \dot{x}(t) &= Ax(t) + Bx(t - \sigma(t)) \\ x(t) &= \phi(t), t \in [-n, 0], n = \max\{\sigma_2\}, \end{cases}$$

*where*

$$A_1 = \begin{bmatrix} -2 & 1 \\ 0.5 & -1 \end{bmatrix}, \quad A_2 = \begin{bmatrix} -2 & 0 \\ 0 & -1 \end{bmatrix}, \quad B_1 = \begin{bmatrix} -1 & 0 \\ -1 & -1 \end{bmatrix}, \quad B_2 = \begin{bmatrix} -1.6 & 0 \\ 0 & -1 \end{bmatrix},$$

$$H_{11} = \begin{bmatrix} 1.6 & 0 \\ 0 & 0.05 \end{bmatrix}, \quad H_{12} = \begin{bmatrix} 1.6 & 0 \\ 0 & -0.05 \end{bmatrix}, \quad H_{21} = \begin{bmatrix} 0.1 & 0 \\ 0 & 0.3 \end{bmatrix}, \quad H_{22} = \begin{bmatrix} 0.1 & 0 \\ 0 & 0.3 \end{bmatrix},$$

$$F = \begin{bmatrix} 0.03 & 0 \\ 0 & -0.03 \end{bmatrix}, \quad I = \begin{bmatrix} 1 & 0 \\ 0 & 1 \end{bmatrix},$$

$$\rho_1(\theta(t)) = \left(1 - \frac{1}{1 + exp(-3(x_2/0.5 - \pi/2))}\right)\left(\frac{1}{1 + exp(-3(x_2/0.5 - \pi/2))}\right),$$

$$\rho_2(\theta(t)) = 1 - \rho_1(\theta(t)).$$

*The purpose of example 3 is compare the maximum allowable bounds for tolerable delays of $\sigma(t)$ which ensure the exponential stability with the fuzzy convergent rate $\alpha$ of the T-S fuzzy dynamic*

**Table 2. The maximum allowable bounds of $\sigma_2$ with Example 2.**

| $\tau_d = \sigma_d$ | $\alpha = 0.5$ | $\alpha = 0.4$ | $\alpha = 0.3$ | $\alpha = 0.2$ | $\alpha = 0.1$ | $\alpha = 0$ |
|---|---|---|---|---|---|---|
| 0.5 | 31.9513 | 35.8276 | 37.4216 | 42.5164 | 47.5689 | 55.7421 |

**Table 3. The maximum allowable bounds of $\sigma_2$ for $\sigma_d$ and $\alpha$ of Example 3.**

| $\sigma_d$ | $\alpha$ | Li [27] | Lien [28] | Lien [7] | Liu [26] | Pin [29] | Corollary 2 |
|---|---|---|---|---|---|---|---|
| 0.5 | 0 | 0.637 | 0.929 | 0.934 | 1.147 | 1.1841 | 3.5687 |
| 0.5 | 0.5 | — | — | — | — | 0.7225 | 1.5416 |
| 0.7 | 0.7 | — | — | — | — | 0.6471 | 1.2948 |
| 0.9 | 0.9 | — | — | — | — | 0.5885 | 1.1121 |

system above. Based on Table 3, the results present-day available for comparison purposes are recorded. This present the proposed method is less conservative than the immemorial method.

Fig 1 gives the state trajectory of the T-S fuzzy dynamical system (1) where $C_i + \Delta C_i(t) = 0$ $D_i + \Delta D_i(t) = 0$ and $E_i + \Delta E_i(t) = 0$ with parameters in Example 3 where $u(t) = 0$ and the initial condition $[x_1(t), x_2(t)]^T = [-0.1\cos(t), 0.1\cos(t)]^T$, which shows that the T-S fuzzy for dynamical system is stable

**Example 4** *Analyze the uncertainty neutral of T-S fuzzy dynamic system by the parameters as following*:

$$\begin{cases} \dot{x}(t) & = Ax(t) + Bx(t - \sigma(t)) + Eu(t) \\ z(t) & = \tilde{A}x(t) + \tilde{B}x(t - \sigma(t)) + \tilde{E}u(t) \\ x(t) & = \phi(t), t \in [-n, 0], n = \max\{\sigma_2\}, \end{cases}$$

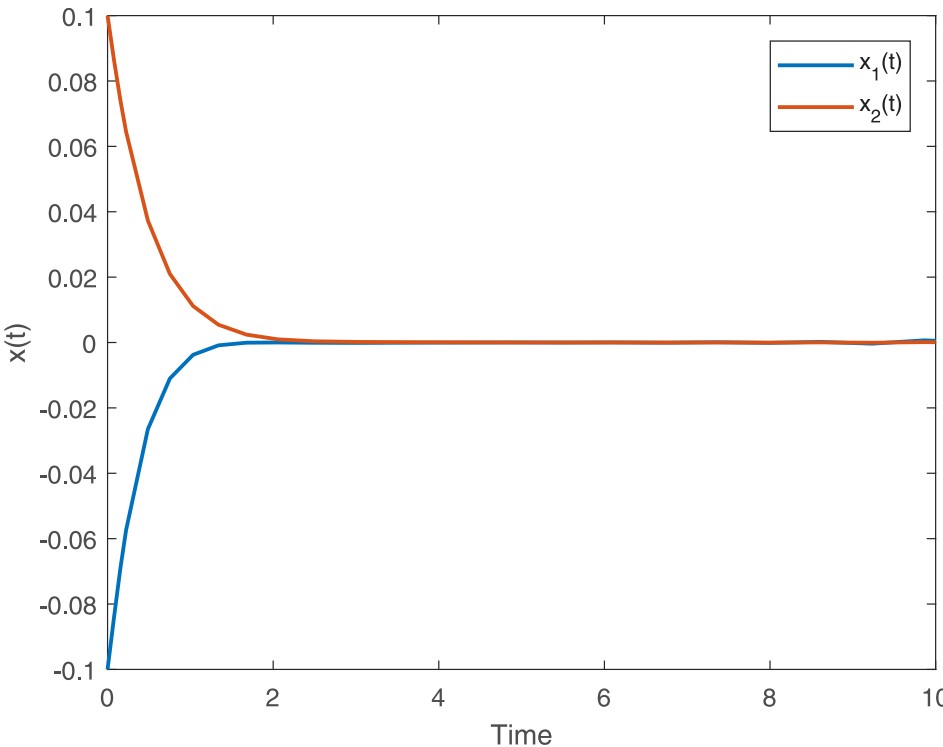

**Fig 1. State trajectory of T-S fuzzy dynamical system for Example 3.**

*Where*

$$A_1 = \begin{bmatrix} -2 & 0 \\ 0 & -0.9 \end{bmatrix}, \quad A_2 = \begin{bmatrix} -1 & 0.5 \\ 0 & -1 \end{bmatrix}, \quad B_1 = \begin{bmatrix} -1 & 0 \\ -1 & -1 \end{bmatrix}, \quad B_2 = \begin{bmatrix} -1 & 0 \\ 0.1 & -1 \end{bmatrix},$$

$$E_1 = \begin{bmatrix} 3 & 0 \\ 0.2 & -1 \end{bmatrix}, \quad E_2 = \begin{bmatrix} 1 & 0 \\ 0.2 & -2 \end{bmatrix}, \quad \tilde{A}_1 = \begin{bmatrix} -2 & 0.1 \\ 0.2 & 0.9 \end{bmatrix}, \quad \tilde{A}_2 = \begin{bmatrix} -2 & 0.1 \\ 0.1 & 0.8 \end{bmatrix},$$

$$\tilde{B}_1 = \begin{bmatrix} 1 & 0.2 \\ 0.1 & 0.5 \end{bmatrix}, \quad \tilde{B}_2 = \begin{bmatrix} 1 & 0.2 \\ 0.3 & 0.3 \end{bmatrix}, \quad \tilde{E}_1 = \begin{bmatrix} 2 & 0.3 \\ 0.1 & 0.8 \end{bmatrix}, \quad \tilde{E}_2 = \begin{bmatrix} 2 & 0.2 \\ 0.1 & 0.5 \end{bmatrix},$$

$$H_{11} = \begin{bmatrix} 1.6 & 0 \\ 0 & 0.05 \end{bmatrix}, \quad H_{12} = \begin{bmatrix} 1.6 & 0 \\ 0 & -0.05 \end{bmatrix}, \quad H_{21} = \begin{bmatrix} 0.1 & 0 \\ 0 & 0.3 \end{bmatrix}, \quad H_{22} = \begin{bmatrix} 0.1 & 0 \\ 0 & 0.3 \end{bmatrix},$$

$$H_{31} = \begin{bmatrix} 0.5 & 0 \\ 0 & 0.2 \end{bmatrix}, \quad H_{32} = \begin{bmatrix} 0.4 & 0 \\ 0 & 0.3 \end{bmatrix}, \quad F = \begin{bmatrix} 0.03 & 0 \\ 0 & -0.03 \end{bmatrix}, \quad I = \begin{bmatrix} 1 & 0 \\ 0 & 1 \end{bmatrix},$$

$$\rho_1(\theta(t)) = \frac{1}{1 + exp(-2x_1(t))}, \quad \rho_2(\theta(t)) = 1 - \rho_1(\theta(t)).$$

*LMI* (13) *is solved where* $\alpha = 0.2, \sigma(t) = 0.05 + \frac{\cos(t)}{10}$.

In example 4, we used to discuss the stability criterion and passivity performance of the T-S fuzzy for dynamical system (1) where $C_i + \Delta C_i(t) = 0$ and $D_i + \Delta D_i(t) = 0$. LMIs in Theorem 3 is solved by the MATLAB control toolbox to obtain the largest allowable bounds of $\sigma_2$ for dissimilar values of $\sigma_d, \alpha$ in example 4 are shown in Table 4.

Fig 2 gives the state trajectory of the T-S fuzzy for dynamical system (1) where $C_i + \Delta C_i(t) = 0$ and $D_i + \Delta D_i(t) = 0$ with parameter in Example 4 where the initial condition $[x_1(t), x_2(t)]^T = [-0.1 \cos(t), 0.1 \cos(t)]^T$, which shows that the T-S fuzzy for dynamical system is stable.

**Example 5** *Analyze the uncertainty neutral of T-S fuzzy dynamic system by the parameters as ollowing*:

$$\begin{cases} \dot{x}(t) &= Ax(t) + Bx(t - \sigma(t)) + C(\dot{x} - \tau(t)) + Eu(t) \\ z(t) &= \tilde{A}x(t) + \tilde{B}x(t - \sigma(t)) + \tilde{E}u(t) \\ x(t) &= \phi(t), t \in [-n, 0], n = \max\{\tau_2, \sigma_2\}, \end{cases}$$

*where*

$$A_1 = \begin{bmatrix} -0.9 & 0.2 \\ 0.1 & -0.9 \end{bmatrix}, \quad A_2 = \begin{bmatrix} -1 & 1 \\ 1.5 & -2 \end{bmatrix}, \quad B_1 = \begin{bmatrix} -1.1 & -0.2 \\ 0.1 & -1.1 \end{bmatrix}, \quad B_2 = \begin{bmatrix} -1 & -0.6 \\ 0.5 & -1.2 \end{bmatrix},$$

$$C_1 = \begin{bmatrix} -0.2 & 0 \\ 0.2 & -0.1 \end{bmatrix}, \quad C_2 = \begin{bmatrix} 0.2 & 0.1 \\ -0.4 & 0.8 \end{bmatrix}, \quad E_1 = \begin{bmatrix} 0.1 & 0.2 \\ -0.1 & 0.3 \end{bmatrix} \quad E_2 = \begin{bmatrix} 0.2 & 0.3 \\ -0.3 & 0.1 \end{bmatrix}$$

$$\tilde{A}_1 = \begin{bmatrix} -2 & 0.1 \\ 0.2 & 0.6 \end{bmatrix}, \quad \tilde{A}_2 = \begin{bmatrix} -2 & 0.1 \\ 0.3 & 0.5 \end{bmatrix}, \quad \tilde{B}_1 = \begin{bmatrix} 1 & 0.2 \\ 0.1 & 0.5 \end{bmatrix}, \quad \tilde{B}_2 = \begin{bmatrix} 0.5 & 0.2 \\ 0.1 & 0.3 \end{bmatrix},$$

$$\tilde{E}_1 = \begin{bmatrix} 2 & 0.3 \\ 0.1 & 0.8 \end{bmatrix}, \quad \tilde{E}_2 = \begin{bmatrix} 1 & 0.3 \\ 0.2 & 0.8 \end{bmatrix}, \quad H_{11} = \begin{bmatrix} 1.6 & 0 \\ 0 & 0.05 \end{bmatrix}, \quad H_{12} = \begin{bmatrix} 1.6 & 0 \\ 0 & -0.05 \end{bmatrix},$$

$$H_{21} = \begin{bmatrix} 0.1 & 0 \\ 0 & 0.3 \end{bmatrix}, \quad H_{22} = \begin{bmatrix} 0.1 & 0 \\ 0 & 0.3 \end{bmatrix}, \quad H_{31} = \begin{bmatrix} 0.5 & 0 \\ 0 & 0.2 \end{bmatrix}, \quad H_{32} = \begin{bmatrix} 0.4 & 0 \\ 0 & 0.3 \end{bmatrix},$$

$$H_{41} = \begin{bmatrix} 0.1 & 0 \\ 0 & 1 \end{bmatrix}, \quad H_{42} = \begin{bmatrix} 0.1 & 0 \\ 0 & 0.2 \end{bmatrix}, \quad F = \begin{bmatrix} 0.1459 & 0 \\ 0 & 0.1459 \end{bmatrix}, \quad I = \begin{bmatrix} 1 & 0 \\ 0 & 1 \end{bmatrix}.$$

*LMI* (14) *is solved where* $\alpha = 0.1, \sigma(t) = \frac{\cos(t)}{10}$ and $\tau(t) = \frac{\sin(t)}{10}$.

**Table 4. The maximum allowable bounds of $\sigma_2$ for $\sigma_d$ and $\alpha$ of Example 4.**

| $\sigma_d$ | $\alpha = 0.5$ | $\alpha = 0.4$ | $\alpha = 0.3$ | $\alpha = 0.2$ | $\alpha = 0.1$ | $\alpha = 0$ |
|---|---|---|---|---|---|---|
| 0.5 | 0.1998 | 0.2029 | 0.2061 | 0.2094 | 0.2128 | 0.2165 |

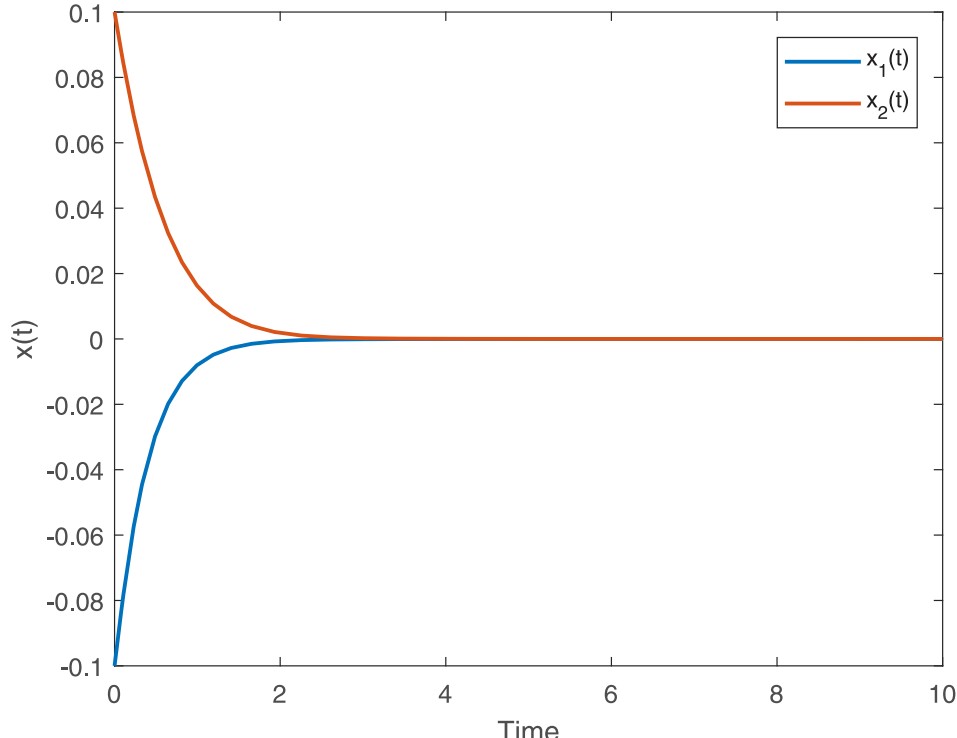

**Fig 2. State trajectory of T-S fuzzy dynamical system for Example 4.**

In this example, we used to discuss the stability criterion and passivity performance of the T-S fuzzy for neutral differential system (1) where $D_i + \Delta D_i(t) = 0$. LMIs in Theorem 4 is solved by MATLAB control toolbox to obtain the largest allowable bounds of $\sigma_2$ for dissimilar values of $\tau_d$, $\alpha$, $\sigma_d$ in example 5 are shown in Table 5.

**Example 6** Analyze the uncertainty of T-S fuzzy dynamic system presented in [30] by the parameters as following:

$$\begin{cases} \dot{x}(t) & = Ax(t) + Bx(t - \sigma(t)) + C(\dot{x} - \tau(t)) \\ x(t) & = \phi(t), t \in [-n, 0], n = \max\{\tau_2, \sigma_2\}, \end{cases}$$

where

$$A_1 = \begin{bmatrix} -0.9 & 0.2 \\ 0.1 & -0.9 \end{bmatrix}, \quad A_2 = \begin{bmatrix} -1 & 1 \\ 1.5 & -2 \end{bmatrix}, \quad B_1 = \begin{bmatrix} -1.1 & -0.2 \\ 0.1 & -1.1 \end{bmatrix}, \quad B_2 = \begin{bmatrix} -1 & -0.6 \\ 0.5 & -1.2 \end{bmatrix},$$

$$C_1 = \begin{bmatrix} -0.2 & 0 \\ 0.2 & -0.1 \end{bmatrix}, \quad C_2 = \begin{bmatrix} 0.2 & 0.1 \\ -0.4 & 0.8 \end{bmatrix}, \quad H_{11} = \begin{bmatrix} 1.6 & 0 \\ 0 & 0.05 \end{bmatrix}, \quad H_{12} = \begin{bmatrix} 1.6 & 0 \\ 0 & -0.05 \end{bmatrix},$$

$$H_{21} = \begin{bmatrix} 0.1 & 0 \\ 0 & 0.3 \end{bmatrix}, \quad H_{22} = \begin{bmatrix} 0.1 & 0 \\ 0 & 0.3 \end{bmatrix}, \quad H_{31} = \begin{bmatrix} 0.5 & 0 \\ 0 & 0.2 \end{bmatrix}, \quad H_{32} = \begin{bmatrix} 0.4 & 0 \\ 0 & 0.3 \end{bmatrix},$$

$$F = \begin{bmatrix} 0.1459 & 0 \\ 0 & 0.1459 \end{bmatrix}, \quad I = \begin{bmatrix} 1 & 0 \\ 0 & 1 \end{bmatrix}.$$

**Table 5. The maximum allowable bounds of $\sigma_2$ for $\tau_d$, $\sigma_d$ and $\alpha$ of Example 5.**

| $\tau_d = \sigma_d$ | $\alpha = 0.5$ | $\alpha = 0.4$ | $\alpha = 0.3$ | $\alpha = 0.2$ | $\alpha = 0.1$ | $\alpha = 0$ |
|---|---|---|---|---|---|---|
| 0.5 | 0.0719 | 0.0797 | 0.0874 | 0.0951 | 0.1028 | 0.1105 |

**Table 6. The maximum allowable bounds of $\sigma_2$ for $\tau$ of Example 6.**

| Methods | $\tau = 0.2445$ |
|---|---|
| X. Ding, L. Shu, and C. Xiang [30] | 0.2450 |
| Corollary 3 | 0.5021 |

*The purpose of example 6 is compare the largest allowable bounds for tolerable delays of $\sigma(t)$ which ensure the exponential stability with the fuzzy convergent rate $\alpha = 0.23$ of the T-S fuzzy dynamic system above. Based on* Table 6, *the results present-day available for comparison purposes are recorded. This present the proposed method is less conservative than the immemorial method.*

**Remark 7** *New results on robust exponential stability of Takagi–Sugeno fuzzy for neutral differential systems with mixed time-varying delays* [40] *only focus on exponential stability for neutral differential equations of the uncertain Takagi—Sugeno fuzzy system. This paper studies exponential stable and exponentially passive neutral differential equations of the uncertain Takagi—Sugeno fuzzy system for further improvement. Furthermore, we consider mixed interval time-varying delays: mixed interval discrete time-varying delay, interval distributed time-varying delay, and interval neutral time-varying delay, i.e., $\sigma(t)$ is interval discrete time-varying delay which satisfies $0 \leq \sigma_1 \leq \sigma(t) \leq \sigma_2$, $h(t)$ is interval distributed time-varying delays which satisfies $0 \leq h_1 \leq h(t) \leq h_2$, and $\tau(t)$ is interval neutral time-varying delay which satisfy $0 \leq \tau_1 \leq \tau(t) \leq \tau_2$.*

## 5 Conclusion

This study rectifies the exponentially passive analysis of the neutral difference of the uncertain Takagi-Sukeno fuzzy system with neutral, discrete, and distributed interval time-varying delay. By spending the Newton-Leibniz formulas, Lyapunov-Krasovskii Functions (LKF), zero equations, and matrix inequality techniques. The form of linear matrix inequalities (LMIs) is constructed from the exponentially passive, in which the numerical efficiency can be verified. Hence, this study shows the example of numbers to demonstrate the effectiveness of our theoretical results and to illustrate that our results are less conservative than the results available in other works: according to Corollary 2, we get the upper bounds of the time-varying delay $\sigma_2$ for various $\sigma_d$ and $\alpha$. Summarize them in Table 3 for comparison with the results obtained in [7, 26–29]. It is concluded that our results have the upper bounds of the time-varying delay $\sigma_2$ at the amount of 1.1121 for $\sigma_d = 0.9$ and $\alpha = 0.9$. Moreover, in Corollary 3, we get the upper bounds of the time-varying delay $\sigma_2$. Summarize them in Table 6 for comparison with the results obtained in [30]. It is concluded that our results have the upper bounds of the time-varying delay $\sigma_2$ at the amount of 0.5021 for $\tau = 0.2450$ and $\alpha = 0.23$. We can see that our obtained results are less conservative than some existing results. In future work, the derived results and methods in this paper are expected to be applied to other systems such as fuzzy generalized complex-valued, guaranteed cost, pinning control of neural networks for impulsive effects on stability and passivity analysis, and so on [41–45].

## Author Contributions

**Conceptualization:** Janejira Tranthi, Thongchai Botmart.

**Data curation:** Janejira Tranthi, Thongchai Botmart.

**Formal analysis:** Janejira Tranthi, Thongchai Botmart.

**Funding acquisition:** Thongchai Botmart.

**Methodology:** Janejira Tranthi, Thongchai Botmart.

**Software:** Thongchai Botmart.

**Supervision:** Janejira Tranthi, Thongchai Botmart.

**Writing – original draft:** Janejira Tranthi, Thongchai Botmart.

**Writing – review & editing:** Thongchai Botmart.

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
