## [Decision Letter · Decision Letter 0]

1 Apr 2022

PONE-D-22-00021A novel criteria on exponentially passive analysis for Takagi-Sugeno fuzzy of neutral dynamic system with various time-varying delaysPLOS ONE

Dear Dr. Botmart,

Thank you for submitting your manuscript to PLOS ONE. After careful consideration, we feel that it has merit but does not fully meet PLOS ONE’s publication criteria as it currently stands. Therefore, we invite you to submit a revised version of the manuscript that addresses the points raised during the review process.

We look forward to receiving your revised manuscript.

Kind regards,

Yiming Tang, Ph.D.

Academic Editor

PLOS ONE

Journal Requirements:

“This research has received funding support from the NSRF via the Program Management Unit for Human Resources & Institutional Development, Research and Innovation [grant number B05F640088]”

“NO”

“NO”

Reviewers' comments:

Reviewer's Responses to Questions

**Comments to the Author**

1. Is the manuscript technically sound, and do the data support the conclusions?

Reviewer #1: Yes

Reviewer #2: Yes

2. Has the statistical analysis been performed appropriately and rigorously? 

Reviewer #1: Yes

Reviewer #2: Yes

3. Have the authors made all data underlying the findings in their manuscript fully available?

Reviewer #1: Yes

Reviewer #2: Yes

4. Is the manuscript presented in an intelligible fashion and written in standard English?

Reviewer #1: Yes

Reviewer #2: Yes

5. Review Comments to the Author

Reviewer #1: This study is relevant to designing reliable exponentially passive analysis for T-S fuzzy of dynamic systems with various time-varying delays such as neutral, discrete, and distributed time-varying delays. By using the Lyapunov-Krasovskii function, The Newton–Leibniz theory, the zero equations, and the matrix inequality technique. The multiple delay-dependent criterion with assure exponential stability and passive performance on the discussed T-S fuzzy system, are defined in respect of linear matrix inequalities (LMIs) that can be checked easily using LMI toolbox of MATLAB. A less conservative exponentially passive criterion given for special cases of general stability of a neutral differential system, by those approaches. The results of this study are delay-dependent, which depend on the lower bound and the upper bound with the time-varying delay. Lastly, the examples of Numerical are presented the performance of our criteria with the results obtained summarize some of the previous results as well.

I think the readers of this journal will appreciate the results of this manuscript. Generally speaking, the manuscript is well written, the material is judiciously divided and organized and correct from scientific point of view.

Reviewer #2: In this study, the authors investigate an exponentially passive analysis for T-S fuzzy dynamical systems with various time-varying delays, including neutral, discrete, and distributed time-varying delays using the Lyapunov-Krasovskii function, Newton-Leibniz theory, zero equations, and matrix inequality. As compared to literature, they obtained less conservative results.

The theoretical analysis appears to be correct, but the paper must undergo careful and critical editing of both the English language and English grammar. There are many linguistic omissions, errors, and oversights throughout. There are several such errors scattered throughout the manuscript, so it is very important to check it carefully.

6. PLOS authors have the option to publish the peer review history of their article (what does this mean?). If published, this will include your full peer review and any attached files.

Reviewer #1: No

Reviewer #2: No

---

## [Author Response · Author response to Decision Letter 0]

7 Apr 2022

Original Article Title: “A novel criteria on exponentially passive analysis for Takagi-Sugeno fuzzy of neutral dynamic system with various time-varying delays”

To: PLOS ONE Editor

Re: Response to reviewers

Dear Editor,

Thank you for allowing a resubmission of our manuscript, with an opportunity to address the reviewers’ comments.

We are uploading (a) our point-by-point response to the comments (below) (response to reviewers), (b) an updated manuscript with yellow highlighting indicating changes, and (c) a clean updated manuscript without highlights (PDF main document).

Best regards,

<Thongchai Botmart> et al.

Reviewer #1

 This study is relevant to designing reliable exponentially passive analysis for T-S fuzzy of dynamic systems with various time-varying delays such as neutral, discrete, and distributed time-varying delays. By using the Lyapunov-Krasovskii function, The Newton–Leibniz theory, the zero equations, and the matrix inequality technique. The multiple delay-dependent criterion with assure exponential stability and passive performance on the discussed T-S fuzzy system, are defined in respect of linear matrix inequalities (LMIs) that can be checked easily using LMI toolbox of MATLAB. A less conservative exponentially passive criterion given for special cases of general stability of a neutral differential system, by those approaches. The results of this study are delay-dependent, which depend on the lower bound and the upper bound with the time-varying delay. Lastly, the examples of Numerical are presented the performance of our criteria with the results obtained summarize some of the previous results as well. 

 I think the readers of this journal will appreciate the results of this manuscript. Generally speaking, the manuscript is well written, the material is judiciously divided and organized and correct from scientific point of view. For these reasons I can recommend the acceptance of this paper after some corrections presented below:

Answer : We really appreciate your suggestion. We have revised all of them as the follows:

 It seems that the proof of theorem 1 is so complicated, and for such many complex and large-scale matrices, how to guarantee the existence of the LMI problem for eq. (2).

Author response: The stability criterion of Theorem 1 in the form LMIs (2) can be easily to examine by using LMI toolbox in MATLAB. The improved stability conditions by constructing new Lyapunov functionals is based on LMIs and the dimension of the LMIs depends on the number of premise variables of fuzzy set . Thus, the computational burden problem goes up. This problem is the issue in studying needs of LMI optimization in applied mathematics and the optimization research. Hence, in the further, new techniques to reduce the conservativeness caused by the time-delays such as the delay-fractioning approach and so on. Moreover, the vectors can be solve directly solution of LMI approach. Moreover, the effectiveness of computer should update the performance computer which can be helpful a computational problem.

Author action: -

 The novelty of this paper should be highlighted in the main contributions through comparing it with the existing works.

Author response: We have already rewritten by added the main contribution of our paper.

Author action: We updated the manuscript in paragraph 5 of the introduction by adding the main contribution of our paper as follows:

 • We construct the suitable Lyapunov-Krasovskii functional, which consists of single, double, triple, and quadruple integral terms containing information about the lower and upper bounds of the delays σ_2,τ_2 and a state x(t). Furthermore, the LKF contains a new triple integral term 

σ_2^2 ∫_(-σ_2)^0▒∫_θ^0▒∫_(t+β)^t▒〖e^2α(s-t) x ˙^T (s)W_1 x ˙(s)dsdβdθ〗,τ_2^2 ∫_(-τ_2)^0▒∫_θ^0▒∫_(t+β)^t▒〖e^2α(s-t) x ˙^T (s)W_2 x ˙(s)dsdβdθ〗

and a new quadruple integral term 

σ_2^3 ∫_(-σ_2)^0▒∫_v^0▒∫_θ^0▒∫_(t+β)^t▒〖e^2α(s-t) x ˙^T (s)U_1 x ˙(s)dsdβdθdv〗,τ_2^3 ∫_(-τ_2)^0▒∫_v^0▒∫_θ^0▒∫_(t+β)^t▒〖e^2α(s-t) x ˙^T (s)U_2 x ˙(s)dsdβdθdv〗

that do not appear in [7, 25-29].

 • For the first time, an improved Wirtinger inequality, a new triple integral inequality, and zero equation together with convex combination approach are used in this work; as a result, we obtain more general results and maximum allowable delay bounds greater than in previous literature [7, 25-29].

 3. The linguist should be thoroughly examined and corrected. Numerous typos and grammatical errors appear in the manuscript. For example,

 3.1 On page 11, line 14, "-D∫_(t-h(t))^t▒〖x(s)ds- ├ ├ Eu(t))]0〗 " should be "-D∫_(t-h(t))^t▒〖x(s)ds- ├ ├ Eu(t))] 〗 ".

 3.2 On page 19, line 143, “since lemma 5” should be “Since lemma 5”.

 3.3 On page 29, line 181, "D+∆D_i (t)=0 and E+∆E_i (t)=0" should be "D_i+∆D_i (t)=0 and E_i+∆E_i (t)=0".

 3.4. On page 33, line 214, "C+∆C_i (t)=0, D+∆D_i (t)=0 and E+∆E_i (t)=0" should be "C_i+∆C_i (t)=0, D_i+∆D_i (t)=0 and E_i+∆E_i (t)=0".

Author response: Thanks for the suggestion we have already and carefully rechecked and revised the entire paper.

Author action: We have corrected the manuscript by rewriting it as follows:

 1. On page 1, line 4, “exponential stability and passive performance” should be “exponentially passive”. 

 2. On page 3, line 105, “exponential stability and passivity criteria” should be “exponentially passive criteria”.

 3. On page 3, line 107, “exponential stability and passive performance” should be “exponentially passive”.

 4. On page 3, line 109, “the general phase value system of the neutral differential system” should be “the general phase value system for the neutral dynamic system”.

 5. On page 3, line 112, “the neutral differential system” should be “the neutral dynamic system”.

 6. On page 11, line 14, "-D∫_(t-h(t))^t▒〖x(s)ds- ├ ├ Eu(t))]0〗 " should be "-D∫_(t-h(t))^t▒〖x(s)ds- ├ ├ Eu(t))] 〗 ".

 7. On page 12, line 8, "[[x(t)"┤┤ should be "[x(t)┤".

 8. On page 12, line 9, "[[x(t)"┤┤ should be "[x(t)┤".

 9. On page 12, line 16, "[[x(t)"┤┤ should be "[x(t)┤".

 10. On page 12, line 17, "[[x(t)"┤┤ should be "[x(t)┤".

 11. On page 12, line 21, "[[x(t)"┤┤ should be "[x(t)┤".

 12. On page 12, line 22, "[[x(t)"┤┤ should be "[x(t)┤".

 13. On page 13, line 7, "[[x(t)"┤┤ should be "[x(t)┤".

 14. On page 13, line 8, "[[x(t)"┤┤ should be "[x(t)┤".

 15. On page 19, line 143, “since lemma 5” should be “Since lemma 5”.

 16. On page 29, line 181, "D+∆D_i (t)=0 and E+∆E_i (t)=0" should be "D_i+∆D_i (t)=0 and E_i+∆E_i (t)=0".

 17. On page 30, line 194, “exponential stability criterion and passivity performance” should be “exponentially passive”.

 18. On page 33, line 214, "C+∆C_i (t)=0, D+∆D_i (t)=0 and E+∆E_i (t)=0" should be "C_i+∆C_i (t)=0, D_i+∆D_i (t)=0 and E_i+∆E_i (t)=0".

4. What is “interval discrete time-varying delay”? What are the differences between it and time-varying delay? Besides, what is “mixed interval time-varying delays”?

Author response: 

-Interval discrete time-varying delay is interval time-varying delay and the lower bound is not restricted to zero which is the state variable.

-Time-varying delay is the delay which depends on time such as state delay, distributed delay, or interval time-varying delay.

-In this work, mixed interval time-varying delays are the delays which are mixed interval discrete time-varying delay, interval distributed time-varying delay and interval neutral time-varying delay i.e., σ(t)is interval discrete time-varying delay which satisfies 0≤σ_1≤σ(t) ≤ σ_2, h(t) is interval distributed time-varying delays which satisfy 0≤h_1≤h(t)≤h_2. and τ(t) is interval neutral time-varying delay which satisfy 0≤τ_1≤τ(t)≤τ_2.

Author action: -

5. The main conclusions of this paper are given in the form of LMI, while, the LMI derived in this paper contains so many information of the system, which is very difficult to verify, so some simple method should be considered, i.e., a simple Lyapunov-Krasovskii functional should be considered.

Author response: In this work, the Lyapunov–Krasovskii functional contains single, double, triple, and quadruple integral terms, in which full information on the delays and a state variable is used. Therefore, the construction and the technique for computation of the Lyapunov–Krasovskii functional are the main keys to improve results of this work. In the proof of Theorems 1 improved Wirtinger’s single integral inequality, a novel triple integral inequality, and convex combination technique are used to bound the derivative of Lyapunov– Krasovskii functional, which provide tighter bound than the inequalities. All of these lead to a reduction of the conservatism of our results compared to those in some exiting works and, in particular, numerical examples. However, the complex computation of the Lyapunov–Krasovskii functional leads to the LMI derived in this work which contains much information about the system. Hence, for further work, it is interesting for researchers to improve the technique for a simple Lyapunov–Krasovskii functional and also achieve better results

Author action: -

6. Based on the obtained results, some potential improvements or future work could be discussed in the Conclusion.

Author response: We have already rewritten in paragraph 3 of the introduction and the conclusion of our paper.

Author action: We have updated the manuscript by rewritten in paragraph 3 of the introduction and the conclusion. In addition, we added the six references as follows:

“In paragraph 3 of the introduction

In addition, passivity theory is another proficient tool for analyzing system stability. The passive theory is the main pointed to the system can keep the systems internal stability which is the passive properties. So that, the problem of inactivity is therefore an important part of recent research. Then, the passive control uses the product of output and input as the power rating, which captures the attenuation properties of the system under the bounded input. In particular, passivity theory is more general than stability theory because it can be illustrated Lyapunov function under the theory of stabilization. This theory is used for issue of engineering i.e. electric circuits and heat energy systems. Nowadays many researchers have studied passive theory and passive control problems extensively, for instance, Zhang et al. [11] who studied the passive controller design issue with both state and input delays for a class of continuous-time T-S fuzzy systems. Another researcher such as Wu et al. [12] identified the problem of passive control for fuzzy network systems, considering the random uncertainty variable sampling interval and the delay caused by the fixed network. Similarly, Song and He [13] who researched the robust passive control is offered for a limited time for nonlinear systems with time-delay. The studied of Yu et al. [14] focused both passive analysis and passive control for erratic intermittent switching delay systems through a simple switching signal design. Likewise, Yotha et al. Improved delay-dependent approach to passivity analysis for uncertain neural networks with discrete interval and distributed time-varying delays [15]. So, it is challenging to solve exponentially passive for T-S fuzzy of neutral dynamic system with various time-varying delays.”

“Conclusion

This study rectify the exponentially passive analysis for the neutral difference of the uncertain Takagi-Sukeno fuzzy system with neutral, discrete and distributed interval time-varying delayed. By spending the Newton-Leibniz formulas, Lyapunov-Krasovskii functions (LKF), zero equations and matrix inequality techniques. The form of linear matrix inequalities (LMIs) is constructed from the exponentially passive which the numerical efficiency can be verified. Hence, this study shows the example of numbers to demonstrate the effectiveness of our theoretical results, and to illustrate, the results we have obtained are less conservative than the results available in other works. In the future work, the derived results and methods in this paper are expected to be applied to other systems such as fuzzy generalized complex-valued, guaranteed cost, pinning control of neural networks for impulsive effects on stability and passivity analysis, and so on [40–44].”

Reviewer #2: In this study, the authors investigate an exponentially passive analysis for T-S fuzzy dynamical systems with various time-varying delays, including neutral, discrete, and distributed time-varying delays using the Lyapunov-Krasovskii function, Newton-Leibniz theory, zero equations, and matrix inequality. As compared to literature, they obtained less conservative results.

The theoretical analysis appears to be correct, but the paper must undergo careful and critical editing of both the English language and English grammar. There are many linguistic omissions, errors, and oversights throughout. There are several such errors scattered throughout the 

manuscript, so it is very important to check it carefully.

Answer : We really appreciate your suggestion. We have checked the grammatical problems throughout the manuscript thoroughly already.

---

## [Decision Letter · Decision Letter 1]

30 Jun 2022

PONE-D-22-00021R1A novel criteria on exponentially passive analysis for Takagi-Sugeno fuzzy of neutral dynamic system with various time-varying delaysPLOS ONE

Dear Dr. Botmart,

Thank you for submitting your manuscript to PLOS ONE. After careful consideration, we feel that it has merit but does not fully meet PLOS ONE’s publication criteria as it currently stands. Therefore, we invite you to submit a revised version of the manuscript that addresses the points raised during the review process.

We look forward to receiving your revised manuscript.

Kind regards,

Yiming Tang, Ph.D.

Academic Editor

PLOS ONE

Journal Requirements:

Reviewers' comments:

Reviewer's Responses to Questions

**Comments to the Author**

1. If the authors have adequately addressed your comments raised in a previous round of review and you feel that this manuscript is now acceptable for publication, you may indicate that here to bypass the “Comments to the Author” section, enter your conflict of interest statement in the “Confidential to Editor” section, and submit your "Accept" recommendation.

Reviewer #2: (No Response)

Reviewer #3: (No Response)

2. Is the manuscript technically sound, and do the data support the conclusions?

Reviewer #2: Yes

Reviewer #3: Yes

3. Has the statistical analysis been performed appropriately and rigorously? 

Reviewer #2: I Don't Know

Reviewer #3: Yes

4. Have the authors made all data underlying the findings in their manuscript fully available?

Reviewer #2: Yes

Reviewer #3: Yes

5. Is the manuscript presented in an intelligible fashion and written in standard English?

Reviewer #2: No

Reviewer #3: Yes

6. Review Comments to the Author

Reviewer #2: The authors have improved the revised manuscript based on the reviewers' comments, but some additional corrections are needed. Please see the attached file.

Reviewer #3: 1. The abstract needs to be written in a more clarified way indicating the contribution.

2. The authors have to state the weakness in previous works which motivated this study.

3. The authors did not explain the difference between the work in the following paper [New results on robust exponential stability of

Takagi–Sugeno fuzzy for neutral differential systems with mixed time-varying delays] and the work in this paper.

4. The novelty of this study has been explicitly stated.

5. What is the research object in this paper?

6. The format and English writing of the paper should be improved.

7. The conclusion is descriptive. It is void of quantitative and numerical improvement and comparison.

7. PLOS authors have the option to publish the peer review history of their article (what does this mean?). If published, this will include your full peer review and any attached files.

Reviewer #2: No

Reviewer #3: No

---

## [Author Response · Author response to Decision Letter 1]

26 Jul 2022

Original Manuscript ID: PONE-D-22-00021R1

Original Article Title: “A novel criteria on exponentially passive analysis for Takagi-Sugeno fuzzy of neutral dynamic system with various time-varying delays”

To: PLOS ONE Editor

Re: Response to reviewers

Dear Editor,

Thank you for allowing a resubmission of our manuscript, with an opportunity to address the reviewers’ comments.

We are uploading (a) our point-by-point response to the comments (below) (response to reviewers), (b) an updated manuscript with yellow highlighting indicating changes, and (c) a clean updated manuscript without highlights (PDF main document).

Best regards,

<Thongchai Botmart> et al.

Reviewer #2: The authors have improved the revised manuscript based on the reviewers' comments, but some additional corrections are needed. Please see the attached file.

Answer : We appreciate your suggestion. We revised as the suggestion already and the English of the whole paper is carefully checked in the revised paper.

Reviewer #3:

1. The abstract needs to be written in a more clarified way indicating the contribution.

Author response: We have already rewritten the abstract.

Author action: We updated the manuscript in the abstract part as follows:

 This paper is the first studying on designing exponentially passive analysis for T-S fuzzy of dynamic systems with various time-varying delays such as neutral, discrete, and distributed time-varying delays. Constructing the new Lyapunov-Krasovskii function and the Newton-Leibniz theory, the zero equations, and the matrix inequality techniques, the multiple delay-dependent criteria, with assuring exponentially passive on the discussed T-S fuzzy system, are defined in respect of linear matrix inequalities (LMIs) that can be checked easily using the LMI toolbox of MATLAB. Those approaches give less conservative, exponentially passive criteria for special cases of general stability of a neutral differential system. Furthermore, the results of this study are delay-dependent, which depend on the lower and upper bound with the time-varying delay. Lastly, some numerical examples illustrate the performance of our criteria based on the results obtained and summarize some of the previous achievements.

2. The authors have to state the weakness in previous works which motivated this study.

Author response: Many thanks for your valuable remarks. 

Author action: We have revised in introduction part and added a Remark 1. as follows the above mentioned, there is still room for further improvement: the fuzzy T-S method with delay-dependent based on latency to the possible extent of the thresholds for exponential stability and passivity performance.

 Remark 1 This study constructs the suitable Lyapunov-Krasovskii functional, which consists of single, double, triple, and quadruple integral terms containing information about the lower and upper bounds of the delays σ_2,τ_2 and a state x(t). Furthermore, the LKF contains a new triple integral term as follows:

σ_2^2 ∫_(-σ_2)^0▒∫_θ^0▒∫_(t+β)^t▒〖e^2α(s-t) x ˙^T (s)W_1 x ˙(s)dsdβdθ〗,τ_2^2 ∫_(-τ_2)^0▒∫_θ^0▒∫_(t+β)^t▒〖e^2α(s-t) x ˙^T (s)W_2 x ˙(s)dsdβdθ〗

and a new quadruple integral term 

σ_2^3 ∫_(-σ_2)^0▒∫_v^0▒∫_θ^0▒∫_(t+β)^t▒〖e^2α(s-t) x ˙^T (s)U_1 x ˙(s)dsdβdθdv〗,τ_2^3 ∫_(-τ_2)^0▒∫_v^0▒∫_θ^0▒∫_(t+β)^t▒〖e^2α(s-t) x ˙^T (s)U_2 x ˙(s)dsdβdθdv〗

that do not appear in [7, 25-29]. These improvement techniques enhance to get better results.

3. The authors did not explain the difference between the work in the following paper [New results on robust exponential stability of Takagi–Sugeno fuzzy for neutral differential systems with mixed time-varying delays] and the work in this paper.

Author response: Many thanks for your valuable suggestion. We have added in the Remark 7 and add referent.

Author action: 

 Remark 7 New results on robust exponential stability of Takagi - Sugeno fuzzy for neutral differential systems with mixed time-varying delays [40] only focus on exponential stability for neutral differential equations of the uncertain Takagi - Sugeno fuzzy system. This paper studies exponential stable and exponentially passive neutral differential equations of the uncertain Takagi - Sugeno fuzzy system for further improvement. Furthermore, we consider mixed interval time-varying delays: mixed interval discrete time-varying delay, interval distributed time-varying delay, and interval neutral time-varying delay, i.e., σ(t) interval discrete time-varying delay which satisfies 0≤σ_1≤σ(t)≤σ_2, h(t) is interval distributed time-varying delays which satisfy 0≤h_1 ≤ h(t)≤h_2. and τ(t) is interval neutral time-varying delay which satisfy 0≤τ_1≤τ(t) ≤ τ_2. 

 [40] Tranthi J, Botmart T, Weera W, La-inchua T, Pinjai S. New results on robust exponential stability of Takagi - Sugeno fuzzy for neutral differential systems with mixed time-varying delays. Math Comput Simul. 2022; 201 : 714-738.

4. The novelty of this study has been explicitly stated.

Author response: We appreciate your compliments.

Author action: -

5. What is the research object in this paper?

Author response: this paper investigates the delay-dependent approach to exponential passive of T-S fuzzy for uncertain neutral dynamical system with discrete interval and distributed time-varying delays. Based on delay partitioning, a LKF is constructed to obtain several improved delay-dependent passivity conditions which guarantee the passivity of T-S fuzzy for uncertain neutral dynamical system. We consider the additional useful terms with the distributed delays and estimate some integral terms by Wirtinger’s inequality provided a tighter lower bound than Jensen’s inequality. The Lyapunov–Krasovskii functional contains single, double, triple, and quadruple integral terms, in which full information on the delays and a state variable is used. Therefore, the construction and the technique for computation of the Lyapunov–Krasovskii functional are the main keys to improving the results of this work. In the proof of Theorems 1 improved Wirtinger’s single integral inequality, a novel triple integral inequality, and convex combination technique are used to bound the derivative of Lyapunov– Krasovskii functional, which provide a tighter bound than the inequalities. All of these lead to a reduction of the conservatism of our results compared to those in some exciting works and, in particular, numerical examples. However, the complex computation of the Lyapunov–Krasovskii functional leads to the LMI derived in this work which contains much information about the system. Hence, for further work, it is interesting for researchers to improve the technique for a simple Lyapunov–Krasovskii functional and also achieve better results.

Author action: -

6. The format and English writing of the paper should be improved.

Author response: We really appreciate your suggestion. We have checked the grammatical problems already in this paper.

Author action: We have corrected the manuscript by rewriting it as follows:

 1. On page 1, line 3 “The Newton-Leibniz theory” should be “the Newton-Leibniz theory”

 2. On page 1, line 10 “Lastly, the examples of Numerical are presented the performance of our criteria with the results obtained summarize some of the previous results as well.” should be “Lastly, some numerical examples illustrate the performance of our criteria base on the results obtained and summarize some of the previous achievement as well.”

 3. On page 3, line 95 “Fang Liu, et al. [?], Li et al. [?], Lien et al. [7, 28] and Pin-Lin Liu [?].” should be “by Fang Liu, et al. [26], Li et al. [27], Lien et al. [7, 28] and Pin-Lin Liu [29].”

 4. On page 3, line 106 “that do not appear in [?,?,?, 7, 28, 30].” should be “that do not appear in [7, 26{30].”

 5. On page 4, line 110 “previous literature [?,?,?, 7, 28, 30].” should be “previous literature [7, 26{30].”

 6. On page 5, line 133 “Definition 1 [?] The system (1)” should be “Definition 1 [31] The system (1)”

 7. On page 6, line 1 “Lemma 1 [?]” should be “Lemma 1 [32]”

 8. On page 6, line 6 “Lemma 2 [?], [?]” should be “Lemma 2 [33], [34]”

 9. On page 6, line 7 “the following inequality hold:” should be “the following inequality holds:”

 10. On page 6, line 14 “Lemma 3 [?]” should be “Lemma 3 [35]”

 11. On page 6, line 15 “the following inequality hold:” should be “the following inequality holds:”

 12. On page 6, line 21 “Lemma 4 [?]” should be “Lemma 4 [36]”

 13. On page 6, line 22 “the following inequality hold:” should be “the following inequality holds:”

 14. On page 18, line 142 “The proof is completes” should be “The proof is completed”

 15. On page 20, line 155 “exponential stability” should be “exponential stable”

7. The conclusion is descriptive. It is void of quantitative and numerical improvement and comparison.

Author response: We appreciate your valuable remark. We have already rewritten the conclusion part.

Author action: We updated the manuscript as follows:

 This study rectifies the exponentially passive analysis of the neutral difference of the uncertain Takagi-Sukeno fuzzy system with neutral, discrete, and distributed interval time-varying delay. By spending the Newton-Leibniz formulas, Lyapunov-Krasovskii Functions (LKF), zero equations, and matrix inequality techniques. The form of linear matrix inequalities (LMIs) is constructed from the exponentially passive, in which the numerical efficiency can be verified. Hence, this study shows the example of numbers to demonstrate the effectiveness of our theoretical results and to illustrate the results we have obtained are less conservative than the results available in other works: according to Corollary 2, we get the upper bounds of the time-varying delay σ_2for various σ_d and α. Summarize them in Table 3 for comparison with the results obtained in [7, 26–29]. It is concluded that our results have the upper bounds of the time-varying delay σ_2 at the amount of 1.1121 for σ_d = 0.9 and α = 0.9. Moreover, in Corollary 3, we get the upper bounds of the time-varying delay〖 σ〗_2. Summarize them in Table 6 for comparison with the results obtained in [30]. It is concluded that our results have the upper bounds of the time-varying delay σ_2 at the amount of 0.5021 for τ= 0.2450 and α = 0.23. We can see that our obtained results are less conservative than some existing results. In future work, the derived results and methods in this paper are expected to be applied to other systems such as fuzzy generalized complex-valued, guaranteed cost, pinning control of neural networks for impulsive effects on stability and passivity analysis, and so on [41-45].

Finally, authors are once again thankful to anonymous reviewer, editor in chief, associate editor and editorial staff for their time, help, efforts and support.

Corresponding author

Thongchai Botmart

---

## [Decision Letter · Decision Letter 2]

23 Aug 2022

PONE-D-22-00021R2A novel criteria on exponentially passive analysis for Takagi-Sugeno fuzzy of neutral dynamic system with various time-varying delaysPLOS ONE

Dear Dr. Botmart,

Thank you for submitting your manuscript to PLOS ONE. After careful consideration, we feel that it has merit but does not fully meet PLOS ONE’s publication criteria as it currently stands. 

We feel that the manuscript requires additional copyediting before it can be published. As such, we ask that you thoroughly copyedit the manuscript at this point.

We look forward to receiving your revised manuscript.

Kind regards,

Hanna Landenmark

Staff Editor, PLOS ONE

on behalf of

Yiming Tang

Journal Requirements:

Reviewers' comments:

Reviewer's Responses to Questions

**Comments to the Author**

1. If the authors have adequately addressed your comments raised in a previous round of review and you feel that this manuscript is now acceptable for publication, you may indicate that here to bypass the “Comments to the Author” section, enter your conflict of interest statement in the “Confidential to Editor” section, and submit your "Accept" recommendation.

Reviewer #2: All comments have been addressed

Reviewer #3: All comments have been addressed

2. Is the manuscript technically sound, and do the data support the conclusions?

Reviewer #2: Yes

Reviewer #3: Yes

3. Has the statistical analysis been performed appropriately and rigorously? 

Reviewer #2: I Don't Know

Reviewer #3: Yes

4. Have the authors made all data underlying the findings in their manuscript fully available?

Reviewer #2: Yes

Reviewer #3: Yes

5. Is the manuscript presented in an intelligible fashion and written in standard English?

Reviewer #2: Yes

Reviewer #3: Yes

6. Review Comments to the Author

Reviewer #2: In response to the reviewers' comments, the authors improved their revised manuscript, and now it can be published.

Reviewer #3: The authors have improved the revised manuscript based on the reviewers' comments, All my comments have been addressed—no further comments.

7. PLOS authors have the option to publish the peer review history of their article (what does this mean?). If published, this will include your full peer review and any attached files.

Reviewer #2: No

Reviewer #3: No

---

## [Author Response · Author response to Decision Letter 2]

27 Aug 2022

Original Manuscript ID: PONE-D-22-00021R2

Original Article Title: “A novel criteria on exponentially passive analysis for Takagi-Sugeno fuzzy of neutral dynamic system with various time-varying delays”

To: PLOS ONE Editor

Re: Response to reviewers

Dear Editor,

Thank you for allowing a resubmission of our manuscript, with an opportunity to address the reviewers’ comments.

We are uploading (a) our point-by-point response to the comments (below) (response to reviewers), (b) an updated manuscript with yellow highlighting indicating changes, and (c) a clean updated manuscript without highlights (PDF main document).

Best regards,

<Thongchai Botmart> et al.

Journal Requirements: Please review your reference list to ensure that it is complete and correct. If you have cited papers that have been retracted, please include the rationale for doing so in the manuscript text, or remove these references and replace them with relevant current references. Any changes to the reference list should be mentioned in the rebuttal letter that accompanies your revised manuscript. If you need to cite a retracted article, indicate the article’s retracted status in the References list and also include a citation and full reference for the retraction notice.

Answer: We have added the referent [40] because a reviewer requires us to compare the difference between the work in the following paper [New results on robust exponential stability of Takagi–Sugeno fuzzy for neutral differential systems with mixed time-varying delays] and the work in this paper. I.e., New results on robust exponential stability of Takagi - Sugeno fuzzy for neutral differential systems with mixed time-varying delays [40] only focus on exponential stability for neutral differential equations of the uncertain Takagi - Sugeno fuzzy system. This paper studies exponential stable and exponentially passive neutral differential equations of the uncertain Takagi - Sugeno fuzzy system for further improvement. Furthermore, we consider mixed interval time-varying delays: mixed interval discrete time-varying delay, interval distributed time-varying delay, and interval neutral time-varying delay, i.e., σ(t) is interval discrete time-varying delay which satisfies 0≤σ_1≤σ(t)≤σ_2, h(t) is interval distributed time-varying delays which satisfy 0≤h_1≤h(t)≤h_2. And τ(t) is an interval neutral time-varying delay that satisfies 0≤τ_1≤τ(t)≤τ_2. 

 [40] Tranthi J, Botmart T, Weera W, La-inchua T, Pinjai S. New results on robust exponential stability of Takagi - Sugeno fuzzy for neutral differential systems with mixed time-varying delays. Math Comput Simul. 2022; 201 : 714-738.

Review Comments to the Author

Reviewer #2: In response to the reviewers' comments, the authors improved their revised manuscript, and now it can be published.

Answer: we greatly appreciate your kind comments and acceptance for our manuscript.

Reviewer #3: The authors have improved the revised manuscript based on the reviewers' comments, All my comments have been addressed—no further comments.

Answer: we greatly appreciate your kind comments and acceptance for our manuscript.

---

## [Decision Letter · Decision Letter 3]

12 Sep 2022

A novel criteria on exponentially passive analysis for Takagi-Sugeno fuzzy of neutral dynamic system with various time-varying delays

PONE-D-22-00021R3

Dear Dr. Botmart,

We’re pleased to inform you that your manuscript has been judged scientifically suitable for publication and will be formally accepted for publication once it meets all outstanding technical requirements.

Kind regards,

Yiming Tang, Ph.D.

Academic Editor

PLOS ONE

Additional Editor Comments (optional):

Reviewers' comments:

Reviewer's Responses to Questions

**Comments to the Author**

1. If the authors have adequately addressed your comments raised in a previous round of review and you feel that this manuscript is now acceptable for publication, you may indicate that here to bypass the “Comments to the Author” section, enter your conflict of interest statement in the “Confidential to Editor” section, and submit your "Accept" recommendation.

Reviewer #2: All comments have been addressed

Reviewer #3: All comments have been addressed

2. Is the manuscript technically sound, and do the data support the conclusions?

Reviewer #2: Yes

Reviewer #3: Yes

3. Has the statistical analysis been performed appropriately and rigorously? 

Reviewer #2: I Don't Know

Reviewer #3: Yes

4. Have the authors made all data underlying the findings in their manuscript fully available?

Reviewer #2: Yes

Reviewer #3: Yes

5. Is the manuscript presented in an intelligible fashion and written in standard English?

Reviewer #2: Yes

Reviewer #3: Yes

6. Review Comments to the Author

Reviewer #2: The authors have revised the paper in response to reviewers' comments. Now the paper can be accepted.

Reviewer #3: Dear Authors:

Thanks for your efforts to make the required corrections. All my previous comments have be addressed, and I have no more. The manuscript can be accepted in the present form.

7. PLOS authors have the option to publish the peer review history of their article (what does this mean?). If published, this will include your full peer review and any attached files.

Reviewer #2: No

Reviewer #3: No

---

## [Editor Report · Acceptance letter]

27 Sep 2022

PONE-D-22-00021R3 

A novel criteria on exponentially passive analysis for Takagi-Sugeno fuzzy of neutral dynamic system with various time-varying delays 

Dear Dr. Botmart:

I'm pleased to inform you that your manuscript has been deemed suitable for publication in PLOS ONE. Congratulations! Your manuscript is now with our production department. 

Kind regards, 

on behalf of

Professor Yiming Tang 

Academic Editor

PLOS ONE